# On the Role of Information Structure in Reinforcement Learning for Partially-Observable Sequential Teams and Games

**Awni Altabaa**
Department of Statistics & Data Science
Yale University
awni.altabaa@yale.edu

**Zhuoran Yang**
Department of Statistics & Data Science
Yale University
zhuoran.yang@yale.edu

## Abstract

In sequential decision-making problems, the *information structure* describes the causal dependencies between system variables, encompassing the dynamics of the environment and the agents' actions. Classical models of reinforcement learning (e.g., MDPs, POMDPs) assume a restricted and highly regular information structure, while more general models like predictive state representations do not explicitly model the information structure. By contrast, real-world sequential decision-making problems typically involve a complex and time-varying interdependence of system variables, requiring a rich and flexible representation of information structure. In this paper, we formalize a novel reinforcement learning model which explicitly represents the information structure. We then use this model to carry out an information-structural analysis of the statistical complexity of general sequential decision-making problems, obtaining a characterization via a graph-theoretic quantity of the DAG representation of the information structure. We prove an upper bound on the sample complexity of learning a general sequential decision-making problem in terms of its information structure by exhibiting an algorithm achieving the upper bound. This recovers known tractability results and gives a novel perspective on reinforcement learning in general sequential decision-making problems, providing a systematic way of identifying new tractable classes of problems.

## 1   Introduction

The *information structure* of a sequential decision-making problem is a description of the causal dependencies between system variables. In particular, the information structure specifies the subset of past events that causally influence the present state. This includes information affecting system dynamics and information available to each agent when choosing an action. The control community has long recognized the importance of information structure, leading to the development of the celebrated Witsenhausen intrinsic model [1], and extensive study since the 1970s [e.g., 2–14].

In general, sequential decision-making problems with arbitrary causal dependencies between system variables can be computationally and statistically intractable in both control and learning settings [3, 15]. In response, early research has identified classes of tractable information structures. For example, Markov Decision Processes (MDP) and Markov teams/games assume an observable Markovian state, which serves as a sufficient statistic for the system's evolution. These structural assumptions enable practical planning and learning algorithms, for example based on dynamic programming [16, 17]. Although such restricted model classes can enable more fruitful analysis, they lack the expressiveness needed to naturally capture the causal structures of complex real-world sequential decision-making problems, where each event in the system may depend arbitrarily on past events.

38th Conference on Neural Information Processing Systems (NeurIPS 2024).

The concept of information structure is also fundamental to studying the phenomenon of *partial observability*. In general, partial observability refers to situations where a system's evolution depends on a potentially large number of sequential events, but only a subset of these events is observable by the learning agent. However, commonly studied models capture only a restrictive form of this phenomenon. For example, in a Partially-Observable Markov Decision Process (POMDP)—the standard model of partial observability —it is assumed that a Markovian state exists and that each observation is a noisy measurement of the current state. This assumption is often unrealistic, as general systems may not have clearly defined "states", and observations may be generated by more complex dependencies. Information structure provides a more powerful framework for understanding partial observability, capturing a general notion in which system variables evolve with an arbitrary causal structure (not necessarily Markovian), and the set of observables is any subset of all system variables.

The highly-regular information structures of classical models make analysis more tractable, enabling favorable theoretical results [e.g., 18–22] and driving notable empirical success across a wide range of domains  [e.g., 23–28]. Despite this empirical success, a general theory addressing the role of information structure in the statistical aspects of reinforcement learning is missing. We argue for the perspective that information structure is an important component of analyzing and solving reinforcement learning problems. A rich and flexible representation of information structure is essential to faithfully capture real-world sequential decision-making problems, where the system evolves according to a complex and time-varying dependence on the past, and different agents have different information available to them at different points in time.

***In this work, we formulate a general model of sequential decision-making with an explicit representation of information structure, and study the role of information structure in the statistical complexity of reinforcement learning.*** Explicitly modeling information structure allows us to identify a broader class of tractable decision-making problems and ultimately develop more tailored reinforcement learning approaches that leverage this structure.

**Summary of Contributions**

*A model for representing information structure.* We propose *partially-observable sequential teams* (POST) and *partially-observable sequential games* (POSG) as general models that contain an explicit representation of information structure as part of the problem specification. This forms a unifying framework that enables an analysis of the role of information structure in RL and captures commonly studied RL models as special cases, including MDPs, Markov teams/games, POMDPs, and Dec-POMDPs/POMGs (Figure 1). The models introduced in this work draw inspiration from Witsenhausen's intrinsic model [1] and its variants from the control literature [29, 30], with added elements to model partial observability in the context of reinforcement learning.

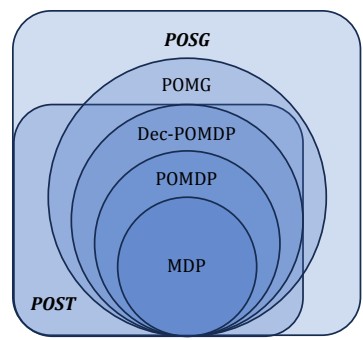

Figure 1: A depiction of the generality of our proposed models. POSTs and POSGs capture MDPs, POMDPs, Dec-POMDPs, and POMGs as special cases.

*Theoretical analysis of sequential decision-making through information structure.* We characterize the complexity of the observable dynamics of any sequential decision-making problem through a graph-theoretic analysis of the information structure. Moreover, we propose a generalization of predictive state representations—which may be of independent interest—and construct such a representation for POSTs/POSGs by exploiting information structure. This provides a robust and efficient parameterization amenable to learning.

*Learning theory & information-structural analysis of statistical complexity.* We prove an upper bound on the sample complexity of learning general sequential decision-making problems, expressed as a function of information structure. In particular, the dependence is on an interpretable quantity derived from a graphical representation that can be thought of as an effective "information-structural state", generalizing the typical notion of a Markovian state. We prove this result by exhibiting an algorithm achieving this upper bound, making use of the generalized predictive state representation constructed earlier which provides a robust representation amenable to learning. In doing so, we identify a larger class of statistically tractable reinforcement learning problems.

**Related Work.**    We provide a detailed discussion of related work in Appendix B.

## 2 Background & Preliminaries

What follows is an abridged description of the relevant background and preliminaries. We refer the reader to Appendix C for a more detailed treatment, and to Appendix A for a summary of notations.

### 2.1 Generic Sequential Decision Making Problems

Consider a controlled stochastic process $(X_1, \ldots, X_H)$, where $X_h$ is a random variable corresponding to the variable at time $h$. At each time $h \in [H]$, the variable $X_h$ may be either an 'observation' (i.e., observable system variable) or an 'action'. The dynamics of this stochastic process are described by a tuple $(H, \{\mathbb{X}_h\}_h, \mathcal{O}, \mathcal{A}, \mathbb{P})$, where $H$ is the time horizon, $\mathbb{X}_h$ is the variable space at time $h$, $\mathcal{O} \subset [H]$ is the index set of observations (i.e., $X_h$ is an observation if $h \in \mathcal{O}$), $\mathcal{A} \subset [H]$ is the index set of actions, and $\mathbb{P} = \{\mathbb{P}_h\}_{h \in \mathcal{O}}$ is a set of probability kernels which describes the probability of any trajectory $x_1, \ldots, x_H$ given the actions, $\mathbb{P}\left[\{x_s : s \in \mathcal{O}\} \mid \{x_s : s \in \mathcal{A}\}\right] = \prod_{h \in \mathcal{O}} \mathbb{P}_h\left[x_h \mid x_1, \ldots, x_{h-1}\right]$. A choice of policy $\pi$ induces a probability distribution $\mathbb{P}^\pi$ on $\mathbb{X}_1 \times \cdots \times \mathbb{X}_H$. The objective of the agent(s) is to choose a policy which maximizes their expected reward $V^R(\pi) = \mathbb{E}^\pi\left[R(X_1, \ldots, X_H)\right]$.

Let $\mathbb{H}_h = \prod_{s \in 1:h} \mathbb{X}_s$ denote the space of histories at time $h$ and $\mathbb{F}_h = \prod_{s \in h+1:H} \mathbb{X}_s$ denote the space futures at time $h$. Similarly, let $\mathbb{H}_h^o = \mathtt{obs}(\mathbb{H}_h) = \prod_{s \in \mathcal{O}_{1:h}} \mathbb{X}_s$ denote the observation component of histories and let $\mathbb{H}_h^a = \mathtt{act}(\mathbb{H}_h) = \prod_{s \in \mathcal{A}_{1:h}} \mathbb{X}_s$ denote the action component. The observation and action components of the futures, $\mathbb{F}_h^o$ and $\mathbb{F}_h^a$ respectively, are defined similarly. Here, $\mathtt{obs}(\cdot)$ and $\mathtt{act}(\cdot)$ extract the observation and action components, respectively, of any trajectory.

We define the *system dynamics matrix* $\boldsymbol{D}_h \in \mathbb{R}^{|\mathbb{H}_h| \times |\mathbb{F}_h|}$ as the matrix giving the probability of each possible pair of history and future at time $h$ given the execution of the actions, $[\boldsymbol{D}_h]_{\tau_h, \omega_h} := \overline{\mathbb{P}}\left[\tau_h, \omega_h\right] \equiv \mathbb{P}\left[\tau_h^o, \omega_h^o \mid \mathrm{do}(\tau_h^a, \omega_h^a)\right]$, where $\tau_h$ is the history, $\omega_h$ is the future, and $\tau_h^o, \tau_h^a, \omega_h^o, \omega_h^a$ separate the history and future into observation components and action components.

The *rank* of the dynamics of a sequential decision-making problem is a measure of its complexity. It is defined as the maximal rank of its dynamics matrices.

**Definition 1** (Rank of dynamics). *The rank of the dynamics $\{\boldsymbol{D}_h\}_{h \in [H]}$ is $r = \max_{h \in [H]} \mathrm{rank}(\boldsymbol{D}_h)$.*

### 2.2 Generalized Predictive State Representations

Predictive state representations (PSR) [31, 32] are a representation of dynamical systems and sequential decision-making problems based on predicting future observations given the past, without explicitly modeling a latent state. In this section, we propose and formalize a generalization of standard PSRs which allows for an arbitrary order of observations and actions. This generalization is necessary to capture POSTs/POSGs (introduced in Section 3), but may also be of independent interest.

The "PSR rank" of a sequential decision-making problem coincides with the rank of its dynamics (Definition 1). Denote $r_h := \mathrm{rank}(\boldsymbol{D}_h)$. At the heart of predictive state representations is the concept of "core test sets." A core test set at time $h$ is a set of futures such that the vector of probabilities of those futures conditioned on the past encodes all the information that the past contains about the future.

**Definition 2** (Core test sets). *A core test set at time $h$ is a subset of $d_h \geq r_h$ futures, $\mathbb{Q}_h := \{q_h^1, \ldots, q_h^{d_h}\} \subset \mathbb{F}_h$, such that the submatrix $\boldsymbol{D}_h[\mathbb{Q}_h] \in \mathbb{R}^{|\mathbb{H}_h| \times d_h}$ is full-rank.*

The $d_h$-dimensional vector $\psi_h(\tau_h) := (\overline{\mathbb{P}}[\tau_h, q_h^1], \ldots, \overline{\mathbb{P}}[\tau_h, q_h^{d_h}])$ serves as a sufficient statistic for computing the probability of any future conditioned on $\tau_h$. Intuitively, core test sets relate the rank of the dynamics to a representation based on predicting future outcomes. A PSR parameterization of the system dynamics represents the probability of any trajectory using the sufficient statistics $\psi_h$ via a set of operators that iteratively update the predictive representations at each step as new observations come in. We proceed to formally define our generalized predictive state representation.

**Definition 3** (Generalized Predictive State Representations). *Consider a sequential decision-making problem $(X_h)_{h \in [H]}$ where $\mathcal{A}, \mathcal{O}$ partition $[H]$ into actions and observations, respectively. Then, a generalized predictive state representation for this sequential decision-making problem is a tuple $(\{\mathbb{Q}_h\}_{0 \leq h \leq H-1}, \phi_H, \{M_h\}_{1 \leq h \leq H-1}, \psi_0)$ satisfying*

$$\overline{\mathbb{P}}\left[x_1, \ldots, x_H\right] = \phi_H(x_H)^\top M_{H-1}(x_{H-1}) \cdots M_1(x_1)\psi_0 \tag{1}$$

$$\psi_h(x_1, \ldots, x_h) = M_h(x_h) \cdots M_1(x_1)\psi_0, \ \forall h, \tag{2}$$

*where* $M_h : \mathbb{X}_h \to \mathbb{R}^{d_h \times d_{h-1}}$, $\phi_H : \mathbb{X}_H \to \mathbb{R}^{d_{H-1}}$, *and* $\psi_0 = \psi(\emptyset)$.

An important condition for the learnability of PSR models, which was used in prior work [including 33–35], is the so-called "well-conditioning assumption". We state the analogous assumption for our generalized PSR model below. For a core test set $\mathbb{Q}_h$, let $\mathbb{Q}_h^A = \texttt{act}(\mathbb{Q}_h)$ be the set of core action sequences, let $Q_A = \max_h |\mathbb{Q}_h^A|$ be the number of core action sequences, and let $d = \max_h d_h$.

**Assumption 1** ($\gamma$-well-conditioned generalized PSR). *A generalized PSR model is said to be $\gamma$-well conditioned for $\gamma > 0$ if, for any $h \in [H]$ and any $\pi$, it satisfies*

$$\max_{\substack{z \in \mathbb{R}^{d_h} \\ \|z\|_1 \leq 1}} \sum_{\omega_h \in \mathbb{F}_h} \pi(\omega_h) \left| m_h(\omega_h)^\top z \right| \leq \frac{1}{\gamma}, \qquad \max_{\substack{z \in \mathbb{R}^{d_h} \\ \|z\|_1 \leq 1}} \sum_{x_h \in \mathbb{X}_h} \|M_h(x_h)z\|_1 \, \pi(x_h) \leq \frac{\left| \mathbb{Q}_{h+1}^A \right|}{\gamma},$$

*where* $m_h(\omega_h)^\top = \phi_H(x_H)^\top M_{H-1}(x_{H-1}) \cdots M_{h+1}(x_{h+1})$ *and* $\forall \omega_h^a, \sum_{\omega_h^o} \pi(\omega_h^o, \omega_h^a) = 1$.

To understand this condition, recall that $m_h(\omega_h)^\top \psi_h(\tau_h) = \overline{\mathbb{P}}[\tau_h, \omega_h]$. Thus, we can interpret $z$ as the error in estimating the prediction features $\psi_h$. This condition ensures that the error in estimating the probability of system trajectories remains controlled when the estimation error of $\psi_h$ is small.

Although we focus on finite spaces $\mathbb{X}_h$ in this work, we briefly discuss possible extensions to continuous spaces. In finite settings, the core tests $q \in \mathbb{Q}_h$ are represented by future trajectories $q = (x_{h+1}, ..., x_H) \in \mathbb{F}_h$. In continuous spaces, the core tests become trajectories over subsets of the underlying continuous space: $q = (\mathcal{X}_{h+1}, \ldots, \mathcal{X}_H) \subset \mathbb{F}_h$, where each $\mathcal{X}_k \subset \mathbb{X}_k$ is a measureable subset. We point to [33] for more discussion on how to extend (standard) PSRs to continuous spaces.

# 3 Information Structure

The "information structure" of a sequential decision-making problem defines the causal dependencies between events in the system occurring at different points time, whether those events are observable by the learning agent or not. In this section, we will introduce a novel reinforcement learning model that explicitly represents information structure. We demonstrate that this enables a rich analysis of the system's dynamics and is crucial for characterizing the statistical complexity of general RL problems.

## 3.1 Partially-Observable Sequential Teams

We propose a model of sequential decision-making problems that includes an explicit representation of information structure, which we call partially-observable sequential teams (POST). A POST is a controlled stochastic process consisting of a sequence of variables, where each variable is either a "system variable" or an "action variable". POSTs also model the *observability* of each system variable with respect to the learning algorithm (i.e., which system variables are available to the learning algorithm). The information structure of a POST describes the causal dependence between these variables. The *information set* of each system variable describes the subset of past variables that are coupled to it in the dynamics. The information set of an action variable describes the information available to the agent when choosing an action, hence defining the policy class they optimize over.

**Definition 4** (POST). *A partially-observable sequential team is a controlled stochastic process that specifies the joint distribution of $T$ variables $(X_t)_{t \in [T]}$, together with a specification of the observability of each variable. Here each $X_t$ is either a system variable or an action variable, and is either observable by the learning agent or not. A POST is specified by the following components.*

1. *__Variable Structures.__ The variables $\{X_t\}_{t \in [T]}$ are partitioned into two disjoint subsets — system variables and action variables. $\mathcal{S} \subset [T]$ indexes system variables and $\mathcal{A} \subset [T]$ indexes action variables, with $\mathcal{S} \cap \mathcal{A} = \emptyset$, $\mathcal{S} \cup \mathcal{A} = [T]$.*
2. *__Information Structure.__ For $t \in [T]$, the "information set" $\mathcal{I}_t \subset [t-1]$ of the variable $X_t$ is the set of past variables that are coupled to $X_t$ in the dynamics. That is, the transition to $X_t$ depends on the value of $I_t := (X_s : s \in \mathcal{I}_t)$. We call $I_t$ the "information variable" at time $t$, and call $\mathbb{I}_t = \prod_{s \in \mathcal{I}_t} \mathbb{X}_s$ the "information space".*
3. *__System Kernels.__ For any $t \in \mathcal{S}$, $\mathcal{T}_t \in \mathcal{P}(\mathbb{X}_t | \mathbb{I}_t)$ is kernel from $\mathbb{I}_t$ to $\mathbb{X}_t$ that specifies the conditional distribution of a system variable $X_t$ given the information variable $I_t$.*
4. *__Decision Kernels.__ Each agent chooses a decision kernel $\pi_t : \mathbb{I}_t \to \mathcal{P}(\mathbb{X}_t)$, specifying a (potentially randomized) policy for choosing an action at time $t \in \mathcal{A}$.*

5. ***Observability.*** *We denote the observable system variables by $\mathcal{O} \subset \mathcal{S}$. We require that the information sets of the action variables are observable to the learning algorithm: $\mathcal{O} \supset \cup_{t \in \mathcal{A}}(\mathcal{I}_t \cap \mathcal{S})$. We define $\mathcal{U} := \mathcal{O} \cup \mathcal{A}$, and let $H := |\mathcal{U}|$ be the time-horizon of the observable variables.*

6. ***Reward Function.*** *At the end of an episode, the team receives a reward determined by the function $R : \prod_{s \in \mathcal{U}} \mathbb{X}_s \to [0, 1]$.*

With the above components, any set of decision kernels (i.e., joint policy) $\boldsymbol{\pi} = (\pi_t : t \in \mathcal{A})$ induces a unique probability measure over $\mathbb{X}_1 \times \cdots \times \mathbb{X}_T$, which is given by

$$\mathbb{P}^{\boldsymbol{\pi}}[x_1, \ldots, x_T] = \prod_{t \in \mathcal{S}} \mathcal{T}_t(x_t | \{x_s : s \in \mathcal{I}_t\}) \prod_{t \in \mathcal{A}} \pi_t(x_t | \{x_s : s \in \mathcal{I}_t\}). \tag{3}$$

We will be interested in modeling the *observable* dynamics of the POST. We index the observable variables by their position among the observables $h \in [H]$ rather than their position among all variables, as follows: $(X_t)_{t \in \mathcal{U}} = (X_{t(1)}, \ldots, X_{t(H)})$, where $t : [H] \to \mathcal{U} \subset [T]$ maps the index over observables to the index over all variables. The distribution of the observables is obtained by marginalizing over the unobservable variables, $\mathbb{P}^{\boldsymbol{\pi}}[x_{t(1)}, \ldots, x_{t(H)}] = \sum_{s \in \mathcal{O}^{\complement}} \sum_{x_s \in \mathbb{X}_s} \mathbb{P}^{\boldsymbol{\pi}}[X_1 = x_1, \ldots X_T = x_T]$.

The value of a policy is given by its expected reward $V(\boldsymbol{\pi}) := \mathbb{E}^{\boldsymbol{\pi}}[R(X_{t(1)}, \ldots, X_{t(H)})]$, where $\mathbb{E}^{\pi}$ is the expectation associated with the probability measure $\mathbb{P}^{\boldsymbol{\pi}}$. The objective of a POST is to learn a policy $\boldsymbol{\pi} = (\pi_t)_{t \in \mathcal{A}}$ which maximizes the expected reward, $\sup_{\boldsymbol{\pi}} V(\boldsymbol{\pi})$. When the variable spaces $\mathbb{X}_t$ are finite, this supremum is attained by a deterministic policy, $\pi_t : \mathbb{I}_t \to \mathbb{X}_t, t \in \mathcal{A}$.

**Representation of the information structure as a directed acyclic graph.** The information structure of a POST can be naturally represented as a labeled directed acyclic graph (DAG). Given the variable structure and information structure of a POST, $(\mathcal{S}, \mathcal{A}, \{\mathcal{I}_t\}_t)$, its DAG representation is given by $\mathcal{G}(\mathcal{V}, \mathcal{E}, \mathcal{L})$. The nodes of the graph are the set of variables, $\mathcal{V} = [T] = \mathcal{S} \cup \mathcal{A}$. The edges $\mathcal{E} \subset \mathcal{V} \times \mathcal{V}$ of the DAG are given by $\mathcal{E} = \{(i, t) : t \in [T], i \in \mathcal{I}_t\}$. That is, there exists an edge from $i$ to $t$ if $i$ is in the information set of $t$. Finally, $\mathcal{L}$ contains labels for each node as being a system variable (in $\mathcal{S}$) or an action variable (in $\mathcal{A}$). Further, the observability of system variables (in $\mathcal{O}$) is also labeled.

This DAG represents a directed *graphical model* for the POST. In particular, the probability distribution on $\mathbb{X}_1 \times \cdots \times \mathbb{X}_T$ factors according to $\mathcal{G}$,

$$\mathbb{P}[X_1, \ldots, X_T] = \prod_{t \in \mathcal{V}} \mathbb{P}[X_t \mid \mathrm{pa}(X_t)], \tag{4}$$

where $\mathrm{pa}(X_t)$ is the set of parents of $X_t$ in $\mathcal{G}$ (which are $\mathcal{I}_t$). This representation of the information structure as a DAG will be crucial for our analysis of the dynamics of sequential teams in Section 3.2.

**Partially observable sequential *games*.** We define partially-observable sequential games (POSGs) in a similar manner. The main difference is that, in the game setting, there exists an expanded reward structure with $N$ different reward objectives $R^1, \ldots, R^N$, with different agents pursuing different objectives. The action index set is partitioned into $N$ subsets $\mathcal{A} = \mathcal{A}^1 \cup \cdots \mathcal{A}^N$, where $\mathcal{A}^i \subset \mathcal{A}$ denotes the action index set associated with the agent(s) optimizing for objective $R^i$. The extension to the game setting is treated in detail in Appendix F.

### 3.2 Information Structure Determines the Rank of POSTs/POSGs

POSTs and POSGs form a highly general framework that captures any sequential decision-making problem that can be described by a controlled stochastic process, subsuming classical models such as MDPs, POMDPs, Dec-POMDPs, and POMGs. By introducing a model with an explicit representation of information structure, we gain the ability to perform a richer analysis of sequential decision-making problems. In particular, we will show that the rank of the dynamics, and ultimately the statistical complexity of reinforcement learning, can be characterized as a function of the information structure.

We begin by defining the central quantity in our analysis, which we call the "information-structural state", hinting at the role it will play. The information structural state is defined for each point in time as a subset of the past (whether observed or latent) which forms a sufficient statistic for the future.

**Definition 5** (Information-structural state)**.** *Let $\mathcal{G}^{\dagger}$ be the DAG obtained from $\mathcal{G}$ by removing all edges directed towards actions. That is, it consists of the edges $\mathcal{E}^{\dagger} := \mathcal{E} \setminus \{(x, a) : x \in \mathcal{N}, a \in \mathcal{A}\}$. For each $h \in [H]$, let $\mathcal{I}_h^{\dagger} \subset [t(h)]$ be the minimal set of past variables (observed or unobserved) which d-separates the past observations $(X_{t(1)}, \ldots, X_{t(h)})$ from the future observations $(X_{t(h+1)}, \ldots, X_{t(H)})$ in the DAG $\mathcal{G}^{\dagger}$. Define $\mathbb{I}_h^{\dagger} := \prod_{s \in \mathcal{I}_h^{\dagger}} \mathbb{X}_s$ as the joint space of those variables.*

The following theorem, whose proof is given in Appendix H, states that the rank of the observable system dynamics of POSTs and POSGs is bounded by the cardinality of $\mathbb{I}_h^\dagger$.

**Theorem 1.** *The rank of the observable system dynamics of a POST or POSG is bounded in terms of its information structure by $r \leq \max_{h \in [H]} |\mathbb{I}_h^\dagger|$.*

This result shows that the complexity of the observable system dynamics is characterized by the information structure through $\mathbb{I}_h^\dagger$. In particular, $i_h^\dagger \in \mathbb{I}_h^\dagger$ describes a set of system variables, either observable or latent, which provide a sufficient statistic of the past at time $h$ for predicting future observations—$I_h^\dagger$ "separates" the past from the future. Hence, the quantity $|\mathbb{I}_h^\dagger|$ admits an interpretation as the size an effective state space at time $h$. This is a generalization of the standard notion of a latent state. For example, in the case of POMDPs or Dec-POMDPs, the information-structural state indeed coincides with the latent Markovian state (Figure 2a). We emphasize that $\mathbb{I}_h^\dagger$ may contain observable variables as well as *unobservable* system variables. In fact, unobservable system variables can introduce crucial structure that simplifies the observable system dynamics.

### 3.3 Examples of Information Structures and their Rank

In this section, to provide some intuition, we present examples of information structures and apply Theorem 1 to obtain a bound the rank of their observable system dynamics. We see that classical models such as MDPs, POMDPs, and POMGs are special cases of the POST/POSG framework, and known results about their rank [31] are recovered by the generalized graph-theoretic analysis of their information structure. For notational convenience, we adopt a modified notation in this section where information sets $\mathcal{I}$ are indexed by the symbol of the variable rather than its time-index. For example, in an MDP, we write $\mathcal{I}(s_t) = \{s_{t-1}, a_{t-1}\}$ for the information set of the state variable $s_t$.

**Decentralized POMDPs and POMGs.** At each time $t$, the system variables of a decentralized POMDP (or POMG) consists of a latent state $s_t$, observations for each agent $o_t^1, \ldots, o_t^N$, and actions of each agent $a_t^1, \ldots, a_t^N$. The latent state transitions are Markovian and depend on the agents' joint action. The observations are sampled via a kernel conditioned on the latent state. Each agent can use their own history of observations to choose an action. Thus, the information structure is given by

$$\mathcal{I}(s_t) = \{s_{t-1}, a_{t-1}^1, \ldots, a_{t-1}^N\}, \, \mathcal{I}(o_t^i) = \{s_t\}, \, \mathcal{I}(a_t^i) = \{o_{1:t-1}^i, a_{1:t-1}^i\}.$$

Here, the observable variables are $\mathcal{U} = \{o_{1:T}^i, a_{1:T}^i, i \in [N]\}$. By Theorem 1, we have $\mathcal{I}^\dagger(o_t^i) = \mathcal{I}^\dagger(a_t^i) = \{s_t\}$, $\forall t, i$, as shown in Figure 2a. Thus, the rank of a Dec-POMDP is bounded by $|\mathbb{S}|$, where $\mathbb{S}$ is the state space. Note that in the case of models with a true latent state (e.g., POMDPs, Dec-POMDPs, and POMGs), the information-structural state coincides with the Markovian latent state.

**Point-to-Point Real-Time Communication with Feedback.** Consider the following model of real-time communication with feedback. Let $x_t$ be a Markov source. At time $t$, the encoder receives the source $x_t \in \mathbb{X}$ and sends an encoded symbol $z_t \in \mathbb{Z}$. The symbol is sent through a memoryless noisy channel which outputs $y_t$ to the receiver. The decoder produces the estimate $\widehat{x}_t$. The output of the noisy channel is also fed back to the encoder. The encoder and decoder have full memory of their observations and previous "actions". The observation variables are $\mathcal{O} = \{x_{1:T}, y_{1:T}\}$ and the "actions" are $\mathcal{A} = \{z_{1:T}, \widehat{x}_{1:T}\}$. Hence, the information structure is given by the following,

$$\mathcal{I}(x_t) = \{x_{t-1}\}, \, \mathcal{I}(z_t) = \{x_{1:t}, y_{1:t-1}, z_{1:t-1}\}, \, \mathcal{I}(y_t) = \{z_t\}, \, \mathcal{I}(\widehat{x}_t) = \{y_{1:t}\}.$$

By Theorem 1, we have that,

$$\mathcal{I}^\dagger(x_t) = \{x_t\}, \, \mathcal{I}^\dagger(z_t) = \{x_t\}, \, \mathcal{I}^\dagger(y_t) = \{x_t, z_t\}, \, \mathcal{I}^\dagger(\widehat{x}_t) = \{x_t\}.$$

Hence, the rank is bounded by $|\mathbb{X}||\mathbb{Z}|$. This is depicted in Figure 2c.

**Limited-memory information structures.** Consider a sequential decision making problem with variables $o_t, a_t, t \in [T]$ and an information structure with length-$m$ "memory". That is, observations do not directly depend on variables more than $m$ steps in the past. That is, the information structure is

$$\mathcal{I}(o_t) = \{o_{t-m:t-1}, a_{t-m:t-1}\}, \mathcal{I}(a_t) = \{o_{1:t}, a_{1:t-1}\}.$$

The observables consist of all observations and actions, $\mathcal{U} = \{o_{1:T}, a_{1:T}\}$. By Theorem 1 we have that $\mathcal{I}^\dagger(o_t) = \{o_{t-m:t-1}, a_{t-m:t-1}\}$, as shown in Figure 2d. Hence, the rank of this sequential decision-making problem is bounded by $|\mathbb{O}|^m |\mathbb{A}|^m$.

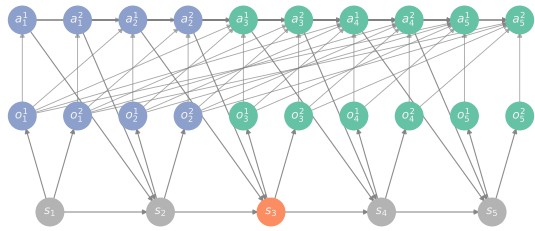

(a) Decentralized POMDP/POMG information-structure.

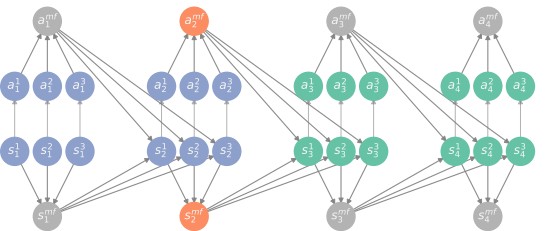

(b) "Mean-field" information structure.



(c) Point-to-point real-time communication with feedback information structure.

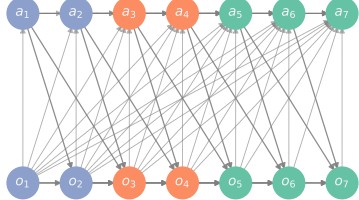 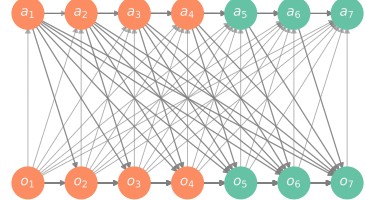

(d) Limited-memory ($m = 2$) information structures.  (e) Fully connected information structure.

Figure 2: DAG representation of various information structures. Solid edges indicate the edges in $\mathcal{E}^{\dagger}$ and light edges indicate the information sets of action variables. Grey nodes represent unobservable variables, blue nodes represent past observable variables, green nodes represent future observable variables, and red nodes represent $\mathcal{I}_h^{\dagger}$. To find $\mathcal{I}_h^{\dagger}$, as per Theorem 1, we first remove the incoming edges into the action variables, then we find the minimal set among all past variables (both observable and unobservable) which $d$-separates the past observations from the future observations.

The examples above show that the complexity of the dynamics of a sequential decision-making problem depends directly on its information structure. An expanded version of this discussion is provided in Appendix D. Next, we use an information-structural analysis to construct a generalized PSR representation for a class of POSTs/POSGs, which we will ultimately use to prove an upper bound on the *statistical complexity* of reinforcement learning as a function of information structure.

## 4 Constructing a PSR Parameterization for POSTs and POSGs

A key challenge in reinforcement learning is constructing representations which enable robustly and efficiently modeling probabilities of system trajectories (i.e., probabilities of the form $\mathbb{P}[\texttt{future} \mid \texttt{history}]$). In this section, we will construct a generalized predictive state representation for a class of POSTs/POSGs, ultimately enabling sample-efficient reinforcement learning.

### 4.1 Core test sets for POSTs/POSGs

Recall that a core test set is a set of futures such that the probabilities of those futures given the past encode all the information that the past contains about the future. For systems with a simple and

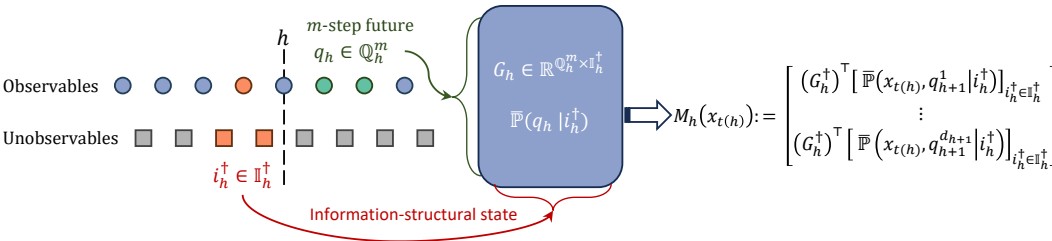

Figure 3: A depiction of the construction of a generalized generalized predictive state representation for POST/POSG models.

regular information structure, a core test set may be simple to obtain. For example, undercomplete POMDPs with a full-rank emission matrix admit the 1-step observation futures as core test sets.

For POSTs/POSGs with arbitrary information structures, obtaining a core test set is much more challenging without knowing the system dynamics. In this section, we identify a condition in terms of the information structure under which $m$-step futures are a core test set for POSTs/POSGs. Let us denote the candidate core test set of $m$-step futures at time $h$ by $\mathbb{Q}_h^m := \prod_{s \in \mathcal{U}_{h+1:\min(h+m,H)}} \mathbb{X}_s$. We define the matrix $\boldsymbol{G}_h$ as encoding the probability of observing each $m$-step future conditioned on the information-structural state $i_h^\dagger \in \mathbb{I}_h^\dagger$:

$$\boldsymbol{G}_h := \left[\overline{\mathbb{P}}\big[q \mid i_h^\dagger\big]\right]_{q, i_h^\dagger} \in \mathbb{R}^{|\mathbb{Q}_h^m| \times |\mathbb{I}_h^\dagger|}, \ q \in \mathbb{Q}_h^m, \ i_h^\dagger \in \mathbb{I}_h^\dagger. \tag{5}$$

The operational meaning of $\boldsymbol{G}_h$ is depicted in Figure 3. Next, we formulate a condition in terms of information structure that we will show implies that the $m$-step futures are core test sets.

**Definition 6** ($m$-step $\mathcal{I}^\dagger$-weakly revealing). *We say that a sequential team is $m$-step $\mathcal{I}^\dagger$-weakly revealing if for all $h \in [H]$, $\mathrm{rank}(\boldsymbol{G}_h) = |\mathbb{I}_h^\dagger|$. Furthermore, we say that the sequential team is $\alpha$-robustly $m$-step $\mathcal{I}^\dagger$-weakly revealing if for all $h \in [H - m + 1]$, $\sigma_{|\mathbb{I}_h^\dagger|}(\boldsymbol{G}_h) \geq \alpha$.*

The $\mathcal{I}^\dagger$-weakly revealing condition is essentially a statistical identifiability condition. If a POST/POSG is $\mathcal{I}^\dagger$-weakly revealing, then, for any two mixtures of the information-structural state, the distributions of the $m$-step futures are distinct. Formally, for any $\nu_1, \nu_2 \in \mathcal{P}(\mathbb{I}_h^\dagger)$ with $\mathrm{supp}(\nu_1) \cap \mathrm{supp}(\nu_2) = \emptyset$, we have $\boldsymbol{G}_h \nu_1 \neq \boldsymbol{G}_h \nu_2$. That is, the future observations contain information that can distinguish between mixtures of the latent information-structural state. The $\alpha$-robust version of the $\mathcal{I}^\dagger$-weakly revealing condition requires that $\boldsymbol{G}_h$ is not only full rank, but that its $|\mathbb{I}_h^\dagger|$-th singular value is bounded away from zero, so that $\|G_h \nu_1 - G_h \nu_2\| \geq \alpha \|\nu_1 - \nu_2\|$.

The condition holds whenever there exists a sequence of actions within the $m$-step futures such that executing these actions results in a sequence of observations that is informative about the information-structural state $i_h^\dagger \in \mathbb{I}_h^\dagger$. In general, this condition will be harder to satisfy when $\mathbb{I}_h^\dagger$ is large since it would require the $m$-step future observations to encode more information. In particular, $\boldsymbol{G}_h$ cannot be full rank when $|\mathbb{Q}_h^m| < |\mathbb{I}_h^\dagger|$. As a heuristic, when we don't have prior knowledge about the dynamics (e.g., in the learning setting), we can choose $m$ such that $|\mathbb{Q}_h^m| \geq |\mathbb{I}_h^\dagger|$. In general, it will be possible to find a smaller core test set when the $d$-separating set $\mathcal{I}_h^\dagger$ is small. This happens when the system dynamics contain state-like variables that are low-dimensional.

The $\mathcal{I}^\dagger$-weakly-revealing condition is a generalization of the "weakly-revealing" condition for POMDPs introduced by Liu et al. [36]. Liu et al. [37] proposed an algorithm for learning weakly-revealing POMGs. Our analysis here recovers weakly-revealing POMGs as a special case and enables learning a much more general class of problems. We note that such an identifiability condition is necessary, and is reflective of the fundamental difficulty of the partially-observable setting. For example, in the case of POMDPs, there exist hardness results that state that if an analogous condition does not hold, the statistical complexity can scale exponentially with the relevant quantities [36].

We now proceed to show that under the $\mathcal{I}^\dagger$-weakly-revealing condition, $m$-step futures are core test sets for POSTs/POSGs which can be used to construct a generalized PSR representation amenable

to learning. Recall that the vector of core test set probabilities for the history $\tau_h$ is given by the mappings $\psi_h, \overline{\psi}_h : \mathbb{H}_h \to \mathbb{R}^{|\mathbb{Q}_h^m|}$,

$$\psi_h(\tau_h) = \left[\mathbb{P}\left[q^o, \tau_h^o \mid \operatorname{do}(\tau_h^a), \operatorname{do}(q^a)\right]\right]_{q \in \mathbb{Q}_h^m}, \quad \overline{\psi}_h(\tau_h) = \left[\mathbb{P}\left[q^o \mid \tau_h^o; \operatorname{do}(\tau_h^a), \operatorname{do}(q^a)\right]\right]_{q \in \mathbb{Q}_h^m}.$$

Define the mapping $m_h : \mathbb{F}_h \to \mathbb{R}^{|\mathbb{Q}_h^m|}$ as,

$$m_h(\omega_h) := (\boldsymbol{G}_h^\dagger)^\top \left[\overline{\mathbb{P}}[\omega_h \mid i_h^\dagger]\right]_{i_h^\dagger \in \mathbb{I}_h^\dagger}. \tag{6}$$

The following lemma, whose proof is given in Appendix I, shows that the $m$-step futures $\mathbb{Q}_h^m$ are core test sets for any $m$-step $\mathcal{I}^\dagger$-weakly revealing POST/POSG. In particular, given any future $\omega_h \in \mathbb{F}_h$ and history $\tau_h \in \mathbb{H}_h$, the conditional probability $\overline{\mathbb{P}}[\omega_h \mid \tau_h]$ can be written as a linear combination of the core test probabilities $\overline{\psi}_h(\tau_h)$, with weights given by $m_h(\omega_h)$, depending only on the future $\omega_h$.

**Lemma 1** (Core test set for POSTs)**.** *Suppose that the POST is $m$-step $\mathcal{I}^\dagger$-weakly revealing. Then, $\mathbb{Q}_h^m$ is a core test set for all $h \in [H]$. Furthermore, we have $\overline{\mathbb{P}}[\tau_h, \omega_h] = \langle m_h(\omega_h), \psi_h(\tau_h)\rangle$ and $\overline{\mathbb{P}}[\omega_h \mid \tau_h] = \langle m_h(\omega_h), \overline{\psi}_h(\tau_h)\rangle$.*

### 4.2 Generalized PSR parameterization of POST/POSG

Consider a POST/POSG which is $m$-step $\mathcal{I}^\dagger$-weakly revealing. Lemma 1 shows that the $m$-step futures $\mathbb{Q}_h^m$ are core test sets. In this section, we will explicitly construct a generalized PSR parameterization for this class of sequential decision-making problems. Moreover, we will show that this generalized PSR representation is well-conditioned when the weakly revealing condition is robust.

Let $d_h := |\mathbb{Q}_h^m|$. The key observation that allows us to construct the generalized PSR representation is that the vector mappings $m_h : \mathbb{F}_h \to \mathbb{R}^{d_h}$ and $\psi_h : \mathbb{H}_h \to \mathbb{R}^{d_h}$ can be used to derive a recursive form of the dynamics of the POST/POSG. We define the operator mapping $M_h : \mathbb{X}_{t(h)} \to \mathbb{R}^{d_h \times d_{h-1}}$ by

$$\left[M_h(x_{t(h)})\right]_{q,\cdot} = m_{h-1}(x_{t(h)}, q)^\top, \; q \in \mathbb{Q}_h. \tag{7}$$

That is, $M_h(x_{t(h)})$ is the matrix whose rows are indexed by the core tests at the $h$-th observable step, where the row corresponding to each $q \in \mathbb{Q}_h$ is the weights returned by the mapping $m_{h-1}$ when applied to the future consisting of $x_{t(h)}$ followed by core test $q$. The operator map $M_h$ allows us to update the probabilities of the core test sets after receiving an additional observation $x_{t(h)}$: $\psi_h(x_{t(1)}, ..., x_{t(h)}) = M_h(x_{t(h)})\psi_{h-1}(x_{t(1)}, ..., x_{t(h-1)})$. The following result, whose proof is given in Appendix I, states that the set of operators $\{M_h\}_h$ forms a generalized PSR.

**Theorem 2.** *Consider an $m$-step $\mathcal{I}^\dagger$-weakly revealing POST/POSG. Let $\{M_h\}_{h \in [H-1]}$ be defined as above, and let $\psi_0 = \left[\overline{\mathbb{P}}[q]\right]_{q \in \mathbb{Q}_0^m}$, $\phi_H(x_{t(H)}) = \boldsymbol{e}_{x_{t(H)}}$. Then, $(\{\mathbb{Q}_h^m\}_h, \phi_H, \{M_h\}_{h \in [H-1]}, \psi_0)$ forms a generalized predictive state representation. Moreover, if the weakly-revealing property is $\alpha$-robust, then this PSR is $\gamma$-well-conditioned with $\gamma = \alpha / \max_h |\mathbb{I}_h^\dagger|^{1/2}$.*

Thus, we have constructed a robust parameterization of the sequential decision-making problem, making use of its information structure, that will enable us to design an efficient learning algorithm.

## 5 Characterizing the Statistical Complexity of General Reinforcement Problems via Information Structure

In this section, we establish an upper bound on the achievable sample complexity of general reinforcement learning problems in terms of their information structure. In plain language, this is a result that roughly says "any sequential decision-making problem with an information structure $\mathcal{I}$ can be learned with a sample complexity at most $f(\mathcal{I})$". This identifies a class of sequential decision-making problems that are statistically tractable via conditions on the information structure, expanding the set of known-tractable problems while recovering existing tractability results as a special case.

We will prove this result by exhibiting an algorithm that achieves this upper bound. Our approach will be to use the generalized predictive state representation constructed for POSTs/POSGs in Section 4, which provides a robust representation amenable to learning. We will introduce an algorithm for

learning generalized PSRs and prove a corresponding sample complexity result, which will in turn imply a bound on the sample complexity of learning POST/POSG models via the information-structural characterization of the rank of observable dynamics established in Section 3.2.

There exist several works in the literature which study learning in PSRs [e.g. 33–35, 38, 39]. Using the technical tools developed in this paper, most of these algorithms can be directly extended to our generalization of PSRs. With such an algorithm, Theorems 1 and 2 then imply a bound on the achievable sample complexity for learning general sequential decision-making problems in terms of their information structure. In this work, we will adapt the model-based UCB-type algorithm of Huang et al. [35], extending it to generalized PSRs to obtain a bound on the achievable sample complexity for POSTs/POSGs. We will defer the details of the algorithm to Appendix E and formally verify the proof in Appendix J. Here, we will focus on discussing the role of information structure in determining the statistical complexity of reinforcement learning.

The following result states that the size of the information-structural state, $|\mathbb{I}_h^\dagger|$, characterizes an upper bound on the statistical complexity of learning a sequential decision-making problem.

**Theorem 3.** *Suppose a sequential decision-making problem described by a POST is $\alpha$-robustly $m$-step $\mathcal{I}^\dagger$-weakly revealing. Let $Q_m := \max_h |\mathbb{Q}_h^m|$ be the size of the $m$-step observable trajectories, and let $A = \max_{s \in \mathcal{A}} |\mathbb{X}_s|$ be the size of largest action space. Then, there exists an algorithm that can learn an $\epsilon$-optimal policy with a sample complexity (omitting log factors) bounded by*

$$\frac{1}{\epsilon^2} \times \mathrm{poly}\left(\frac{1}{\alpha}, \max_h \left|\mathbb{I}_h^\dagger\right|, Q_m, A, H\right).$$

*Under the game setting, the same assumption implies the existence of a self-play algorithm that learns an $\epsilon$ (Nash or coarse-correlated) equilibrium with the same sample complexity.*

This result identifies $\mathcal{I}^\dagger$-weakly revealing POSTs/POSGs as a class of statistically tractable sequential decision-making problems. The sample complexity is polynomial in the size of the information-structural state space $\max_h |\mathbb{I}_h^\dagger|$, the size of the action space $A$, and the time horizon $H$. Further, it depends on $Q_m$, the size of $m$-step observable trajectories, and the robustness parameter $\alpha^{-1}$, which corresponds to the $m$-step $\mathcal{I}^\dagger$-weakly revealing identifiability condition. We note that the algorithm constructed to prove Theorem 3 only needs to know the parameters of the $\mathcal{I}^\dagger$-weakly revealing condition (i.e., $m$ and $\alpha$), and does not need to know the full information structure.

This result shows that the size of the information-structural state is a fundamental measure of the statistical complexity of reinforcement learning. As a result, learning is tractable when $\max_h |\mathbb{I}_h^\dagger|$ is of modest size, and the information structural state is strongly coupled to the observable system variables.

One notable special case of POSTs/POSGs is POMDPs. Learning in POMDPs has been studied extensively in the literature. Theorem 3 implies a $\mathrm{poly}(S, O, A, H, \alpha^{-1}) \cdot \epsilon^{-2}$ bound on the sample complexity of learning in $\alpha$-weakly POMDPs and $\mathrm{poly}(S, (OA)^m, H, \alpha^{-1}) \cdot \epsilon^{-2}$ for learning $m$-step weakly revealing POMDPs. This recovers a similar sample complexity as was shown by more specialized analysis tailored to POMDPs [e.g., 36, 40].

## 6 Conclusion

**Summary.** This paper examines the role of information structure in general reinforcement learning problems. We introduced novel models that explicitly represent information structure, and proved an upper bound on the sample complexity of general reinforcement learning problems in terms of information structure. The central quantity in this upper bound is derived from a graphical representation of the information structure, and admits an interpretation as an effective information-structural state, generalizing the typical notion of a Markovian state.

**Limitations.** The results of this paper are theoretical in nature, and the algorithm proposed to prove our main result is not computationally practical. Current SOTA algorithms for partially-observable algorithms are based on recurrent neural networks or other deep learning models, which create internal representations of latent states based on the history, akin to belief states. Our theory may offer insights into the design of improved architectures in such deep learning-based algorithms. For example, the information structure can be incorporated as an inductive bias of the neural network.

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

## A    Summary of Notation

**Generic Sequential Decision-Making Problems**

| | |
|---|---|
| $\mathbb{X}_t$ | Space that the variable $X_t$ lies in within the stochastic process $(X_1, \ldots, X_H)$. |
| $\mathcal{O}$ | $\mathcal{O} \subset [H]$ denotes the set of observations among the variables $(X_1, \ldots, X_H)$. |
| $\mathcal{A}$ | $\mathcal{A} \subset [H]$ denotes the set of actions among the variables $(X_1, \ldots, X_H)$. |
| $\mathbb{H}_h$ | The space of histories at time $h$. $\mathbb{H}_h := \prod_{s=1}^{h} \mathbb{X}_s$. |
| $\mathbb{F}_h$ | The space of futures at time $h$. $\mathbb{F}_h := \prod_{s=h+1}^{H} \mathbb{X}_s$. |
| $\mathrm{obs}(\cdot)$ | The observation component of a trajectory. $\mathrm{obs}(x_i, \ldots, x_j) := (x_s : s \in \mathcal{O}_{i:j})$. |
| $\mathrm{act}(\cdot)$ | The action component of a trajectory. $\mathrm{act}(x_i, \ldots, x_j) := (x_s : s \in \mathcal{A}_{i:j})$. |
| $\mathbb{H}_h^{\{o,a\}}$ | The space of observation (resp., action) histories. E.g., $\mathbb{H}_h^o := \prod_{s \in \mathcal{O}_{1:h}} \mathbb{X}_s = \mathrm{obs}(\mathbb{H}_h)$. |
| $\mathbb{F}_h^{\{o,a\}}$ | The space of observation (resp., action) histories. E.g., $\mathbb{F}_h^o := \prod_{s \in \mathcal{O}_{h+1:H}} \mathbb{X}_s = \mathrm{obs}(\mathbb{F}_h)$. |
| $\overline{\mathbb{P}}[\tau_h]$ | The probability of a trajectory given actions are executed. $\overline{\mathbb{P}}[\tau_h] := \mathbb{P}[\mathrm{obs}(\tau_h) \mid \mathrm{do}(\mathrm{act}(\tau_h))]$. |
| $\boldsymbol{D}_h$ | Dynamics matrix at time $h$. $\boldsymbol{D}_h \in \mathbb{R}^{|\mathbb{H}_h| \times |\mathbb{F}_h|}$, $[\boldsymbol{D}_h]_{\tau_h, \omega_h} := \overline{\mathbb{P}}[\tau_h, \omega_h]$. $r_h := \mathrm{rank}(\boldsymbol{D}_h)$. |
| $\pi(\tau_h)$ | For $\tau_h = (x_1, \ldots, x_h)$ and policy $\pi$, $\pi(\tau_h) := \prod_{s \in \mathcal{A}_{1:h}} \pi(x_s \mid x_1, \ldots, x_{s-1})$. |
| $\pi(\omega_h \mid \tau_h)$ | For $\tau_h = (x_1, \ldots, x_h)$, $\omega_h = (x_{h+1}, \ldots, x_{h'})$, $\pi(\omega_h \mid \tau_h) = \prod_{s \in \mathcal{A}_{h+1:h'}} \pi(x_s \mid x_1, \ldots, x_{s-1})$. |

**POSTs and POSGs**

| | |
|---|---|
| $\mathbb{X}_t$ | Space that the variable $X_t$ lies in within the stochastic process $(X_1, \ldots, X_T)$. |
| $\mathcal{S}$ | $\mathcal{S} \subset [T]$ denotes the set of system variables among the variables $(X_1, \ldots, X_T)$. |
| $\mathcal{A}$ | $\mathcal{A} \subset [T]$ denotes the set of action variables among the variables $(X_1, \ldots, X_T)$. |
| $\mathcal{O}$ | $\mathcal{O} \subset \mathcal{S}$ denotes the subset of system variables which are observable. |
| $\mathcal{U}$ | The union of observable system variables and action variables. $\mathcal{U} := \mathcal{O} \cup \mathcal{A}$. Let $H := |\mathcal{U}|$. |
| $t(h)$ | For $h \in [H]$ indexing the order among observables, $t(h) \in \mathcal{U}$ denotes the order among all variables. |
| $\mathcal{I}_t$ | The information set of the $t$-th variable. |
| $\mathbb{I}_t$ | $\mathbb{I}_t := \prod_{s \in \mathcal{I}_t} \mathbb{X}_s$ denotes the information space at time $t$. |
| $\mathcal{I}_h^\dagger$ | The minimal $d$-separating set at the $h$-th observable. See Definition 5. |
| $\mathbb{I}_h^\dagger$ | $\mathbb{I}_h^\dagger := \prod_{s \in \mathcal{I}_h^\dagger} \mathbb{X}_s$ denotes the "information-structural state". |

**Generalized PSRs**

| | |
|---|---|
| $\mathbb{Q}_h$ | Core test set at time $h$. Let $d_h := |\mathbb{Q}_h|$ and $d = \max_h d_h$. |
| $\mathbb{Q}_h^A$ | Action component of core test set at time $h$. $\mathbb{Q}_h^A = \mathrm{act}(\mathbb{Q}_h)$. |
| $Q_A$ | Maximum size of the action component of core test sets. $Q_A := \max_h |\mathbb{Q}_h^A|$. |
| $M_h$ | Observable operators of PSR representation mapping $M_h : \mathbb{X}_h \to \mathbb{R}^{d_h \times d_{h-1}}$. |
| $\psi_h$ | Prediction features. $\psi_h(\tau_h) := \left(\overline{\mathbb{P}}[\tau_h, q]\right)_{q \in \mathbb{Q}_h}$. In PSR, $\psi_h(x_1, \ldots, x_h) = M_h(x_h) \cdots M_1(x_1) \psi_0$. |
| $m_h$ | Prediction coefficients. $m_h(\omega_h)^\top := \phi_H(x_H)^\top M_{H-1}(x_{H-1}) \cdots M_{h+1}(x_{h+1})$. |
| $\psi_h$ | Normalized prediction features. $\overline{\psi}_h(\tau_h) := \psi_h(\tau_h)/\overline{\mathbb{P}}[\tau_h] = \left(\mathbb{P}[q \mid \tau_h]\right)_{q \in \mathbb{Q}_h}$. |

**General Mathematical Notation**

| | |
|---|---|
| $\mathsf{D}_{\mathrm{TV}}(p, q)$ | Total variation distance. $\mathsf{D}_{\mathrm{TV}}(p, q) := \sum_{x \in \mathbb{X}} |p(x) - q(x)|$. |
| $\mathsf{D}_{\mathrm{H}}^2(p, q)$ | Hellinger squared distance. $\mathsf{D}_{\mathrm{H}}^2(p, q) := \frac{1}{2} \sum_{x \in \mathbb{X}} (\sqrt{p(x)} - \sqrt{q(x)})^2$. |
| $\sigma_k(A)$ | $k$-th largest singular value of the matrix $A$. |
| $\|A\|_p$ | The matrix $p$-norm. $\|A\|_p := \max_{\|x\|_p = 1} \|Ax\|_p$. |
| $\|x\|_A$ | The vector norm induced by the positive semi-definite matrix $A$. $\|x\|_A := \sqrt{x^\top A x}$. |
| $A^\dagger$ | Moore-Penrose pseudoinverse. |
| $\mathcal{P}(\mathbb{A})$ | Space of probability distributions over $\mathbb{A}$. $\mathcal{P}(\mathbb{B}|\mathbb{A})$ is the space of kernels from $\mathbb{A}$ to $\mathbb{B}$. |
| $\mathcal{N}_{i:j}$ | For an index set $\mathcal{N} \subset [H]$, $\mathcal{N}_{i:j} = \mathcal{N} \cap \{i, \ldots, j\}$. |

# B Related Work

**The study of information structure in the control literature.** In stochastic control, the construct of the "information structure" is used to model the structural properties of a system which may restrict the flow, storage, and processing of information. The role of information structure in decentralized control has been extensively studied since Witsenhausen [15] and Ho and Chu [3] began investigating information structures in the context of team decision theory. For example, early work showed that the information structure can determine the tractability of optimal decentralized control problems [15, 41]. More generally, information structure plays an important role in the analysis of multi-agent team decision problems and games, as well as in the design of efficient algorithms, especially in the decentralized setting. See, e.g., Mahajan et al. [11], Nayyar et al. [42–44], Ouyang et al. [45], Dave et al. [46], and Guan et al. [47] and the references therein. We also refer the reader to Yuksel and Basar [48] for a comprehensive overview of the interaction between information and control, including recent progress in the field. The models we propose in this paper are closely related to Witsenhausen's intrinsic model [1], but with some added elements to model partial-observability in the context of reinforcement learning. Related to this is the framework of multi-agent influence diagrams [e.g., 49, 50] which allow for an explicit representation of dependence relations among variables in games. Whereas the above-mentioned work studies the role of information structure in planning and control, we study the the role of information structure in *reinforcement learning* (i.e., statistical estimation and sample complexity). We note that specific types of structural assumptions related to information structure have been studied in the learning setting as well. For example, previous work has studied factored MDPs [51], which assume an information structure where the state transitions and the reward function are factored, enabling improved learning results. In the present work, we study the role of information structure in reinforcement learning in greater generality, focusing particularly on the partially-observable setting.

**Learning under partial observations.** In an MDP, where the system dynamics obey a Markovian property and are fully observable, reinforcement learning has been shown to be both computationally and statistically efficient [e.g., 52–55]. However, under partial observability, reinforcement learning can be computationally and statistically intractable in the worst case, even when assuming a Markovian latent state. Such worst-case hardness results are well-known. For example, [41, 56, 57] show that planning is computational intractable and [58, 59] show that learning is statistically intractable, in the worst-case. In these worst cases, the hardness comes from instances where the observations reveal little information about the latent state, which causes errors in learned representations to be uncontrollable. Accordingly, sub-classes of POMDPs have been identified in recent work where added structural conditions make efficient learning possible. One such condition is decodability [see e.g., 59–62], which assumes that the latent state can be decoded from the current observation (i.e., Block MDP), or an $m$-step history of observations. Another set of conditions is the "observability" condition [63–65] and its cousin the "weakly revealing" condition [36, 40] which require different belief states to induce distinguishable distributions over observations. These conditions are further extended to POMDP models in the function approximation setting, where the state or observation spaces are large and function approximators (e.g., linear functions) are used to represent the model. See, e.g., Uehara et al. [34], Guo et al. [61], Zhang et al. [62], Cai et al. [66], and Wang et al. [67]. In our work, we build on the above by identifying a class of POSTs/POSGs which can be learned efficiently. This is significant since POSTs/POSGs are much more general than POMDPs and do not assume the existence of a latent state.

**Predictive state representations.** Predictive state representations were introduced by Littman and Sutton [31] building on prior work on observable operator models by Jaeger [32] which proposed the idea of predictive representations as an alternative to belief states for modeling HMMs and POMDPs [see also 68–71]. PSRs are a way to *represent* the dynamics of a sequential decision-making problem by modeling the (conditional) probabilities of a small set of future trajectories, typically called "core tests". In a PSR, the probability of any future trajectory is a deterministic function of the conditional probabilities of the core tests. That is, the probabilities of the core tests encode all the information that the past contains about the future. Littman and Sutton [31] showed that any POMDP can be represented as a PSR. Various reinforcement learning methods for PSRs have been proposed under the assumption that data distribution is explorative, including spectral algorithms [72–74] and supervised learning approaches [75]. In addition, when it comes to the online setting where the algorithm needs to explore, there is a line of work that extends the theory and algorithms for online POMDP learning to PSRs. Moreover, some of these works propose generic theory and algorithms that can be applied to a large class of models including MDPs, POMDPs, and two-player zero-sum dynamic games with partial observability. See, e.g., Liu et al. [33], Huang et al. [35], Liu et al. [36],

Chen et al. [38], Zhan et al. [39], Zhong et al. [76], Qiu et al. [77], and Liu et al. [78]. Our work is particularly related to the works that combine the idea of optimism in the face of uncertainty [79] and maximum likelihood model estimation [33, 35, 36, 38, 39]. Specifically, our algorithm extends the UCB-type algorithm proposed by Huang et al. [35] for standard PSRs to learn a generalization of PSRs which captures POSTs/POSGs.

**Learning in multi-agent systems.** Most applications of interest in reinforcement learning involve the participation of multiple agents in the same environment. Empirical research has achieved striking success in several domains, including for example in the games of Go [26], Starcraft [28], and Poker [80], as well as in robotic control [24] and autonomous driving [27]. There also exists a growing literature of theoretical work. For example, Brafman and Tennenholtz [81], Bai et al. [82], and Song et al. [83] tackle learning in Markov games (MGs)—a generalization of single-agent MDPs that assumes the existence of a Markovian state which is observable by all agents. Another model which has been explored in the literature is imperfect-information extensive-form games (IIEFG), which assumes tree-structured transitions and deterministic emission, and can be viewed as a subclass of partially-observable Markov games (POMGs). Learning under this model has been studied in Zinkevich et al. [84], Kozuno et al. [85], and Farina and Sandholm [86]. More recently, Liu et al. [37] studied reinforcement learning in POMGs using an MLE-based algorithm building on their previous work in the single-agent setting [36]. To address the computational intractability of the planning step for such model-based algorithms, Liu and Zhang [87] proposed a quasi-efficient algorithm for multi-agent POMGs that runs in quasi-polynomial time with quasi-polynomial sample complexity. Their proposed algorithm leverages the common information approach [see 12] to construct an approximate Markov game where the state space of this new game corresponds to the space of approximate common information among agents. The idea of leveraging an information-sharing structure in multi-agent reinforcement learning has also appeared in Subramanian et al. [88], Mao et al. [89], Kara and Yuksel [90], Kao and Subramanian [91], and Tang et al. [92].

In each of the above-mentioned models (e.g., MDP, POMDP, MG, IIEFG, POMG, etc.), a particular fixed information structure is assumed. We emphasize that the POST/POSG model proposed in our work allows the information structure to be specified arbitrarily and hence captures these models as special cases within a unifying framework. Moreover, our analysis and proposed algorithm significantly expand the class of multi-agent sequential decision-making problems that can be efficiently learned.

## C Preliminaries

### C.1 Generic Sequential Decision-Making Problems

Consider a controlled stochastic process $(X_1, \ldots, X_H)$, where $X_h$ is a random variable corresponding to the variable at time $h$. At each time $h \in [H]$, the variable $X_h$ may be either an 'observation' (i.e., observable system variable) or an 'action'. The dynamics of this stochastic process are described by a tuple $(H, \{\mathbb{X}_h\}_h, \mathcal{O}, \mathcal{A}, \mathbb{P})$, where $H$ is the time horizon, $\mathbb{X}_h$ is the variable space at time $h$ (i.e., $X_h \in \mathbb{X}_h$), $\mathcal{O} \subset [H]$ is the index set of observations (i.e., $X_h$ is an observation if $h \in \mathcal{O}$), $\mathcal{A} \subset [H]$ is the index set of actions, and $\mathbb{P} = \{\mathbb{P}_h\}_{h \in \mathcal{O}}$ is a set of probability kernels which describes the the probability of any trajectory $x_1, \ldots, x_H$ given that the actions are executed,

$$\mathbb{P}\left[\{x_s : s \in \mathcal{O}\} \mid \{x_s : s \in \mathcal{A}\}\right] = \prod_{h \in \mathcal{O}} \mathbb{P}_h\left[x_h \mid x_1, \ldots, x_{h-1}\right]. \tag{8}$$

A choice of policy $\pi = \{\pi_h\}_{h \in \mathcal{A}}$ induces a probability distribution on $\mathbb{X}_1 \times \cdots \times \mathbb{X}_H$ as follows

$$\mathbb{P}^\pi\left(x_1, \ldots, x_H\right) = \prod_{h \in \mathcal{O}} \mathbb{P}_h\left(x_h \mid x_1, \ldots, x_{h-1}\right) \cdot \prod_{h \in \mathcal{A}} \pi_h\left(x_h \mid x_1, \ldots, x_{h-1}\right). \tag{9}$$

We now define some notation. Let $\mathbb{H}_h = \prod_{s \in 1:h} \mathbb{X}_s$ denote the space of histories at time $h$ and $\mathbb{F}_h = \prod_{s \in h+1:H} \mathbb{X}_s$ denote the space futures at time $h$. Similarly, let $\mathbb{H}_h^o = \mathrm{obs}(\mathbb{H}_h) = \prod_{s \in \mathcal{O}_{1:h}} \mathbb{X}_s$ denote the observation component of histories and let $\mathbb{H}_h^a = \mathrm{act}(\mathbb{H}_h) = \prod_{s \in \mathcal{A}_{1:h}} \mathbb{X}_s$ denote the action component. Here, $\mathcal{O}_{i:j}$ denotes $\mathcal{O} \cap \{i, \ldots, j\}$, and similarly for $\mathcal{A}_{i:j}$. The observation and action components of the futures, $\mathbb{F}_h^o$ and $\mathbb{F}_h^a$ respectively, are defined similarly.

We define the *system dynamics matrix* $\boldsymbol{D}_h \in \mathbb{R}^{|\mathbb{H}_h| \times |\mathbb{F}_h|}$ as the matrix giving the probability of each possible pair of history and future at time $h$ given the execution of the actions,

$$[\boldsymbol{D}_h]_{\tau_h, \omega_h} = \overline{\mathbb{P}}\left[\tau_h, \omega_h\right] = \mathbb{P}\left[\tau_h^o, \omega_h^o \mid \mathrm{do}(\tau_h^a, \omega_h^a)\right], \quad \tau_h \in \mathbb{H}_h, \omega_h \in \mathbb{F}_h, \tag{10}$$

where $\omega_h^o = \mathtt{obs}(\omega_h)$ are is the observation component of the future $\omega_h$, $\omega_h^a = \mathtt{act}(\omega_h)$ is the action component, and similarly for $\tau_h^o, \tau_h^a$. Note that the actions are actively executed via the $\mathrm{do}$-operation. Hence, the system dynamics matrices are independent of any action-selection criteria. Note that $\boldsymbol{D}_H \in \mathbb{R}^{|\mathbb{H}_H| \times 1}$ is defined as $[\boldsymbol{D}_H]_{\tau_H} = \overline{\mathbb{P}}\left[\tau_H\right]$, and $\boldsymbol{D}_0 = \boldsymbol{D}_H^\top$.

We introduce the notion of the *rank* of the dynamics. The rank of such a controlled stochastic process is the maximal rank of its dynamics matrices. This is a measure of the complexity of the dynamics.

**Definition** (Rank of dynamics; Definition 1). *The rank of the dynamics* $\{\boldsymbol{D}_h\}_{h \in [H]}$ *is* $r = \max_{h \in [H]} \mathrm{rank}(\boldsymbol{D}_h)$.

This defines the dynamics of the system. A sequential decision-making problem is such a controlled stochastic process together with an *objective*. The objective is defined by a reward function $R : \mathbb{X}_1 \times \cdots \times \mathbb{X}_H \to [0, 1]$ mapping a trajectory to a reward in $[0, 1]$. The agent(s) can affect the dynamics of the system through their choice of actions or policies. Each action $X_h, h \in \mathcal{A}$ may be chosen by either a single agent or one of several agents (e.g., a team). The policy at time $h \in \mathcal{A}$ is a mapping $\pi_h : \mathbb{H}_{h-1} \to \mathcal{P}(\mathbb{X}_h)$ from previous observations to an action (or a distribution over actions, if randomized). The collection of policies at all time steps is denoted $\boldsymbol{\pi} = (\pi_h : h \in \mathcal{A})$, and induces a probability distribution over trajectories, denoted $\mathbb{P}^{\boldsymbol{\pi}}$. Then, the value of a policy $\boldsymbol{\pi}$ is the expected value of the reward under the measure $\mathbb{P}^{\boldsymbol{\pi}}$, $V^R(\boldsymbol{\pi}) := \mathbb{E}^{\boldsymbol{\pi}}\left[R(X_1, \ldots, X_H)\right]$, where $\mathbb{E}^{\boldsymbol{\pi}}$ is the expectation associated with $\mathbb{P}^{\boldsymbol{\pi}}$.

The formalism of sequential decision-making problems introduced in this section is highly generic, but does not explicitly model the *information structure*. In the next section, we introduce the models of *partially observable sequential teams/games*, which explicitly represent information structures. We then show that the information structure characterizes the rank of a sequential decision-making problem as per Definition 1.

## C.2 (Generalized) Predictive State Representations

Predictive state representations (PSR) [31, 32] are a model of dynamical systems and sequential decision-making problems based on predicting future observations given the past, without explicitly modeling a latent state. In this section, we propose and formalize a generalization of standard PSRs.

In the standard formulation of sequential decision-making and predictive state representations, the sequence of variables is such that observations and actions always occur in an alternating manner (i.e., $o_h, a_h, o_{h+1}, a_{h+1}, \ldots$). The POST/POSG models we will propose are more general, and hence require a more flexible formalization of PSRs which allows for arbitrary order of observations and actions as well as arbitrary variable spaces at each time point. This generalization of PSRs will be used in our reinforcement learning algorithms.

The "PSR rank" of a sequential decision-making problem coincides with the rank of its dynamics, as defined in Definition 1. Recall that the system dynamics matrix $\boldsymbol{D}_h \in \mathbb{R}^{|\mathbb{H}_h| \times |\mathbb{F}_h|}$ is indexed by all possible observable histories $\tau_h$ and futures $\omega_h$. Denote the rank of the system dynamics at time $h$ by $r_h := \mathrm{rank}(\boldsymbol{D}_h)$.

Consider a sequential decision-making problem as defined in Section 2.1 (i.e., with an arbitrary order of observations and actions, and arbitrary variable spaces). At the heart of predictive state representations is the concept of "core test sets." A core test set at time $h$ is a set of futures such that the set of probabilities of those futures conditioned on the past encodes all the information that the past contains about the future. This is formalized in the definition below as a set of futures such that the submatrix of the full dynamics matrix restricted to those futures is full rank.

**Definition** (Core test sets). *A core test set at time $h$ is a subset of $d_h \geq r_h$ futures,* $\mathbb{Q}_h := \left\{q_h^1, \ldots, q_h^{d_h}\right\} \subset \mathbb{F}_h$, *such that the submatrix* $\boldsymbol{D}_h[\mathbb{Q}_h] \in \mathbb{R}^{|\mathbb{H}_h| \times d_h}$ *is full-rank,* $\mathrm{rank}(\boldsymbol{D}_h[\mathbb{Q}_h]) = \mathrm{rank}(\boldsymbol{D}_h) = r_h$.

A core test set implies the existence of a matrix $\boldsymbol{W}_h \in \mathbb{R}^{|\mathbb{F}_h| \times d_h}$ such that $\boldsymbol{D}_h = \boldsymbol{D}_h[\mathbb{Q}_h] \cdot \boldsymbol{W}_h^\top$.

Denote the $\tau_h$-th row of $\boldsymbol{D}_h[\mathbb{Q}_h]$ by,

$$\psi_h(\tau_h) := \left( \overline{\mathbb{P}} \left[ \tau_h, q_h^1 \right], \dots, \overline{\mathbb{P}} \left[ \tau_h, q_h^{d_h} \right] \right) \in \mathbb{R}^{d_h}. \tag{11}$$

The vector $\psi_h(\tau_h)$ is a sufficient statistic for the history $\tau_h$ in predicting the probabilities of all futures conditioned on $\tau_h$. This is sometimes called the *prediction features* of a history $\tau_h$.

For any integer $d_h \geq r_h$, there exists a core test set of size $d_h$. In particular, for any low-rank sequential decision-making problem, there exists a minimal core test set of size $r_h$ at each $h$. However, the minimal core test set depends on the system dynamics matrix $\boldsymbol{D}_h$, which is unknown in the learning setting. In the literature on reinforcement learning in PSRs, it is typically assumed that a core test set is known. We address the problem of constructing a PSR representation for POSTs/POSGs in Section 4.

For a core test set $\mathbb{Q}_h$, let $\mathbb{Q}_h^A = \{\text{act}(q) : q \in \mathbb{Q}_h\}$, where $\text{act}(q)$ denotes the action components of the test $q \in \mathbb{Q}_h$. Let $Q_A = \max_h \left| \mathbb{Q}_h^A \right|$ and $d = \max_h d_h$.

With core test sets defined, we are now ready to present the definition of a generalized predictive state representation. The essential element in a PSR is a set of operators $M_h : \mathbb{X}_h \to \mathbb{R}^{d_h \times d_{h-1}}$ for each time point $h \in [H]$. Given the prediction features at time $h - 1$, $\psi_{h-1}(x_1, \dots, x_{h-1}) \in \mathbb{R}^{d_{h-1}}$, the linear map $M_h(x_h)$ computes the prediction features at time $h$, incorporating the additional observation $x_h$. That is, $\psi_h(x_1, \dots, x_h) = M_h(x_h)\psi_{h-1}(x_1, \dots, x_{h-1})$. The full definition is given below.

**Definition** (Generalized Predictive State Representations; Definition 3). *Consider a sequential decision-making problem $(X_h \in \mathbb{X}_h)$ where $\mathcal{A}, \mathcal{O}$ partition $[H]$ into actions and observations, respectively. Then, a predictive state representation of this sequential decision-making problem is a tuple $\theta = (\{\mathbb{Q}_h\}_{0 \leq h \leq H-1}, \phi_H, \boldsymbol{M}, \psi_0)$ given by*

1. *$\{\mathbb{Q}_h\}_{0 \leq h \leq H-1}$ are core test sets, including for $h = 0$, where $\mathbb{Q}_0 = \{q_0^1, \dots, q_0^{d_0}\} \subset \mathbb{F}_0$ are core tests before the system begins.*

2. *$\psi_0 \in \mathbb{R}^{d_0}$ is the vector $\psi(\emptyset) = (\overline{\mathbb{P}}[q_0^1], \dots, \overline{\mathbb{P}}[q_0^{d_0}])$.*

3. *$\boldsymbol{M} = \{M_h\}_{1 \leq h \leq H-1}$ is a set of mappings $M_h : \mathbb{X}_h \to \mathbb{R}^{d_h \times d_{h-1}}$, from an observation/action to a matrix of size $d_h \times d_{h-1}$.*

4. *$\phi_H : \mathbb{X}_H \to \mathbb{R}^{d_{H-1}}$ is a mapping from the final observation to a $d_{H-1}$-dimensional vector.*

*This tuple satisfies,*

$$\overline{\mathbb{P}}[x_1, \dots, x_H] = \phi_H(x_H)^\top M_{H-1}(x_{H-1}) \cdots M_1(x_1) \psi_0 \tag{12}$$

$$\psi_h(x_1, \dots, x_h) = M_h(x_h) \cdots M_1(x_1) \psi_0, \forall h \tag{13}$$

To obtain a probability for a trajectory $\tau_h = (x_1, \dots, x_h)$, with $h < H$, note that $\sum_{\omega_h \in \mathbb{F}_h} \overline{\mathbb{P}}[\tau_h, \omega_h] = |\mathbb{F}_h^a| \overline{\mathbb{P}}[\tau_h]$. Hence

$$\overline{\mathbb{P}}[\tau_h] = \frac{1}{|\mathbb{F}_h^a|} \sum_{\omega_h} \overline{\mathbb{P}}[\tau_h, \omega_h]$$

$$= \frac{1}{\prod_{s \in h+1:H} |\mathbb{X}_s|^{\mathbf{1}\{s \in \mathcal{A}\}}} \sum_{x_H} \cdots \sum_{x_{h+1}} \phi_H^\top M_H(x_H) \cdots M_{h+1}(x_{h+1}) \psi_h(\tau_h).$$

Thus, if we recursively define $\phi_h$, $h < H$ via

$$\frac{1}{|\mathbb{X}_h|^{\mathbf{1}\{h \in \mathcal{A}\}}} \sum_{x_h} \phi_h^\top M_h(x_h) = \phi_{h-1}^\top, \tag{14}$$

with $\phi_H$ as the terminating condition, then, we can obtain $\overline{\mathbb{P}}[\tau_h]$ for any $h < H$, via an inner product between $\phi_h$ and $\psi_h(\tau_h)$,

$$\overline{\mathbb{P}}[\tau_h] = \phi_h^\top \psi_h(\tau_h). \tag{15}$$

Finally, if we define $\overline{\psi}_h(\tau_h) = \psi_h(\tau_h)/\overline{\mathbb{P}}[\tau_h]$, then we obtain the *conditional* probability of the core tests given the history, $\overline{\psi}_h(\tau_h) = (\overline{\mathbb{P}}[q_h^1 \mid \tau_h], \dots, \overline{\mathbb{P}}[q_h^{d_h} \mid \tau_h]) \in \mathbb{R}^{d_h}$. $\overline{\psi}_h(\tau_h)$ is known as the (normalized) prediction feature of the history $\tau_h$ [31].

**Remark 1** (Generality and difference from standard PSRs). *In standard PSRs, observations and actions are assumed to occur in an alternating manner, and hence observable operators are defined on pairs of observations and actions (i.e., $M_h(o_h, a_h)$). This structure leads to a somewhat simpler description compared to the above. However, our formulation is more general, as it allows each variable to be treated independently, and allows for an arbitrary sequence of variables with arbitrary spaces. This generality will be needed when modeling problems with an explicit representation of information structure.*

An important condition for the learnability of PSR models, which was used in prior work [including 33, 35], is the so-called "well-conditioning assumption". We state the analogous assumption for our generalized PSR model below.

**Assumption** ($\gamma$-well-conditioned generalized PSR; Assumption 1). *A PSR model $\theta = \left( \{\mathbb{Q}_h\}_{0 \leq h \leq H-1}, \phi_H, \boldsymbol{M}, \psi_0 \right)$, as defined in Definition 3, is said to be $\gamma$-well conditioned for $\gamma > 0$ if it satisfies*

*1. For any $h \in [H]$,*

$$\max_{\substack{z \in \mathbb{R}^{d_h} \\ \|z\|_1 \leq 1}} \max_{\pi} \sum_{\omega_h \in \mathbb{F}_h} \pi(\omega_h | \tau_h) \left| m_h(\omega_h)^\top z \right| \leq \frac{1}{\gamma}, \tag{16}$$

*where $m_h(\omega_h)^\top = \phi_H(x_H)^\top M_{H-1}(x_{H-1}) \cdots M_{h+1}(x_{h+1})$ with $\omega_h = (x_{h+1}, \ldots, x_H) \in \mathbb{F}_h$. The maximization is over policies $\pi$ such that for any fixed future observations $\omega_h^o$, $\sum_{\omega_h^a} \pi(\omega_h^o, \omega_h^a) = 1$.*

*2. For any $h \in [H-1]$,*

$$\max_{\substack{z \in \mathbb{R}^{d_h} \\ \|z\|_1 \leq 1}} \sum_{x_h \in \mathbb{X}_h} \|M_h(x_h)z\|_1 \, \pi(x_h) \leq \frac{\left| \mathbb{Q}_{h+1}^A \right|}{\gamma},$$

*where $\pi(x_h) = 1$ when $h \notin \mathcal{A}$ and $\sum_{x_h} \pi(x_h) = 1$ when $h \in \mathcal{A}$.*

To understand this condition, recall that $m_h(\omega_h)^\top \psi_h(\tau_h) = \overline{\mathbb{P}}[\tau_h, \omega_h]$. We may think of $z$ in Assumption 1 as representing the error in estimating $\psi_h(\tau_h)$, the probabilities of core tests at time $h$ given the history $\tau_h$. The $\gamma$-well-conditioned assumption ensures that the error in estimating the overall PSR (i.e., the probability of a particular trajectory) does not blow up when the estimation error of $\psi_h(\tau_h)$ is small.

The following result states that any sequential decision-making problem of the form described in Section 2.1 admits a generalized PSR representation. The proof and explicit construction are given in Appendix G.

**Proposition 1.** *Let $(X_1, \ldots, X_H)$ be any sequential decision-making problem with observation index set $\mathcal{O}$, action index set $\mathcal{A}$, and variable spaces $\{\mathbb{X}_h\}_{h \in [H]}$. Let $r_h = \mathrm{rank}(\boldsymbol{D}_h)$, where $D_h, h \in [H]$ are the system dynamics matrices. Then, there exists a PSR representation $\psi_0, \phi_H : \mathbb{X}_H \to \mathbb{R}^{r_{H-1}}$, $M_h : \mathbb{X}_h \to \mathbb{R}^{r_{h+1} \times r_h}, h \in [H-1]$, satisfying Definition 3.*

*Proof.* The proof is given in Appendix G. $\square$

## D  Examples of Information Structures and their Rank

The analysis in Section 3 and Theorem 1 characterizes the rank of any sequential decision-making problem as a function of its information structure. In this section, we illustrate this on several sequential decision-making problems, characterizing the information-structural complexity of their dynamics. The procedure is as follows: **1)** formulate the sequential decision-making problem as a POST/POSG; **2)** represent the information structure as a labeled directed acyclic graph $\mathcal{G}$; **3)** remove incoming edges into the action variables to produce $\mathcal{G}^\dagger$; **4)** apply Theorem 1 to find the information structural state at each point in time through a $d$-separation analysis.

**Illustration: translating to the POST/POSG framework.** We begin by illustrating how an arbitrary sequential decision-making problem can be formulated in the POST/POSG framework. Consider

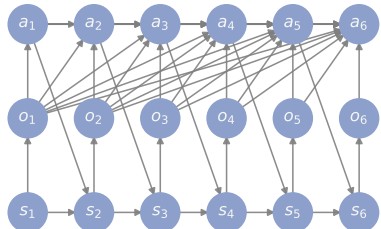 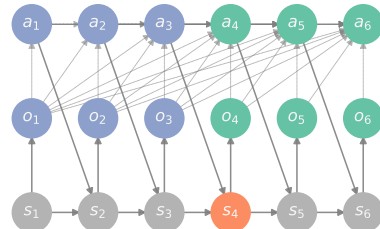

Figure 4: An illustrative example of the information-structural state for POMDPs. **Left.** The DAG representation of the information structure $\mathcal{G}$. **Right.** The DAG $\mathcal{G}^\dagger$ is depicted by drawing the edges corresponding to the information sets of the action variables with dotted lines. The information-structural state coincides with the Markovian state $s_t$, and is depicted in red. Future observables are drawn in green, and past observables are drawn in blue.

a POMDP with variables $(s_1, o_1, a_1, s_2, o_2, a_2, ...)$. This can be formulated as a POST/POSG by a simple relabelling of variables as follows.

$$
\begin{array}{ccccccccccc}
s_1 & o_1 & a_1 & s_2 & o_2 & a_2 & & s_t & o_t & a_t & \\
\downarrow & \downarrow & \downarrow & \downarrow & \downarrow & \downarrow & \cdots & \downarrow & \downarrow & \downarrow & \cdots \\
x_1 & x_2 & x_3 & x_4 & x_5 & x_6 & & x_{3t-2} & x_{3t-1} & x_{3t} &
\end{array}
$$

Here, the system variables $\mathcal{S}$ are the $s$-type and $o$-type variables, with system index set $\mathcal{S} = \{1, 2, 4, 5, 7, 8, ...\}$, and the action variables are the $a$-type variables with action index set $\mathcal{A} = \{3, 6, 9, ...\}$. The observable system variables are the $o$-type variables only, with index set $\mathcal{O} = \{2, 5, 8, ...\} \subset \mathcal{S}$. This can be done for any sequential decision-making problem.

To ease notation, let us not explicitly write the indices in this section, but rather use the original notation for the variables in the problem formulation. For example, we'll write $\mathcal{S} = \{s_t, o_t, t \in [T]\}$. Similarly, we use the notation $\mathcal{I}(x)$ to mean the information set corresponding to the variable $x$. Similarly, $\mathcal{I}^\dagger(x)$ denotes the information-structural state at the time when $x$ occurs. For example, in a POMDP $\mathcal{I}(s_t) = \{s_{t-1}, a_{t-1}\}$, $\mathcal{I}(o_t) = \{s_t\}$, and $\mathcal{I}(a_t) = \{o_{1:t}, a_{1:t}\}$.

Below, we will consider several examples of sequential decision-making problems, and apply the information-structural analysis of Theorem 1 to obtain a bound on the rank of the observable dynamics (which in turn implies a bound on the sample complexity, by Theorem 3).

**Decentralized POMDPs and POMGs.** At each time $t$, the system variables of a decentralized POMDP (or POMG) consist of a latent state $s_t$, observations for each agent $o_t^1, \ldots, o_t^N$, and actions of each agent $a_t^1, \ldots, a_t^N$. The latent state transitions are Markovian and depend on the agents' joint action. The observations are sampled via a kernel conditional on the latent state. Each agent can use their own history of observations to choose an action. Thus, the information structure is given by,

$$
\mathcal{I}(s_t) = \left\{ s_{t-1}, a_{t-1}^1, \ldots, a_{t-1}^N \right\}, \mathcal{I}(o_t^i) = \left\{ s_t \right\}, \mathcal{I}(a_t^i) = \left\{ o_{1:t-1}^i, a_{1:t-1}^i \right\}.
$$

Here, the observable variables are $\mathcal{U} = \left\{ o_{1:T}^i, a_{1:T}^i, i \in [N] \right\}$. By Theorem 1, we have $\mathcal{I}^\dagger(o_t^i) = \{s_t\}$, $\forall t, i$, as shown in Figure 5a. Thus, the rank of a Dec-POMDP is bounded by $|\mathbb{S}|$, where $\mathbb{S}$ is the state space. Note that in the case of models with a true latent state (e.g., POMDPs, Dec-POMDPs, and POMGs), the information-structural state coincides with the true latent state.

**Limited-memory information structures.** Consider a sequential decision making problem with variables $o_t, a_t, t \in [T]$ and an information structure with $m$-length memory. That is, observations can only depend directly on at most $m$ of the most recent observations and actions. That is, the information structure is

$$
\mathcal{I}(o_t) = \{o_{t-m:t-1}, a_{t-m:t-1}\}, \mathcal{I}(a_t) = \{o_{1:t}, a_{1:t-1}\}.
$$

The observables are all observations and actions, $\mathcal{U} = \{o_{1:T}, a_{1:T}\}$. By Theorem 1 we have that $\mathcal{I}^\dagger(o_t) = \{o_{t-m:t-1}, a_{t-m:t-1}\}$, as shown in Figure 5d. Hence, the rank of this sequential decision-making process is bounded by $|\mathbb{O}|^m |\mathbb{A}|^m$.

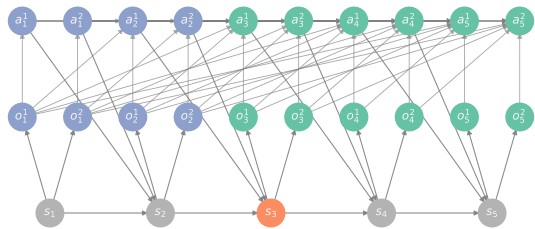

(a) Decentralized POMDP/POMG information-structure.

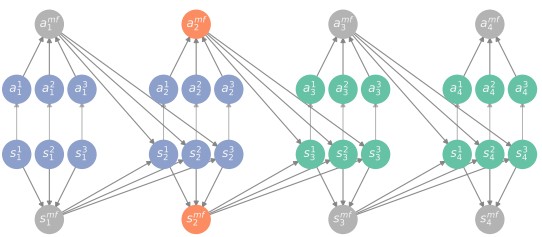

(b) "Mean-field" information structure.



(c) Point-to-point real-time communication with feedback information structure.

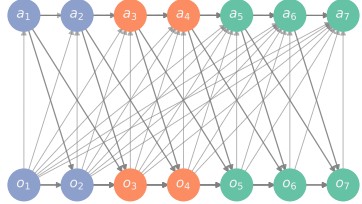

(d) Limited-memory ($m = 2$) information structures.

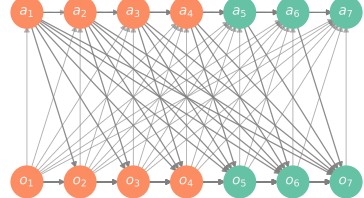

(e) Fully connected information structure.

Figure 5: DAG representation of various information structures. Solid edges indicate the edges in $\mathcal{E}^{\dagger}$ and light edges indicate the information sets of action variables. Grey nodes represent unobservable variables, blue nodes represent past observable variables, green nodes represent future observable variables, and red nodes represent the information structural state $\mathcal{I}_h^{\dagger}$. To find $\mathcal{I}_h^{\dagger}$, as per Theorem 1, we first remove the incoming edges into the action variables, then we find the minimal set among all past variables (both observable and unobservable) which $d$-separates the past observations from the future observations.

**Symmetric / "Mean-field" Information Structures.** Consider a sequential decision-making problem with $N$ agents. Each agent has their own local state, $s_t^i \in \mathbb{S}_{\text{local}}$. Similarly, at each time point, each agent takes an action $a_i^t \in \mathbb{A}_{\text{loc}}$. The global state $s_t = (s_t^1, \dots, s_t^N) \in \mathbb{S}_{\text{loc}}^N =: \mathbb{S}$ is composed by of all agents' local states. Similarly, the joint action space is $\mathbb{A} := \mathbb{A}_{\text{loc}}^N$. Consider a symmetric information structure where the evolution of each agent's local state depends only on a symmetric aggregation of all agents' states and actions, rather than on the local state/action of any particular agent. That is, the identity of who is in what state or takes which action does not matter—only the distribution of states and actions. This is often referred to as a "mean-field" setting (in the limit). Here, the transition depends only on the distribution of local states and actions, defined as $s_t^{\text{mf}} = \text{dist}(s_t) := \frac{1}{N} \delta_{s_t^i}$, $a_t^{\text{mf}} = \text{dist}(a_t) := \frac{1}{N} \delta_{a_t^i}$, for $s_t \in \mathbb{S}, a_t \in \mathbb{A}$. Different agents can have different transition kernels for their local state. Hence, by introducing $\text{dist}(s_t), \text{dist}(a_t)$ as auxiliary unobserved variables at each time $t$, we obtain the following information structure,

$$\mathcal{I}(s_t^i) = \{\text{dist}(s_{t-1}), \text{dist}(a_{t-1})\}, \ \mathcal{I}(a_t^i) = \{s_t^i\}$$

and an application of Theorem 1 bounds the rank by

$$\left|\mathbb{I}^\dagger(s_t^i)\right| < |\mathbb{S}_{\text{loc}}||\mathbb{A}_{\text{loc}}| \left(\frac{N}{|\mathbb{S}_{\text{loc}}| - 1} + 1\right)^{|\mathbb{S}_{\text{loc}}| - 1} \left(\frac{N}{|\mathbb{A}_{\text{loc}}| - 1} + 1\right)^{|\mathbb{A}_{\text{loc}}| - 1}.$$

This is compared to $|\mathbb{S}_{\text{loc}}|^N \cdot |\mathbb{A}_{\text{loc}}|^N$ (e.g., if we modeled this as an MDP with the state $s_t$), which is much larger when the number of agents is large. The information structure and $d$-separation decomposition are depicted in Figure 5b.

**Point-to-Point Real-Time Communication with Feedback.** Consider the following model of real-time communication with feedback. Let $x_t$ be the Markov source. At time $t$, the encoder receives the source $x_t \in \mathbb{X}$ and encodes sending a symbol $z_t \in \mathbb{Z}$. The symbol is sent through a memoryless noisy channel which outputs $y_t$ to the receiver. The decoder produces the estimate $\widehat{x}_t$. The output of the noisy channel is also fed back to the encoder. The encoder and decoder have full memory of their observations and previous "actions". The observation variables are $\mathcal{O} = \{x_{1:T}, \ y_{1:T}\}$ and the "actions" are $\mathcal{A} = \{z_{1:T}, \ \widehat{x}_{1:T}\}$. Hence, the information structure is given by the following,

$$\mathcal{I}(x_t) = \{x_{t-1}\}, \ \mathcal{I}(z_t) = \{x_{1:t}, y_{1:t-1}, z_{1:t-1}\}, \ \mathcal{I}(y_t) = \{z_t\}, \ \mathcal{I}(\widehat{x}_t) = \{y_{1:t}\}.$$

By Proposition 1, we have that,

$$\mathcal{I}^\dagger(x_t) = \{x_t\}, \ \mathcal{I}^\dagger(z_t) = \{x_t\}, \ \mathcal{I}^\dagger(y_t) = \{x_t, z_t\}, \ \mathcal{I}^\dagger(\widehat{x}_t) = \{x_t\}.$$

Hence, the rank is bounded by $|\mathbb{X}||\mathbb{Z}|$. This is depicted in Figure 5c.

**Fully-Connected Information Structures.** Consider a sequential decision making problem with variables $o_t, a_t, t \in [T]$ and a fully-connected information structure. That is, each observation directly depends on the entire history of observations and actions. Thus, the information structure is

$$\mathcal{I}(o_t) = \{o_{1:t-1}, a_{1:t-1}\}, \ \mathcal{I}(a_t) = \{o_{1:t}, a_{1:t-1}\}$$

The observables are all observations and actions, $\mathcal{U} = \{o_{1:T}, a_{1:T}\}$. By Theorem 1 we have that $\mathcal{I}^\dagger(o_t) = \{o_{1:t-1}, a_{1:t-1}\}$, as shown in Figure 5e. Hence, the rank of this sequential decision-making process can be exponential in the time horizon.

The examples above show that the tractability of a sequential decision-making problem in terms of the complexity of its dynamics depends directly on its information structure. This gives an interpretation of why certain models, like POMDPs, are more tractable than those with arbitrary information structures. Previous work primarily considers particular problem classes with fixed and highly regular information structures. In this work we argue for the importance of explicitly modeling the information structure of a sequential decision-making problem.

**Remark 2** (Necessity of generalized PSRs)**.** *The formalization of generalized PSRs in Section 2.2 was necessary to enable the study of information structure through POSTs/POSGs. An alternative (naive) solution to construct PSR representations for models with non-alternating observations and actions is to aggregate consecutive observations and actions to force them to obey the standard formulation of PSRs. This approach results in a loss of "resolution" in the information structure. That is, when you aggregate consecutive system variables, you also aggregate the DAG which represents the information structure, losing potentially important structure. In particular, in the worst case, such aggregation could result in an exponential increase in the rank of the dynamics. The examples given above elucidate this. Consider for example the "mean-field" information structure. If we aggregated local states and actions into a combined global state and joint action, the PSR rank would indeed be $|\mathbb{S}_{\text{loc}}|^N |\mathbb{A}_{\text{loc}}|^N$. By comparison, by considering each local state separately without aggregation, we are able to obtain a decomposition with a much smaller PSR rank.*

# E  UCB-Type Reinforcement Learning Algorithm for *Generalized* PSRs

We will adapt the model-based UCB-type algorithm of Huang et al. [35], extending it to generalized PSRs, including those representing POSTs. The algorithm involves the estimation of an upper confidence bound which captures the uncertainty in the estimated model and drives exploration so as to minimize this uncertainty. The UCB-based approach has the advantage of providing a last-iterate guarantee and requiring a weaker notion of planning oracle (a standard planning oracle instead of an optimistic planning oracle as required by similar algorithms). The technical contribution of this section is to extend the algorithm and its theoretical guarantees to generalized PSRs. The tools developed in doing so can be used to directly extend any other PSR-based algorithm to generalized PSRs.

When learning generalized PSRs, we suppose that the core test sets $\{\mathbb{Q}_h\}_{0 \leq h \leq H-1}$ are known by the algorithm. For example, if the sequential decision-making problem is a POST, Section 4 provides conditions under which $m$-step futures form core test sets. Let $\Theta$ be the set of $\gamma$-well-conditioned generalized PSR representations with $\{\mathbb{Q}_h\}_{0 \leq h \leq H-1}$ as core test sets. Denote by $\overline{\Theta}_\epsilon$ an optimistic $\epsilon$-cover of $\Theta$ (defined formally in Appendix G).

Recall that $d_h := |\mathbb{Q}_h|$ and $d = \max_h d_h$. Moreover, $\mathbb{Q}_h^A := \texttt{act}(\mathbb{Q}_h)$ are the action components of the core test sets and $Q_A := \max_h \left|\mathbb{Q}_h^A\right|$ is the maximal size of those action components. We define the exploration action sequences at time $h$ to be $\mathbb{Q}_{h-1}^{\exp} = \texttt{act}(\mathbb{X}_h \times \mathbb{Q}_h \cup \mathbb{Q}_{h-1})$. Moreover, we define $\mathsf{u}_{h-1}^{\exp}$ as the policy, defined from time $h-1$ onwards, in which each selection of action sequences in $\mathbb{Q}_{h-1}^{\exp}$ are chosen uniformly at random. For a model $\theta$ and reward function $R$, we define the value of a policy under this model and reward as $V_\theta^R(\pi) := \sum_{\tau_H} R(\tau_H) \mathbb{P}_\theta^\pi(\tau_H)$.

The algorithmic description is given in Algorithm 1. At each iteration $k$, the learner collects a trajectory $\tau_H^{k,h}$ for each time index $h \in [H]$ by using a particular policy that drives exploration so as to better estimate the parameters associated with the $h$-th time step. To collect the trajectory $\tau_H^{k,h}$, the learner executes the policy at the previous iteration, $\pi^{k-1}$, until time $h-1$ collecting the trajectory $\tau_{h-1}^{k,h}$ then executes $\mathsf{u}_{h-1}^{\exp}$ which samples action sequences from $\mathbb{Q}_{h-1}^{\exp}$ uniformly. The particular choice of the exploratory action sequences $\mathbb{Q}_{h-1}^{\exp}$ comes out of the proof (see proof of Lemma 5 in the appendix). Intuitively, $\texttt{act}(\mathbb{Q}_{h-1})$ allows us to estimate the prediction features $\overline{\psi}^*(\tau_{h-1}^{k,h}) = [\overline{\mathbb{P}}(q \mid \tau_{h-1}^{k,h})]_{q \in \mathbb{Q}_{h-1}}$, and $\texttt{act}(\mathbb{X}_h \times \mathbb{Q}_h)$ allows us to estimate $M_h^*(x_h)\overline{\psi}^*(\tau_{h-1}^{k,h})$.

The collected trajectories are added to the dataset, together with the policies used to collect them. The next step is model estimation via (constrained) maximum likelihood estimation. The algorithm estimates a model $\widehat{\theta}^k$ by selecting any model in a constrained set $\mathcal{B}^k$ defined as

$$\Theta_{\min}^k = \left\{\theta \in \Theta : \forall h, \ (\tau_h, \pi) \in \mathcal{D}_h^k, \ \mathbb{P}_\theta^\pi(\tau_h) \geq p_{\min}\right\},$$

$$\mathcal{B}^k = \left\{\theta \in \Theta_{\min}^k : \sum_{(\tau_H, \pi) \in \mathcal{D}^k} \log \mathbb{P}_\theta^\pi(\tau_H) \geq \max_{\theta' \in \Theta_{\min}^k} \sum_{(\tau_H, \pi) \in \mathcal{D}^k} \log \mathbb{P}_{\theta'}^\pi(\tau_H) - \beta\right\}. \tag{17}$$

The introduction of $\Theta_{\min}^k$ ensures that $\mathbb{P}_{\theta^*}^{\pi^{k-1}}(\tau_{h-1}^{k,h})$ is not too small so that the estimates of the prediction features $\overline{\psi}^*(\tau_{h-1}^{k,h}) = [\overline{\mathbb{P}}(q \mid \tau_{h-1}^{k,h})]_{q \in \mathbb{Q}_{h-1}}$ are accurate. This design differs from other MLE-based estimators [e.g., 33, 36, 38] due to the estimation of parameters capturing *conditional* probabilities.

Next, the algorithm chooses a policy which drives the algorithm to trajectories $\tau_h$ whose prediction features have so far been unexplored. To do this, Algorithm 1 constructs an upper confidence bound on the total variation distance between the estimated model and the true model. This is done via a bonus function $\widehat{b}^k(\tau_H)$,

$$\widehat{b}^k(\tau_H) = \min\left\{\alpha\sqrt{\sum_{h=0}^{H-1} \left\|\widehat{\overline{\psi}}(\tau_h)\right\|_{(\widehat{U}_h^k)^{-1}}^2}, 1\right\}, \quad \text{where,}$$

$$\widehat{U}_h^k = \lambda I + \sum_{\tau_h \in \mathcal{D}_{h,k}} \widehat{\overline{\psi}}^k(\tau_h)\widehat{\overline{\psi}}^k(\tau_h)^\top, \tag{18}$$

where $\lambda$ and $\alpha$ are pre-specified parameters to the algorithm. Thus, the bonus function captures the degree of uncertainty in the estimated prediction features $\widehat{\overline{\psi}}(\tau_h)$. In particular, the bonus $\widehat{b}(\tau_H)$ will be large for trajectories whose prediction feature $\widehat{\overline{\psi}}(\tau_h)$ lie far away from the empirical distribution of prediction features sampled in the dataset $\mathcal{D}_h^k$. This is captured by computing the norm with respect to the covariance $\widehat{U}_h^k$.

The algorithm then chooses an exploration policy for the next iteration which maximizes this upper confidence bound, hence collecting trajectories that have high uncertainty in their prediction features. When the estimated model is sufficiently accurate on all trajectories, the algorithm terminates and returns the optimal policy with respect to the reward function $R$ under the estimated model.

**Algorithm 1:** Learning Generalized PSRs (e.g., POSTs) via MLE and Exploration with UCB

---

**for** $k \leftarrow 1, \ldots, K$ **do**

    **for** $h \leftarrow 1, \ldots, H$ **do**

        Collect $\tau_H^{k,h} = (\omega_{h-1}^{k,h}, \tau_{h-1}^{k,h})$ using $\nu(\pi^{k-1}, \mathrm{u}_{h-1}^{\mathtt{exp}})$.

        $\mathcal{D}_{h-1}^k \leftarrow \mathcal{D}_{h-1}^{k-1} \cup \left\{ \left( \tau_H^{k,h}, \nu\left(\pi^{k-1}, \mathrm{u}_{h-1}^{\mathtt{exp}}\right) \right) \right\}$.

    **end**

    $\mathcal{D}^k = \left\{ \mathcal{D}_h^k \right\}_{h=0}^{H-1}$

    Compute MLE $\widehat{\theta} \in \mathcal{B}^k$, where

$$\Theta_{\min}^k = \left\{ \theta : \forall h, (\tau_h, \pi) \in \mathcal{D}_h^k, \mathbb{P}_\theta^\pi(\tau_h) \geq p_{\min} \right\},$$

$$\mathcal{B}^k = \left\{ \theta \in \Theta_{\min}^k : \sum_{(\tau_H, \pi) \in \mathcal{D}^k} \log \mathbb{P}_\theta^\pi(\tau_H) \geq \max_{\theta' \in \Theta_{\min}^k} \sum_{(\tau_H, \pi) \in \mathcal{D}^k} \log \mathbb{P}_{\theta'}^\pi(\tau_H) - \beta \right\}.$$

    Define the bonus function, $\widehat{b}^k(\tau_H) = \min \left\{ \alpha \sqrt{\sum_{h=0}^{H-1} \left\| \widehat{\overline{\psi}}(\tau_h) \right\|_{(\widehat{U}_h^k)^{-1}}^2}, 1 \right\}$, where

    $\widehat{U}_h^k = \lambda I + \sum_{\tau_h \in \mathcal{D}_{h^k}} \widehat{\overline{\psi}}^k(\tau_h) \widehat{\overline{\psi}}^k(\tau_h)^\top$.

    Solve the planning problem to maximize the bonus function $\pi^k = \arg\max_\pi V_{\widehat{\theta}^k}^{\widehat{b}^k}(\pi)$.

    **if** $V_{\widehat{\theta}^k}^{\widehat{b}^k}(\pi^k) \leq \epsilon/2$ **then**

        $\theta^\epsilon = \widehat{\theta}^k$. **break.**

    **end**

**end**

**return** $\pi = \arg\max_\pi V_{\theta^\epsilon}^R(\pi)$

---

We extend Huang et al.'s theoretical guarantees to show that Algorithm 1 enjoys polynomial sample complexity for *generalized* PSRs (Definition 3).

**Theorem 4.** *Suppose Assumption 1 holds. Suppose the parameters $p_{\min}, \lambda, \alpha, \beta$ are chosen appropriately. In particular, let*

$$p_{\min} \leq \frac{\delta}{KH \prod_{h=1}^H |\mathbb{X}_h|}, \ \lambda = \frac{\gamma \max_{s \in \mathcal{A}} |\mathbb{X}_s|^2 Q_A \beta \max\{\sqrt{r}, Q_A \sqrt{H}/\gamma\}}{\sqrt{dH}},$$

$$\alpha = O\left( \frac{Q_A \sqrt{dH\lambda}}{\gamma^2} + \frac{\max_{s \in \mathcal{A}} |\mathbb{X}_s| Q_A \sqrt{\beta}}{\gamma} \right), \ \beta = O\left( \log |\overline{\Theta}_\varepsilon| \right), \ \varepsilon \leq \frac{p_{\min}}{KH}.$$

*Then, with probability at least $1 - \delta$, Algorithm 1 returns a model $\theta^\epsilon$ and a policy $\pi$ that satisfy*

$$V_{\theta^\epsilon}^R(\pi^*) - V_{\theta^\epsilon}^R(\pi) \leq \varepsilon, \ and \ \forall \tilde{\pi}, \ \mathtt{D}_{\mathtt{TV}}\left( \mathbb{P}_{\theta^\epsilon}^{\tilde{\pi}}(\tau_H), \mathbb{P}_{\theta^*}^\pi(\tau_H) \right) \leq \varepsilon.$$

*In addition, the algorithm terminates with a sample complexity of,*

$$\tilde{O}\left( \left( r + \frac{Q_A^2 H}{\gamma^2} \right) \cdot \frac{rdH^3 \cdot \max_{s \in \mathcal{A}} |\mathbb{X}_s|^2 \cdot Q_A^4 \beta}{\gamma^4 \epsilon^2} \right).$$

*Proof.* The proof is given in Appendix J. $\qquad \square$

This result shows that the sample complexity of learning a generalized PSR depends on the problem size through a few key quantities. In particular, the sample complexity scales polynomially in the underlying rank $r$, the dimension of the PSR parameterization $d$, the size of the action component of the core tests $Q_A$, the time horizon $H$, the conditioning number $\gamma^{-1}$, the size of the action spaces $\max_{s \in \mathcal{A}} |\mathbb{X}_s|$, the log covering number $\log|\overline{\Theta}_\epsilon|$, and the desired suboptimality error $\epsilon$. Note that $\tilde{O}$ omits logarithmic dependence.

To apply this algorithm to a POST, we can use the generalized PSR parameterization constructed in Section 4. By Theorem 1 the PSR rank is bounded by $r \leq \max_h |\mathbb{I}_h^\dagger|$. If this POST is $\alpha$-robustly $\mathcal{I}^\dagger$-weakly revealing, then by Theorem 2 it admits a $\gamma$-well-conditioned generalized PSR parameterization with $\gamma = \alpha / \max_h |\mathbb{I}_h^\dagger|^{1/2}$ and the $m$-step futures as core test sets. Moreover, we have $d = \max_h d_h = \max_h |\mathbb{Q}_h^m|$. The following corollary states that Algorithm 1 can learn a partially-observable sequential team with a sample complexity which is polynomial in the size of the information-structural state space $\max_h |\mathbb{I}_h^\dagger|$.

**Corollary 1.** *Suppose a partially-observable sequential team is $m$-step $\alpha$-robustly $\mathcal{I}^\dagger$-weakly revealing as per Definition 6. Applying Algorithm 1 to this PSR representation, with parameters $p_{\min}, \lambda, \alpha, \beta$ chosen as in Theorem 4, returns a $\varepsilon$-optimal policy with a sample complexity of,*

$$\tilde{O}\left(\left(1 + \frac{Q_A^2 H}{\alpha^2}\right) \frac{\max_h |\mathbb{I}_h^\dagger|^7 \cdot \max_h |\mathbb{Q}_h^m| \cdot H^5 \cdot \max_{s \in \mathcal{A}} |\mathbb{X}_s|^2 \cdot \max_{s \in \mathcal{U}} |\mathbb{X}_s| \cdot Q_A^4}{\alpha^4 \epsilon^2}\right).$$

We can interpret this result as saying that the information structure of a sequential decision-making problem, through the quantity $\max_h |\mathbb{I}_h^\dagger|$, is fundamentally a measure of the complexity of the dynamics which need to be modeled. As a result, learning is tractable when $\max_h |\mathbb{I}_h^\dagger|$ is of modest size, and intractable otherwise. Recall that $\max_h |\mathbb{I}_h^\dagger|$ is small when there exists "state-like" variables, whether they are observable or unobservable. In this sense, $\max_h |\mathbb{I}_h^\dagger|$ is a fundamental quantity which generalizes the notion of a "state". For example, in the case of an $m$-step $\alpha$-weakly revealing POMDP, our algorithm has a sample complexity of $\text{poly}(S, (OA)^m, H, \alpha^{-1}) \cdot \epsilon^{-2}$, where $S$ is the size of the state space, $O$ is the size of the observation space, and $A$ is the size of the action space. This is similar to the sample complexity of [36, 37], which designed an algorithm tailored specifically for weakly-revealing POMDPs. Our algorithms, together with the POST/POSG models, enable sample-efficient reinforcement learning for a much broader class of models all within a unified framework.

In this section, we extended the algorithm in Huang et al. [35] to generalized PSRs, enabling sample-efficient learning of POSTs. We emphasize that other PSR-based algorithms can be extended in a similar manner. In the next section, we tackle the problem of learning in the game setting where different agents have different objectives.

# F   Extension to the game setting

## F.1   Partially Observable Sequential Games

In a POST, all agents share the same objective. In the game setting, different agents may have different objectives which compete with each other in interesting ways. Information structures play a crucial role in the study of games. The information available to one agent when making its decisions, compared to the information available to competing agents, determines how well it can achieve its objective. In particular, the information structure of a problem determines the set of equilibria it admits. There has been a plethora of work in the game theory community studying such problems.

Analogously to partially-observable sequential teams, we define partially-observable sequential *games* (POSGs). The dynamics of a POSG are identical to a POST, with the same formalization of variable structure, variable spaces, information structure, system kernels, and decision kernels. In contrast to a POST, agents in a POSG may have different objectives. In a POSG, there exists $N$ agents, with agent $i \in [N]$ deciding the actions at times $t \in \mathcal{A}^i$, where $\mathcal{A}^i \subset \mathcal{A}$. Each agent has its own objective defined by a reward function $R^i$. This is defined formally below.

**Definition 7** (Partially-Observable Sequential Game Model). *A partially-observable sequential game (POSG) is a controlled stochastic process consisting of the following components: variable structure, variable spaces, information structure, system kernels, decision kernels, and observability. These are defined in an identical manner to Definition 4. Additionally, POSGs define a **reward structure** as follows. Let $N$ be the number of agents. Each agent may act several times. Denote by $\mathcal{A}^i \subset \mathcal{A}$ the index of action variables associated to agent $i \in [N]$. Each agent has a reward function $R^i : \prod_{t \in \mathcal{U}} \mathbb{X}_t \to [0, 1]$ which they aim to maximize.*

Denote by $\pi^i = (\pi_t : t \in \mathcal{A}^i)$ the collection of decision kernels belonging to agent $i$, one for each action they take. Denote by $\boldsymbol{\pi} = (\pi^1, \ldots, \pi^N)$ the collection of all agents' policies. Fixing $\boldsymbol{\pi}$ induces

a probability distribution over $\mathbb{X}_1 \times \cdots \mathbb{X}_T$ in the same way as in the team setting,

$$\mathbb{P}^{\boldsymbol{\pi}} \left[ X_1 = x_1, \ldots X_T = x_t \right] = \prod_{t \in \mathcal{S}} \mathcal{T}_t(x_t | \{x_s : s \in \mathcal{I}_t\}) \prod_{t \in \mathcal{A}} \pi_t(x_t | \{x_s : s \in \mathcal{I}_t\}). \qquad (19)$$

The value of a policy $\boldsymbol{\pi}$ for agent $i \in [N]$ is defined as the expected value of their reward $R^i$ under $\mathbb{P}^{\boldsymbol{\pi}}$,

$$V^i(\boldsymbol{\pi}) \equiv V^i(\pi^i, \boldsymbol{\pi}^{-i}) := \mathbb{E}^{\boldsymbol{\pi}} \left[ R^i(X_{t(1)}, \ldots, X_{t(H)}) \right], \qquad (20)$$

where $\boldsymbol{\pi}^{-i} = (\pi^j : j \neq i)$.

The nature of randomization in agents' policies is crucial to the analysis of solution concepts in the game setting. To model randomized policies, which are potentially correlated, we introduce a random seed $\omega \in \Omega$ which is sampled at the beginning of an episode. Then, the policy at time $t \in \mathcal{A}$ can be modeled as a deterministic function mapping the seed $\omega$ and information variable $i_t \in \mathbb{I}_t$ to an action $\mathbb{X}_t$. That is, $\pi_t : \Omega \times \mathbb{I}_t \to \mathbb{X}_t$. To model independently randomized policies with each agent having private randomness, we consider the special case where the seed has the product structure $\omega = (\omega_1, \ldots, \omega_N) \in \Omega_1 \times \cdots \times \Omega_N$, and $\omega_i$ is the seed belonging to agent $i \in [N]$. Then, for $t \in \mathcal{A}^i, \pi_t : \Omega_i \times \mathbb{I}_t \to \mathbb{X}_t$. For each agent $i \in [N]$, define the three policy spaces,

1. Deterministic policies, $\Gamma_{\mathrm{det}}^i = \left\{ \pi^i : \pi^i = \left( \pi_t : \mathbb{I}_t \to \mathbb{X}_t, t \in \mathcal{A}^i \right) \right\}$,

2. Independently-randomized policies, $\Gamma_{\mathrm{ind}}^i = \left\{ \pi^i : \pi^i = \left( \pi_t : \Omega_i \times \mathbb{I}_t \to \mathbb{X}_t, t \in \mathcal{A}^i \right) \right\}$,

3. Correlated randomized policies, $\Gamma_{\mathrm{cor}}^i = \left\{ \pi^i : \pi^i = \left( \pi_t : \Omega \times \mathbb{I}_t \to \mathbb{X}_t, t \in \mathcal{A}^i \right) \right\}$.

Define the joint deterministic policy space, as $\boldsymbol{\Gamma}_{\mathrm{det}} = \Gamma_{\mathrm{det}}^1 \times \cdots \times \Gamma_{\mathrm{det}}^N$, and similarly for the independently-randomized policy space $\boldsymbol{\Gamma}_{\mathrm{ind}}$, and the correlated randomized policy space $\boldsymbol{\Gamma}_{\mathrm{cor}}$.

When studying games, a common question is to find an *equilibrium* within a particular policy space. At a high-level, an equilibrium is a joint policy where no agent can do better by deviating from their policy when the other agents keep their policies fixed. We will consider several notions of equilibrium. We begin by defining the notion of a *best-response*. Suppose that agent $i$'s policy space is $\Gamma^i$ (e.g., $\Gamma_{\mathrm{det}}^i, \Gamma_{\mathrm{ind}}^i$, or $\Gamma_{\mathrm{cor}}^i$). Then, we say that agent $i$'s policy $\pi^i$ is a best response to $\boldsymbol{\pi}^{-i}$ if there is no policy in $\Gamma^i$ which achieves a higher value. This is formalized in the definition below.

**Definition 8** (Best response). *For a joint policy $\boldsymbol{\pi}$, $\pi^i$ is said to be a best-response to $\boldsymbol{\pi}^{-i}$ in the policy space $\Gamma^i$ (e.g., $\Gamma_{\mathrm{det}}^i, \Gamma_{\mathrm{ind}}^i$, or $\Gamma_{\mathrm{cor}}^i$), if $V^i(\pi^i, \boldsymbol{\pi}^{-i}) = \max_{\tilde{\pi}^i \in \Gamma^i} V^i(\tilde{\pi}^i, \boldsymbol{\pi}^{-i}) =: V^{i,\dagger}(\boldsymbol{\pi}^{-i})$.*

This leads to the definition of two notions of equilibria. A *Nash Equilibrium* (NE) is a joint policy where all agents are best-responding in the space of independently-randomized policies. A *Coarse Correlated Equilibrium* (CCE) is a joint policy where all agents are best-responding in the space of correlated randomized policies. The difference between NE and CCE is that the randomness in the joint policy must be independent in an NE but can be correlated in a CCE. Since $\Gamma_{\mathrm{ind}} \subset \Gamma_{\mathrm{cor}}$, coarse correlated equilibria are a generalization of Nash equilibria. We define them formally below.

**Definition 9** (Nash Equilibrium). *A joint policy $\boldsymbol{\pi} \in \Gamma_{\mathrm{ind}}$ is said to be a Nash equilibrium if for all agents $i \in [N]$, $V^i(\boldsymbol{\pi}) = \max_{\tilde{\pi}^i \in \Gamma_{\mathrm{ind}}^i} V^i(\tilde{\pi}^i, \boldsymbol{\pi}^{-i}) =: V^{i,\dagger}(\boldsymbol{\pi}^{-i})$. A joint policy $\boldsymbol{\pi} \in \Gamma_{\mathrm{ind}}$ is said to an $\varepsilon$-approximate Nash equilibrium if $V^i(\boldsymbol{\pi}) \geq V^{i,\dagger}(\boldsymbol{\pi}^{-i}) - \varepsilon$ for all $i \in [N]$.*

**Definition 10** (Coarse Correlated Equilibrium). *A joint policy $\boldsymbol{\pi} \in \Gamma_{\mathrm{cor}}$ is said to be a coarse correlated equilibrium if for all agents $i \in [N]$, $V^i(\boldsymbol{\pi}) = \max_{\tilde{\pi}^i \in \Gamma_{\mathrm{cor}}^i} V^i(\tilde{\pi}^i, \boldsymbol{\pi}^{-i}) =: V^{i,\dagger}(\boldsymbol{\pi}^{-i})$. A joint policy $\boldsymbol{\pi} \in \Gamma_{\mathrm{cor}}$ is said to an $\varepsilon$-approximate Nash equilibrium if $V^i(\boldsymbol{\pi}) \geq V^{i,\dagger}(\boldsymbol{\pi}^{-i}) - \varepsilon$ for all $i \in [N]$.*

Since we consider finite-space sequential games, an equilibrium is guaranteed to exist [93].

**Remark 3** (Notion of equilibrium can be represented through information structure). *The policy classes defined above (i.e., deterministic, independently-randomized, correlated randomized) can be directly modeled by the information structure. For example, to represent correlated randomized policies, the random seed $\omega \in \Omega$ can be modeled as an observable variable at time $t = 0$ which is in all agents' information sets. Similarly, independently randomized policies can be represented through a different random seed for each agent at time $t = 0$, and including the appropriate random seed in each action's information set. Hence, the information structure itself can decide which equilibrium notion we are interested in. Moreover, this allows us to consider additional notions of equilibrium*

*where, for example, only subsets of agents can be correlated with each other (e.g., this may be useful in modeling multi-team problems). Note that adding random seeds in order to model randomized policies does not affect the information-structural state $\mathcal{I}_h^\dagger$ since the seeds don't appear in $\mathcal{G}^\dagger$. For concreteness, we focus on NE and CCE in our presentation.*

## F.2   Characterizing the Sample Complexity of Learning an Equilibrium via a Self-Play Algorithm as a function of the Information Structure

We now introduce a sample-efficient reinforcement learning algorithm for learning well-conditioned generalized predictive state representations in the *game* setting with each agent having their own objective. In particular, since partially-observable sequential games with a $\mathcal{I}^\dagger$-weakly revealing information structure admit a well-conditioned generalized PSR representation, they can also be learned sample-efficiently by this algorithm.

The algorithm we propose is a *self-play* algorithm for learning an *equilibrium* of the dynamic game problem. That is, the algorithm specifies the policies of all agents during the learning phase, collecting the trajectory of observables at each episode to improve its estimate of the system dynamics. This can be thought of as a centralized agent playing against itself. We will propose an algorithm which can find a Nash equilibrium or coarse correlated equilibrium in a sample-efficient manner. We begin with some preliminaries.

**Game setting.**   Recall that a sequential decision-making problem falls within the game setting if each agent has their own objective. Following Section 2.1, we consider a sequential decision-making problem $(X_1, \ldots, X_H)$ where $\mathcal{O}$ denotes the index set of (observable) system variables and $\mathcal{A}$ denotes the set of action variables. We suppose the game involves $N$ agents, and denote the action index set of each agent by $\mathcal{A}^i \subset \mathcal{A}$, where $\{\mathcal{A}^i\}_{i \in [N]}$ partitions $\mathcal{A}$. Each agent has their own reward function $R^i(X_1, \ldots, X_H)$. Note that POSGs as defined in Definition 7 are structured models which fall within this framework.

**Equilibria and policy classes.**   Recall that in the game setting, the type of randomization in each agent's policy affects the set of equilibria in the game. In Appendix F.1, we formalized this randomization by introducing a random seed $\omega \in \Omega$ and allowing each agent's policy to be a function of their information set and this seed. If the seed has a product structure $\omega = (\omega_1, \ldots, \omega_N)$ with each agent observing their own seed, this results in independently-randomized policies, denoted by $\Gamma_{\mathrm{ind}}^i$. If all agents use the same seed, this results in correlated randomized policies, which we denote by $\Gamma_{\mathrm{cor}}^i$. An equilibrium among independently randomized policies is called a Nash equilibrium and an equilibrium among correlated randomized policies is called a coarse correlated equilibrium.

**Estimating probabilities in the planner.**   The probability of any trajectory under a joint policy $\boldsymbol{\pi}$ is given by $\mathbb{P}^{\boldsymbol{\pi}}(\tau_H) = \sum_\omega \overline{\mathbb{P}}[\tau_H] \boldsymbol{\pi}(\tau_H|\omega)\mathbb{P}[\omega]$, where $\overline{\mathbb{P}}[\tau_H] = \mathbb{P}[\tau_H^o \mid \tau_H^a]$ as before, and $\boldsymbol{\pi}(\tau_H \mid \omega) = \prod_{h \in \mathcal{A}} \mathbf{1}\{x_h = \pi_h(\tau_{h-1}, \omega)\}$. Recall that the probabilities $\overline{\mathbb{P}}[\tau_H]$ are estimated by the generalized PSR model $\widehat{\theta}$. We assume that the planner has knowledge of the randomization, $\mathbb{P}[\omega]$. Hence, the planner in the self-play algorithm is able to compute the probability of any trajectory for each choice of policy.

**Algorithm.**   The algorithmic description is presented in Algorithm 2. In the first stage of the algorithm, the centralized learning agent has a unified goal: to explore the environment. This is done by executing policies which maximize the bonus function $\widehat{b}^k(\tau_H)$ by visiting trajectories with imprecise estimates of their probability, as measured by the upper confidence bound on the total variation distance. This part is identical to Algorithm 1. Once the algorithm is sufficiently confident about the estimated probabilities of all trajectories, it computes the equilibrium using the estimated model directly. That is, ComputeEquilibrium computes either NE or CCE. The only difference in the exploration stage of the algorithm compared to Algorithm 1 is that the termination condition involves $\varepsilon/4$ rather than $\varepsilon/2$ in order to guarantee an $\varepsilon$-approximate equilibrium under the added complications of the game setting.

**Theorem 5.** *Suppose Assumption 1 holds. Suppose the parameters $p_{\min}, \lambda, \alpha, \beta$ are chosen as in Theorem 4. Then, with probability at least $1 - \delta$, Algorithm 2 returns a model $\theta^\epsilon$ and a policy $\pi$ which is an $\varepsilon$-approximate equilibrium (either NE or CCE). That is,*

$$ V_{\theta^*}^i(\pi) \geq V_{\theta^*}^{i,\dagger}(\pi^{-i}) - \varepsilon, \ \forall i \in [N]. $$

---

**Algorithm 2:** Self-play UCB Algorithm for Sequential Games

---

**for** $k \leftarrow 1, \ldots, K$ **do**

    **for** $h \leftarrow 1, \ldots, H$ **do**

        Collect $\tau_H^{k,h} = (\omega_{h-1}^{k,h}, \tau_{h-1}^{k,h})$ using $\nu(\pi^{k-1}, \mathbf{u}_{h-1}^{\mathtt{exp}})$.

        $\mathcal{D}_{h-1}^{k} \leftarrow \mathcal{D}_{h-1}^{k-1} \cup \left\{ \left( \tau_H^{k,h}, \nu\left(\pi^{k-1}, \mathbf{u}_{h-1}^{\mathtt{exp}}\right) \right) \right\}$.

    **end**

    $\mathcal{D}^k = \left\{ \mathcal{D}_h^k \right\}_{h=0}^{H-1}$

    Compute MLE $\widehat{\theta} \in \mathcal{B}^k$, where

$$\Theta_{\min}^k = \left\{ \theta : \forall h, (\tau_h, \pi) \in \mathcal{D}_h^k, \mathbb{P}_\theta^\pi(\tau_h) \geq p_{\min} \right\},$$

$$\mathcal{B}^k = \left\{ \theta \in \Theta_{\min}^k : \sum_{(\tau_H, \pi) \in \mathcal{D}^k} \log \mathbb{P}_\theta^\pi(\tau_H) \geq \max_{\theta' \in \Theta_{\min}^k} \sum_{(\tau_H, \pi) \in \mathcal{D}^k} \log \mathbb{P}_{\theta'}^\pi(\tau_H) - \beta \right\}.$$

    Define the bonus function, $\widehat{b}^k(\tau_H) = \min\left\{ \alpha \sqrt{\sum_{h=0}^{H-1} \left\| \widehat{\overline{\psi}}(\tau_h) \right\|_{(\widehat{U}_h^k)^{-1}}^2}, 1 \right\}$, where

    $\widehat{U}_h^k = \lambda I + \sum_{\tau_h \in \mathcal{D}_{h^k}} \widehat{\overline{\psi}}^k(\tau_h) \widehat{\overline{\psi}}^k(\tau_h)^\top$.

    Solve the planning problem $\pi^k = \arg\max_\pi V_{\widehat{\theta}^k}^{\widehat{b}^k}(\pi)$.

    **if** $V_{\widehat{\theta}^k}^{\widehat{b}^k}(\pi^k) \leq \epsilon/4$ **then**

        $\theta^\epsilon = \widehat{\theta}^k$. **break.**

    **end**

**end**

**return** $\pi = \mathtt{ComputeEquilibrium}(\theta^\epsilon, \left\{ R^1, \ldots, R^N \right\})$

---

*In addition, the algorithm terminates with a sample complexity of,*

$$\tilde{O}\left( \left( r + \frac{Q_A^2 H}{\gamma^2} \right) \cdot \frac{rdH^3 \cdot \max_{s \in \mathcal{A}} |\mathbb{X}_s|^2 \cdot Q_A^4 \beta}{\gamma^4 \epsilon^2} \right).$$

*Proof.* The proof is given in Appendix K. $\qquad\square$

To apply this algorithm to a partially-observable sequential game, we can use the generalized PSR parameterization constructed in Section 4.

**Corollary 2.** *Suppose a partially-observable sequential game is $m$-step $\alpha$-robustly $\mathcal{I}^\dagger$-weakly revealing as per Definition 6. Applying Algorithm 2 to this PSR representation, with parameters $p_{\min}, \lambda, \alpha, \beta$ chosen as in Theorem 5, returns a $\varepsilon$-approximate equilibrium $\pi$ with a sample complexity of,*

$$\tilde{O}\left( \left( 1 + \frac{Q_A^2 H}{\alpha^2} \right) \cdot \frac{\max_h |\mathbb{I}_h^\dagger|^7 \cdot \max_h |\mathbb{Q}_h^m| \cdot H^5 \cdot \max_{s \in \mathcal{A}} |\mathbb{X}_s|^2 \cdot \max_{s \in \mathcal{U}} |\mathbb{X}_s| \cdot Q_A^4}{\alpha^4 \epsilon^2} \right).$$

## G    Existence of Generalized PSR representations and their covering number

In this section we show that any rank-$r$ sequential decision-making problem (as per Section 2.1) can be represented via a rank-$r$ generalized PSR (Definition 3). Next, we bound the covering number of the class of rank $r$ PSRs, which will be important for our MLE analysis. Similar results have been established in previous work for sequential decision-making problems with alternating observations and actions [e.g., 33]. Recall that our formulation of the generic sequential decision-making problem and generalized PSRs is more general than the standard formulation since it allows for an arbitrary sequence of variables. Here, we follow a similar procedure to prove a slightly generalized result.

**Proposition 2** (Existence of Generalized PSR representation). *Consider a sequential decision-making problem with* $\operatorname{rank}(\boldsymbol{D}_h) = r_h$, $h \in 0 : H - 1$. *There exists a generalized PSR representation (i.e, observable operator model)* $b_0, \{B_h(x_h)\}_{h \in [H], x_h \in \mathbb{X}_h}, \{v_h\}_{h \in 0:H}$ *such that,*

1. $B_h(x_h) \in \mathbb{R}^{r_h \times r_{h-1}}$ *and* $\|B_h(x_h)\|_2 \leq 1$ *for any* $x_h$.

2. $|b_0| \leq \sqrt{|\mathbb{H}_H^a|}$.

3. $\|v_h\|_2 \leq \sqrt{|\mathbb{F}_h^o| / |\mathbb{F}_h^a|}$.

4. *For any* $h$, $\frac{1}{|\mathbb{X}_h|^{1\{h \in \mathcal{A}\}}} v_h^\top \sum_{x_h \in \mathbb{X}_h} B_h(x_h) = v_{h-1}^\top$.

5. *For any* $\tau_h \in \mathbb{H}_h$, $\overline{\mathbb{P}}[\tau_h] = v_h^\top B_h(x_h) \cdots B_1(x_1) b_0$.

*Proof.* We construct the representation via the singular value decomposition of the matrix $\boldsymbol{D}_h^\top$. Let $U_h \in \mathbb{R}^{|\mathbb{F}_h| \times r_h}, \Sigma_h \in \mathbb{R}^{r_h \times r_h}, V_h^\top \in \mathbb{R}^{r_h \times |\mathbb{H}_h|}$ be the SVD such that $\boldsymbol{D}_h^\top = U_h \Sigma_h V_h^\top$. Define $b_0, B_h, v_h^\top$ as follows,

$$b_0 = \|\boldsymbol{D}_0\|_2 \,, \quad B_h(x_h) = U_h^\top [U_{h-1}]_{(x_h, \mathbb{F}_h),:} \,, \quad v_h^\top = \frac{1}{|\mathbb{F}_h^a|} \mathbf{1}^\top U_h.$$

Here, $[U_{h-1}]_{(x_h, \Omega_h),:}$ denotes an $|\mathbb{F}_h|$ by $r_{h-1}$ submatrix of $U_{h-1}$ consisting of the rows $(x_h, \omega_h)$, $\omega_h \in \mathbb{F}_h$ (i.e., the set of futures where the variable at time $h$ is $x_h$). Note that $|\mathbb{F}_H^a| = 1$ by convention, a product over an empty set. We verify each property in turn.

First, $\|B_h(x_h)\|_2 = \|U_h^\top [U_{h-1}]_{(x_h, \mathbb{F}_h),:}\|_2 \leq 1$ since $U_h, U_{h-1}$ are unitary matrices. Second,

$$|b_0| = \|\boldsymbol{D}_0\|_2 = \sqrt{\sum_{\tau_H} \overline{\mathbb{P}}[\tau_H]^2}$$

$$\leq \sqrt{\sum_{\tau_H} \overline{\mathbb{P}}[\tau_H]} = \sqrt{\sum_{\tau_H^a} \sum_{\tau_H^o} \mathbb{P}[\tau_H^o \mid \tau_H^a]} = \sqrt{\sum_{\tau_H^a} 1} = \sqrt{\prod_{s \in \mathcal{A}} |\mathbb{X}_s|},$$

where the inequality is since $\overline{\mathbb{P}}[\tau_H] \in [0, 1]$. For property 3, we have

$$\|v_h\|_2 = \frac{1}{|\mathbb{F}_h^a|} \|\mathbf{1}^\top U_h\|_2$$

$$\leq \frac{1}{|\mathbb{F}_h^a|} \|\mathbf{1}\|_2 = \frac{\sqrt{|\mathbb{F}_h|}}{|\mathbb{F}_h^a|} = \sqrt{|\mathbb{F}_h^o| / |\mathbb{F}_h^a|},$$

where the inequality is since $U_h$ is unitary, and the final equality is since $|\mathbb{F}_h| = |\mathbb{F}_h^o| |\mathbb{F}_h^a|$.

Next, to prove properties 4 and 5, we first show the following claim.

**Claim.** *For any history* $\tau_h = (x_1, \dots, x_h) \in \mathbb{H}_h$, $h \in 0 : H$, *we have* $B_h(x_h) \cdots B_1(x_1) b_0 = U_h^\top [\boldsymbol{D}_h^\top]_{:, \tau_h}$.

*Proof of claim.* We prove the claim by induction. In the base case, $h = 0$, $\boldsymbol{D}_0^\top$ is a vector in $\mathbb{R}^{\mathbb{F}_0}$ (note that $\mathbb{F}_0 = \mathbb{H}_H$). Hence, $U_0$ is simply the normalized vector $U_0 = \boldsymbol{D}_0^\top / \|\boldsymbol{D}_0^\top\|_2$, and hence $U_0^\top \boldsymbol{D}_0^\top = \boldsymbol{D}_0 \boldsymbol{D}_0^\top / \|\boldsymbol{D}_0\|_2 = \|\boldsymbol{D}_0\|_2 = b_0$. Proceeding by induction, suppose the claim holds for $h - 1$. Then, we have,

$$B_h(x_h) \cdots B_1(x_1) b_0 = B_h(x_h) U_{h-1}^\top [\boldsymbol{D}_{h-1}^\top]_{:, \tau_{h-1}}$$

$$= U_h^\top [U_{h-1}]_{(x_h, \mathbb{F}_h),:} U_{h-1}^\top [\boldsymbol{D}_{h-1}^\top]_{:, \tau_{h-1}}$$

$$= U_h^\top [U_{h-1} U_{h-1}^\top \boldsymbol{D}_{h-1}^\top]_{(x_h, \mathbb{F}_h), \tau_{h-1}}$$

$$= U_h^\top [\boldsymbol{D}_{h-1}^\top]_{(x_h, \mathbb{F}_h), \tau_{h-1}}$$

$$= U_h^\top [\boldsymbol{D}_h^\top]_{:, \tau_h},$$

where the final equality is because $\left[\boldsymbol{D}_{h-1}^\top\right]_{(x_h,\omega_h),\tau_{h-1}} = \overline{\mathbb{P}}\left[\tau_{h-1}, x_h, \omega_h\right] = \overline{\mathbb{P}}\left[\tau_h, \omega_h\right] = \left[\boldsymbol{D}_h^\top\right]_{\omega_h,\tau_h}$. $\qquad\square$

Using this fact, we can now show property 5 as follows,

$$
\begin{aligned}
v_h^\top B_h(x_h)\cdots B_1(x_1)b_0 &= \frac{1}{|\mathbb{F}_h^a|}\mathbf{1}^\top U_h U_h^\top \left[\boldsymbol{D}_h^\top\right]_{:,\tau_h} = \frac{1}{|\mathbb{F}_h^a|}\mathbf{1}^\top \left[\boldsymbol{D}_h^\top\right]_{:,\tau_h}\\
&= \frac{1}{|\mathbb{F}_h^a|}\sum_{\omega_h\in\mathbb{F}_h}\overline{\mathbb{P}}\left[\tau_h, \omega_h\right] = \frac{1}{|\mathbb{F}_h^a|}\sum_{\omega_h^a\in\mathbb{F}_h^a}\sum_{\omega_h^o\in\mathbb{F}_h^o}\mathbb{P}\left[\tau_h^o, \omega_h^o \mid \tau_h^a, \omega_h^a\right]\\
&= \frac{1}{|\mathbb{F}_h^a|}\mathbb{P}\left[\tau_h^o \mid \tau_h^a\right]\sum_{\omega_h^a\in\mathbb{F}_h^a}\sum_{\omega_h^o\in\mathbb{F}_h^o}\mathbb{P}\left[\omega_h^o \mid \omega_h^a, \tau_h^a, \tau_h^o\right]\\
&= \frac{1}{|\mathbb{F}_h^a|}\mathbb{P}\left[\tau_h^o \mid \tau_h^a\right]\sum_{\omega_h^a\in\mathbb{F}_h^a}1\\
&= \mathbb{P}\left[\tau_h^o \mid \tau_h^a\right].
\end{aligned}
$$

Finally, it remains to show property 4. Consider the linear equation $x^\top U_h^\top \boldsymbol{D}_h^\top = |\mathbb{F}_h^a|^{-1}\mathbf{1}^\top \boldsymbol{D}_h^\top$. Note that $U_h^\top \boldsymbol{D}_h^\top \in \mathbb{R}^{r_h\times|\mathbb{H}_h|}$ is rank $r_h$. Thus, this equation has a unique solution. Our strategy is to show that $v_h^\top$ and $v_{h+1}^\top \sum_{x_{h+1}} B_{h+1}(x_h)$ are both solutions to this linear equation, and hence $v_h^\top = v_{h+1}^\top \sum_{x_{h+1}} B_{h+1}(x_h)$. That $v_h^\top$ is a solution is clear by definition of $v_h$, $v_h^\top U_h^\top \boldsymbol{D}_h^\top = |\mathbb{F}_h^a|^{-1}\mathbf{1}^\top U_h U_h^\top \boldsymbol{D}_h^\top = |\mathbb{F}_h^a|^{-1}\mathbf{1}^\top \boldsymbol{D}_h^\top$. First, recall by the calculation above that $|\mathbb{F}_h^a|^{-1}\mathbf{1}^\top \boldsymbol{D}_h^\top$ is a vector in $\mathbb{R}^{\mathbb{H}_h}$ where the $\tau_h$-th entry is $\mathbb{P}\left[\tau_h^o \mid \tau_h^a\right]$. We will calculate the $\tau_h$-th entry of the vector $x^\top U_h^\top \boldsymbol{D}_h$ when $x^\top = v_{h+1}^\top \sum_{x_{h+1}} B_{h+1}(x_{h+1})$,

$$
\begin{aligned}
\left(v_{h+1}^\top \sum_{x_{h+1}} B_{h+1}(x_{h+1})\right)\left[U_h^\top \boldsymbol{D}_h^\top\right]_{:,\tau_h} &= \frac{1}{|\mathbb{F}_{h+1}^a|}\sum_{x_{h+1}}\mathbf{1}^\top U_{h+1} U_{h+1}^\top \left[U_h\right]_{(x_{h+1},\mathbb{F}_{h+1}),:}\left[U_h^\top \boldsymbol{D}_h^\top\right]_{:,\tau_h}\\
&= \frac{1}{|\mathbb{F}_{h+1}^a|}\sum_{x_{h+1}}\mathbf{1}^\top U_{h+1} U_{h+1}^\top \left[U_h U_h^\top \boldsymbol{D}_h^\top\right]_{(x_{h+1},\mathbb{F}_{h+1}),\tau_h}\\
&= \frac{1}{|\mathbb{F}_{h+1}^a|}\sum_{x_{h+1}}\left[\mathbf{1}^\top \boldsymbol{D}_h^\top\right]_{(x_{h+1},\mathbb{F}_{h+1}),\tau_h}\\
&= \frac{1}{|\mathbb{F}_{h+1}^a|}\sum_{x_{h+1}}\sum_{\omega_{h+1}}\overline{\mathbb{P}}\left[\tau_h, x_{h+1}, \omega_{h+1}\right] = \frac{1}{|\mathbb{F}_{h+1}^a|}\sum_{\omega_h\in\mathbb{F}_h}\overline{\mathbb{P}}\left[\tau_h, \omega_h\right]\\
&= \frac{1}{|\mathbb{F}_{h+1}^a|}\mathbb{P}\left[\tau_h^o \mid \tau_h^a\right]\sum_{\omega_h^a\in\mathbb{F}_h^a}\sum_{\tau_h^o\in\mathbb{F}_h^o}\mathbb{P}\left[\omega_h^o \mid \tau_h, \omega_h^a\right]\\
&= \frac{1}{|\mathbb{F}_{h+1}^a|}\mathbb{P}\left[\tau_h^o \mid \tau_h^a\right]\sum_{\omega_h^a\in\mathbb{F}_h^a}1 = \frac{1}{|\mathbb{F}_{h+1}^a|}|\mathbb{F}_h^a|\mathbb{P}\left[\tau_h^o \mid \tau_h^a\right]\\
&= \frac{1}{|\mathbb{X}_{h+1}|^{\mathbf{1}\{h+1\in\mathcal{A}\}}}\mathbb{P}\left[\tau_h^o \mid \tau_h^a\right],
\end{aligned}
$$

where the final inequality is since $|\mathbb{F}_h^a| = \prod_{s\in h+1:H}(|\mathbb{X}_s|^{\mathbf{1}\{s\in\mathcal{A}\}})$. $\qquad\square$

**Corollary 3.** *Consider a sequential decision-making problem with* $\mathrm{rank}(D_h)\leq r$. *Then, there exists a generalized PSR* $b_0\in\mathbb{R}^r$, $\{B_h(x_h)\}_{h\in[H],x_h\in\mathbb{X}_h}\subset\mathbb{R}^{r\times r}$, $v_H\in\mathbb{R}^r$ *such that,*

1. $\|B_h(x_h)\|_2\leq 1, \forall h, x_h\in\mathbb{X}_h, \|b_0\|_2\leq\sqrt{|\mathbb{H}_H^a|}$, *and* $\|v_H\|_2\leq 1$.

2. *For any* $\tau_H\in\mathbb{H}_H$, $\overline{\mathbb{P}}\left[\tau_H\right] = v_H^\top B_H(x_H)\cdots B_1(x_1)b_0$.

*Proof.* In Proposition 2 we constructed such a representation with dimensions in terms of $r_h$ instead of $r$. Since $r_h \leq r$, we can pad this representation with dummy columns and/or rows filled with zeros to obtain a representation with dimensions in terms of $r$. $\qquad\square$

An important part of maximum likelihood analysis is the notion of a "bracketing number" which controls the complexity of the model class $\Theta$ [e.g., 94]. In our analysis, the model class is the set of generalized PSRs of a given rank. As shown in the results above, rank-$r$ generalized PSRs can represent any rank-$r$ sequential decision-making problem, with operators whose norm is bounded. In the next result, we will consider a closely related notion to the bracketing number which crucially incorporates optimism. $\overline{\Theta}_\varepsilon$ is said to be an "optimistic $\varepsilon$-cover" for $\Theta$ if for each $\theta \in \Theta$, there exists $\widehat{\theta} \in \overline{\Theta}_\varepsilon$ with an associated probability measure $\overline{\mathbb{P}}_{\widehat{\theta}}^\varepsilon$ such that,

$$\forall h, \tau_h, \ \overline{\mathbb{P}}_{\widehat{\theta}}^\varepsilon(\tau_h) \geq \overline{\mathbb{P}}_\theta\left[\tau_h\right],$$
$$\forall h, \tau_h, \ \sum_{\tau_h} \left| \overline{\mathbb{P}}_{\widehat{\theta}}^\varepsilon(\tau_h) - \overline{\mathbb{P}}_\theta\left[\tau_h\right] \right| \leq \varepsilon.$$

The first condition ensures optimism and the second condition ensures that $\overline{\Theta}_\varepsilon$ $\varepsilon$-covers $\Theta$, in the sense that the probability of any trajectory is approximated within an error $\varepsilon$. Recall that the parameter $\beta$ in Algorithms 1 and 2, which appears in the sample complexity results in Theorems 4 and 5, is defined in terms of $|\overline{\Theta}_\varepsilon|$. The next proposition bounds the size of $|\overline{\Theta}_\varepsilon|$.

**Proposition 3** (Optimistic cover of sequential decision making problems). *Let $\mathfrak{M}$ be the set of all rank-$r$ sequential decision-making problems with a horizon of length $H$, observation index set $\mathcal{O} \subset [H]$, action index set $\mathcal{A} \subset [H]$, and variable spaces $\mathbb{X}_1, \ldots, \mathbb{X}_H$. Then, there exists an optimistic $\varepsilon$-cover $\overline{\Theta}_\varepsilon$ of $\Theta$ with cardinality bounded by,*

$$\log\left|\overline{\Theta}_\varepsilon\right| \leq O\left( r^2 \max_h |\mathbb{X}_h| \, H^2 \log\left( \frac{\max_h |\mathbb{X}_h|}{\epsilon} \right) \right).$$

*Proof.* Define the set of generalized PSR representations constructed in Corollary 3,

$$\Theta := \left\{ b_0 \in \mathbb{R}^r, \{B_h(x_h)\}_{h, x_h}, v_H \in \mathbb{R}^r : \ \|B_h(x_h)\|_2 \leq 1, \forall h, x_h, \|b_0\|_2 \leq \sqrt{|\mathbb{H}_H^a|}, \|v_H\|_2 \leq 1, \right.$$
$$\text{and } \forall \, \tau_H \in \mathbb{H}_H, \ \overline{\mathbb{P}}_m\left[\tau_H\right] = v_H^\top B_H(x_H) \cdots B_1(x_1) b_0,$$
$$\left. \text{where } m \text{ is a sequential decision making problem in } \mathfrak{M} \right\}.$$

Let $\mathcal{C}_\delta$ be a $\delta$-cover of the above set with respect to the $\ell_\infty$-norm. For $\widehat{\theta} = (b_0, \{B_h(x_h)\}, v_H) \in \mathcal{C}_\delta$, define the $\varepsilon$-optimistic probabilities as,

$$\overline{\mathbb{P}}_{\widehat{\theta}}^\varepsilon(\tau_H) := v_H^\top B_H(x_h) \cdots B_1(x_1) b_0 + \varepsilon/2$$

We will show that for an appropriate choice of $\delta$, $\mathcal{C}_\delta$ is an optimistic $\varepsilon$-cover. In particular, for each $\theta \in \Theta$, there exists $\widehat{\theta} \in \mathcal{C}_\delta$ such that,

$$\forall h, \tau_h, \ \overline{\mathbb{P}}_{\widehat{\theta}}^\varepsilon(\tau_h) \geq \overline{\mathbb{P}}_\theta\left[\tau_h\right],$$
$$\forall h, \tau_h, \ \sum_{\tau_h} \left| \overline{\mathbb{P}}_{\widehat{\theta}}^\varepsilon(\tau_h) - \overline{\mathbb{P}}_\theta\left[\tau_h\right] \right| \leq \varepsilon.$$

To choose the value of $\delta$ for which the above holds, observe that

$$\sum_{\tau_H} \left| \widehat{v}_H^\top \widehat{B}_H(x_H) \cdots B_1(x_1)\widehat{b}_0 - v_H^\top B_H(x_H) \cdots B_1(x_1)b_0 \right|$$

$$\leq \sum_{h=1}^{H} \sum_{\tau_H} \left| \widehat{v}_H^\top \widehat{B}_H(x_H) \cdots \widehat{B}_{h+1}(x_{h+1})(\widehat{B}_h(x_h) - B_h(x_h))B_{h-1}(x_{h-1}) \cdots B_1(x_1)b_0 \right|$$

$$+ \sum_{\tau_H} \left| \widehat{v}_H^\top B_H(x_H) \cdots B_1(x_1)(\widehat{b}_0 - b_0) \right|$$

$$\leq \sum_{h} \sum_{\tau_H} r \left\| \widehat{B}_h(x_h) - B_h(x_h) \right\|_{\max} \sqrt{|\mathbb{H}_H^a|} + \sum_{\tau_H} \sqrt{r} \left\| \widehat{b}_0 - b_0 \right\|_\infty$$

$$\leq H \max_h |\mathbb{X}_h|^{H+|\mathcal{A}|/2} r\delta + \max_h |\mathbb{X}_h|^H \sqrt{r}\delta,$$

where the second inequality uses $\|\widehat{v}_H\|_2 = \|v_H\|_2 = \|B_h(x_h)\|_2 = 1$, $\|\widehat{B}_h(x_h) - B_h(x_h)\|_2 \leq r\|\widehat{B}_h(x_h) - B_h(x_h)\|_{\max} \leq r\delta$, $\|b_0\|_2 \leq \sqrt{|\mathbb{H}_H^a|}$, and $\|\widehat{b}_0 - b_0\|_2 \leq \sqrt{r}\|\widehat{b}_0 - b_0\|_\infty \leq \sqrt{r}\delta$. Hence, choosing $\delta := \varepsilon \cdot \max_h |\mathbb{X}_h|^{-cH}$ for $c$ an absolute constant large enough achieves a $\varepsilon$-optimistic covering of $\Theta$. Hence, we let $\overline{\Theta}_\varepsilon = \mathcal{C}_\delta$, with $\delta = \varepsilon \cdot \max_h \cdot |\mathbb{X}_h|^{-cH}$. It remains to bound the size of $|\overline{\Theta}_\varepsilon|$.

Recall that $\|\cdot\|_\infty \leq \|\cdot\|_2$ and that an interval $[-x, x]$ in $\mathbb{R}$ admits a $\delta$-cover of size bounded by $2x/\delta$. Now, observe that $\max_{ij} |[B_h(x_h)]_{ij}| \leq \|B_h(x_h)\|_2 \leq 1$. Hence, for a fixed $h$, $\{B_h(x_h)\}_{x_h}$ admits a cover of size bounded by $(2/\delta)^{r^2 |\mathbb{X}_h|}$. Considering all $h$, the cover is bounded by $(2/\delta)^{r^2 \sum_h |\mathbb{X}_h|} \leq (2/\delta)^{r^2 \max_h |\mathbb{X}_h| H}$. For, $b_0$, we have $\|b_0\|_\infty \leq \|b_0\|_2 \leq \sqrt{|\mathbb{H}_H^a|}$, hence the covering number is bounded by $(2\sqrt{|\mathbb{H}_H^a|}/\delta)^r$. Finally for $v_H$, we have $\|v_H\|_\infty \leq \|v_H\|_2 \leq 1$, hence the covering number is bounded by $(2/\delta)^r$. Thus, we have,

$$\log \left| \overline{\Theta}_\varepsilon \right| \leq O\left( r^2 \max_h |\mathbb{X}_h| H \log\left(\frac{1}{\delta}\right) \right).$$

Recalling that $\delta = \varepsilon \max_s |\mathbb{X}_s|^{-cH}$, we obtain that,

$$\log \left| \overline{\Theta}_\varepsilon \right| \leq O\left( r^2 \max_h |\mathbb{X}_h| H^2 \log\left(\frac{\max_h |\mathbb{X}_h|}{\epsilon}\right) \right).$$

$\square$

## H   Proofs of Section 3.2

**Theorem** (Restatement of Theorem 1)**.** *The rank of the observable system dynamics of a POST or POSG is bounded by*

$$r \leq \max_{h \in [H]} \left| \mathbb{I}_h^\dagger \right|.$$

*Proof.* We have

$$[\boldsymbol{D}_h]_{\tau_h, \omega_h} = \mathbb{P}\left[\tau_h^o, \omega_h^o \mid \mathrm{do}(\tau_h^a, \tau_h^a)\right]$$

$$= \mathbb{P}\left[\tau_h^o \mid \mathrm{do}(\tau_h^a)\right] \mathbb{P}\left[\omega_h^o \mid \tau_h^o; \mathrm{do}(\tau_h^a, \omega_h^a)\right]$$

$$\overset{(a)}{=} \mathbb{P}\left[\tau_h^o \mid \mathrm{do}(\tau_h^a)\right] \sum_{\substack{x_k \in \mathbb{X}_k \\ k \in \mathcal{I}_h^\dagger}} \mathbb{P}\left[\left\{x_k, k \in \mathcal{I}_h^\dagger\right\} \mid \tau_h^o; \mathrm{do}(\tau_h^a, \omega_h^a)\right] \mathbb{P}\left[\omega_h^o \mid \left\{x_k, k \in \mathcal{I}_h^\dagger\right\}, \tau_h^o; \mathrm{do}(\tau_h^a, \omega_h^a)\right]$$

$$\overset{(b)}{=} \sum_{\substack{x_k \in \mathbb{X}_k \\ k \in \mathcal{I}_h^\dagger}} \mathbb{P}\left[\tau_h^o \mid \mathrm{do}(\tau_h^a)\right] \mathbb{P}\left[\left\{x_k, k \in \mathcal{I}_h^\dagger\right\} \mid \tau_h^o; \mathrm{do}(\tau_h^a)\right] \mathbb{P}\left[\omega_h^o \mid \left\{x_k, k \in \mathcal{I}_h^\dagger\right\}, \tau_h^o; \mathrm{do}(\tau_h^a, \omega_h^a)\right]$$

$$\overset{(c)}{=} \sum_{\substack{x_k \in \mathbb{X}_k \\ k \in \mathcal{I}_h^\dagger}} \mathbb{P}\left[\tau_h^o \mid \mathrm{do}(\tau_h^a)\right] \mathbb{P}\left[\left\{x_k, k \in \mathcal{I}_h^\dagger\right\} \mid \tau_h^o; \mathrm{do}(\tau_h^a)\right] \mathbb{P}\left[\omega_h^o \mid \left\{x_k, k \in \mathcal{I}_h^\dagger\right\}; \mathrm{do}(\omega_h^a)\right],$$

where step (a) is simply the law of total probability, step (b) is that $\{x_k, k \in \mathcal{I}_h^\dagger\}$ is conditionally independent of $\mathrm{do}(\omega_h^a)$ (future actions) given $(\tau_h^o; \mathrm{do}(\tau_h^a))$ (the past), and step (c) is that $\omega_h^o$ is conditionally independent of $(\tau_h^o; \mathrm{do}(\tau_h^a))$ given $\{x_k, k \in \mathcal{I}_h^\dagger\}$. This is due to a result by Verma and Pearl [95] which states: for three sets of variables $A, B, C$ in a directed graphical model, if $A$ and $B$ are $d$-separated by $C$, then $A \perp B \mid C$. Recall that $\mathcal{I}_h^\dagger$ is defined as the minimal set which $d$-separates $(X_{t(1)}, \ldots, X_{t(h)})$ from $(X_{t(h+1)}, \ldots, X_{t(H)})$.

As a technical remark, note that $i_h^\dagger = (x_k, k \in \mathcal{I}_h^\dagger)$ may include actions and hence,

$$\mathbb{P}\left[ \left\{ x_k, \, k \in \mathcal{I}_h^\dagger \right\} \, \Big| \, \tau_h^o; \, \mathrm{do}\,(\tau_h^a) \right] = \mathbb{P}\left[ \left\{ x_k, \, k \in \mathcal{I}_h^\dagger \cap \mathcal{S} \right\} \, \Big| \, \tau_h^o; \, \mathrm{do}\,(\tau_h^a) \right] \mathbf{1} \left\{ \left( x_k, \, k \in \mathcal{I}_h^\dagger \cap \mathcal{A} \right) \text{ matches } \tau_h^a \right\},$$

since the action components of $i_h^\dagger$ are contained in the history $\tau_h$.

Now define two matrices

$$\boldsymbol{D}_{h,1} := \left[ \mathbb{P}\left[\tau_h^o \mid \mathrm{do}\,(\tau_h^a)\right] \mathbb{P}\left[ \left\{ x_k, \, k \in \mathcal{I}_h^\dagger \right\} \, \Big| \, \tau_h^o; \, \mathrm{do}\,(\tau_h^a) \right] \right]_{\tau_h, i_h^\dagger}, \quad \tau_h \in \mathbb{H}_h, \; i_h^\dagger \equiv \left( x_k, \, k \in \mathcal{I}_h^\dagger \right) \in \mathbb{I}_h^\dagger,$$

$$\boldsymbol{D}_{h,2} := \left[ \mathbb{P}\left[ \omega_h^o \, \Big| \, \left\{ x_k, \, k \in \mathcal{I}_h^\dagger \right\}; \, \mathrm{do}\,(\omega_h^a) \right] \right]_{i_h^\dagger, \omega_h}, \quad i_h^\dagger \equiv \left( x_k, \, k \in \mathcal{I}_h^\dagger \right) \in \mathbb{I}_h^\dagger, \; \omega_h \in \mathbb{F}_h.$$

We have that $\boldsymbol{D}_h = \boldsymbol{D}_{h,1}\boldsymbol{D}_{h,2}$, where both $\boldsymbol{D}_{h,1}$ and $\boldsymbol{D}_{h,2}$ have rank upper bounded by $|\mathbb{I}_h^\dagger| = \prod_{s \in \mathcal{I}_h^\dagger} |\mathbb{X}_s|$. Hence, $\mathrm{rank}(\boldsymbol{D}_h) \le |\mathbb{I}_h^\dagger|$, and the result follows. $\qquad\square$

# I   Proofs of Section 4

**Lemma** (Restatement of Lemma 1). *Suppose that the POST/POSG is $m$-step $\mathcal{I}^\dagger$-weakly revealing. Then, $\mathbb{Q}_h^m$ is a core test set for all $h \in [H]$. Furthermore, we have*

$$\overline{\mathbb{P}}\left[\tau_h, \omega_h\right] = \langle m_h(\omega_h), \psi_h(\tau_h) \rangle, \text{ and } \overline{\mathbb{P}}\left[\omega_h \mid \tau_h\right] = \langle m_h(\omega_h), \overline{\psi}_h(\tau_h) \rangle. \tag{21}$$

*Proof.* Let $\tau_h \in \mathbb{H}_h$, $\omega_h \in \mathbb{F}_h$ be any history and future, respectively. By Theorem 1, recall that we have

$$\overline{\mathbb{P}}[\omega_h \mid \tau_h] = \sum_{i_h^\dagger \in \mathbb{I}_h^\dagger} \overline{\mathbb{P}}\left[\omega_h \, \Big| \, i_h^\dagger \right] \mathbb{P}\left[i_h^\dagger \, \Big| \, \tau_h\right]. \tag{22}$$

Recall that $i_h^\dagger$ may overlap with $\tau_h$. In particular, the action component of $i_h^\dagger$ is contained in $\tau_h$. Thus, $\mathbb{P}[i_h^\dagger \mid \tau_h] = \mathbb{P}[\{x_k, \, k \in \mathcal{I}_h^\dagger \setminus \mathcal{U}_{1:h}\} \mid \tau_h] \cdot \mathbf{1}\{(x_k, \, k \in \mathcal{I}_h^\dagger \cap \mathcal{U}_{1:h})$ matches $\tau_h\}$. Note that $\mathcal{I}_h^\dagger \setminus \mathcal{U}_{1:h} \subset \mathcal{S}$ does not contain any actions. Hence, the summation over $\mathbb{I}_h^\dagger$ is equivalent to summing over its unobservable components with the restriction that its observable components match $\tau_h$.

Define the mappings $\tilde{m}_h \colon \mathbb{F}_h \to \mathbb{R}^{|\mathbb{I}_h^\dagger|}$ and $p_h \colon \mathbb{H}_h \to \mathbb{R}^{|\mathbb{I}_h^\dagger|}$ by

$$\tilde{m}_h(\omega_h) = \left[ \overline{\mathbb{P}}\left[\omega_h \, \Big| \, i_h^\dagger\right] \right]_{i_h^\dagger \in \mathbb{I}_h^\dagger}, \quad p_h(\tau_h) = \left[ \mathbb{P}\left[i_h^\dagger \, \Big| \, \tau_h\right] \right]_{i_h^\dagger \in \mathbb{I}_h^\dagger}.$$

Then, we have that the conditional probability of the future $\omega_h$ given the past $\tau_h$ is given by the inner product of the above mappings, $\overline{\mathbb{P}}\left[\omega_h \mid \tau_h\right] = \langle \tilde{m}_h(\omega_h), p_h(\tau_h) \rangle$. Recall that the vector of (conditional) core test set probabilities for the history $\tau_h$ is given by $\overline{\psi}_h(\tau_h) = [\mathbb{P}\left[q^o \mid \tau_h^o; \mathrm{do}(\tau_h^a), \mathrm{do}(q^a)\right]]_{q \in \mathbb{Q}_h^m} \in \mathbb{R}^{|\mathbb{Q}_h^m|}$. By the definition of $\boldsymbol{G}_h$ and Equation (22), we have $\boldsymbol{G}_h \, p_h(\tau_h) = \psi_h(\tau_h)$, since, for $q \in \mathbb{Q}_h^m$,

$$
\begin{aligned}
(\boldsymbol{G}_h \, p_h(\tau_h))_q &= \sum_{i_h^\dagger} (\boldsymbol{G}_h)_{q, i_h^\dagger} \, (p_h(\tau_h))_{i_h^\dagger} \\
&= \sum_{i_h^\dagger} \mathbb{P}\left[q^o \, \Big| \, i_h^\dagger; \, \mathrm{do}(q^a)\right] \mathbb{P}\left[i_h^\dagger \, \Big| \, \tau_h\right] \\
&= \mathbb{P}\left[q^o \mid \tau_h^o; \, \mathrm{do}(\tau_h^a), \mathrm{do}(q^a)\right] \\
&=: \left[\overline{\psi}_h(\tau_h)\right]_q
\end{aligned}
$$

Since by assumption $\text{rank}(\boldsymbol{G}_h) = \left|\mathbb{I}_h^\dagger\right|$, its pseudo-inverse $\boldsymbol{G}_h^\dagger$ is a left inverse of $\boldsymbol{G}_h$ (i.e., $\boldsymbol{G}_h^\dagger \boldsymbol{G}_h = I$). Hence, multiplying on the left by $\boldsymbol{G}_h^\dagger$, we obtain $p_h(\tau_h) = \boldsymbol{G}_h^\dagger \overline{\psi}_h(\tau_h)$. Hence,

$$
\begin{aligned}
\overline{\mathbb{P}}\left[\omega_h \mid \tau_h\right] &= \left\langle \tilde{m}_h(\omega_h), \boldsymbol{G}_h^\dagger \overline{\psi}_h(\tau_h) \right\rangle \\
&= \left\langle \underbrace{\left(\boldsymbol{G}_h^\dagger\right)^\top \tilde{m}_h(\omega_h)}_{m_h(\omega_h)}, \overline{\psi}_h(\tau_h) \right\rangle.
\end{aligned}
$$

That $\overline{\mathbb{P}}\left[\tau_h, \omega_h\right] = \langle m_h(\omega_h), \psi_h(\tau_h)\rangle$ follows directly by noting the definition of $\overline{\psi}_h(\tau_h) := \psi_h(\tau_h)/\overline{\mathbb{P}}\left[\tau_h\right]$.

Hence, we have shown that for the test set $\mathbb{Q}_h^m$, the probability of each future $\omega_h$ given a history $\tau_h$ is a linear combination of the probabilities of each test in the core test set with weights $m_h(\omega_h) := (\boldsymbol{G}_h^\dagger)^\top \tilde{m}_h(\omega_h) \in \mathbb{R}^{|\mathbb{Q}_h^m|}$ depending only on the future and not the history. $\qquad \square$

**Theorem** (Restatement of Theorem 2). *Suppose a POST/POSG is $\alpha$-robustly $m$-step $\mathcal{I}^\dagger$-weakly revealing. Then, the corresponding generalized PSR as constructed in Section 4 is $\gamma$-well-conditioned with $\gamma = \alpha/\max_h |\mathbb{I}_h^\dagger|^{1/2}$.*

We will first show that $(\{\mathbb{Q}_h^m\}_h, \phi_H, \{M_h\}_h, \psi_0)$ indeed forms a PSR through a series of simple calculations.

A direct corollary of Lemma 1 is the following.

**Lemma 2.** *For any $h \in [H]$, $\tau_{h-1} \in \mathbb{H}_{h-1}$, $x_{t(h)} \in \mathbb{X}_{t(h)}$, $\omega_h \in \mathbb{F}_h$, we have*

$$
\overline{\mathbb{P}}\left[\tau_{h-1}, x_{t(h)}, \omega_h\right] = \left\langle m_{h-1}(x_{t(h)}, \omega_h), \psi_h(\tau_{h-1}) \right\rangle. \tag{23}
$$

Hence, given a history $\tau_{h-1} = (x_{t(1)}, \ldots, x_{t(h-1)})$, having observed another variable $x_{t(h)}$, we can update our predictions of the future and obtain the probability of trajectories of the form $(\tau_{h-1}, x_{t(h)}, \omega_h)$ for any future $\omega_h \in \mathbb{F}_h$. Note that $x_{t(h)}$ may be either an observation or an action. Hence, we can update our prediction of the future after deciding an action, and before receiving the next observation. This is in contrast to the standard PSR formulation where predictions of the future can only be updated with a *pair* of observation and action. Our formulation provides additional flexibility, which is crucial for the general information structures modeled by POSTs and POSGs.

This means that, after observing $x_{t(h)}$, we can use the $m_h : \mathbb{F}_h \to \mathbb{R}^{d_h}$ mapping constructed in Lemma 1 to update the probability of any candidate future $\omega_h$. We are particularly interested in updating the probabilities of the futures corresponding to the *core test set* at the next time point, since this provides a sufficient statistic of the past. Thus, we define the matrix mapping $M_h : \mathbb{X}_{t(h)} \to \mathbb{R}^{d_h \times d_{h-1}}$ by,

$$
\left[M_h(x_{t(h)})\right]_{q,\cdot} = m_{h-1}(x_{t(h)}, q)^\top, \quad q \in \mathbb{Q}_h. \tag{24}
$$

That is, $M_h(x_{t(h)})$ is the matrix whose rows are indexed by the core tests at the $h$-th observable step, where the $q \in \mathbb{Q}_h$ row is the weights given by the $m_{h-1}$ mapping for the future consisting of $x_{t(h)}$ followed by $q$. This mapping enables us to update the probabilities of the core test sets.

**Lemma 3.** *For any $h \in [H-1]$, $\tau_h \in \mathbb{H}_h$, $x_{t(h+1)} \in \mathbb{X}_{t(h+1)}$, we have*

$$
\psi_h(\tau_{h-1}, x_{t(h)}) = M_h(x_{t(h)})\psi_{h-1}(\tau_{h-1}). \tag{25}
$$

*Hence, for a history $\tau_h = (x_{t(1)}, \ldots, x_{t(h)}) \in \mathbb{H}_h$, we have*

$$
\psi_h(\tau_h) = M_h(x_{t(h)})M_{h-1}(x_{t(h-1)})\cdots M_1(x_{t(1)})\psi_0, \tag{26}
$$

*where $\psi_0 = \psi_0(\emptyset)$.*

Finally, observe that $\mathbb{Q}_{H-1}^m = \mathbb{X}_{t(H)}$. Hence,

$$
\psi_{H-1}(\tau_{H-1}) = \left(\overline{\mathbb{P}}\left[\tau_{h-1}, x_{t(H)}\right]\right)_{x_{t(H)} \in \mathbb{X}_{t(H)}} \in \mathbb{R}^{|\mathbb{X}_{t(H)}|}.
$$

Thus, letting $\phi_H : \mathbb{X}_{t(H)} \to \mathbb{R}^{|\mathbb{X}_{t(H)}|}$ be $\phi_H(x_{t(H)}) = e_{x_{t(H)}}$ (the canonical basis vector), yields

$$\overline{\mathbb{P}}\left[x_{t(h)} : h \in [H]\right] = \phi_H(x_{t(H)})^\top M_{H-1}(x_{t(H-1)}) \cdots M_1(x_{t(1)}) \psi_0. \tag{27}$$

Hence, Equation (27) together with Equation (26) imply that $(\boldsymbol{M}, \phi_H, \psi_0)$ is a valid generalized PSR representation for the POST/POSG (as per Definition 3).

What remains is to show that if the POST/POSG is $\alpha$-robustly $\mathcal{I}^\dagger$-weakly revealing, then the PSR constructed above is well-conditioned.

**Theorem** (Restatement of Theorem 2). *Suppose a POST/POSG is $\alpha$-robustly $m$-step $\mathcal{I}^\dagger$-weakly revealing. Then, the corresponding generalized PSR as constructed in Section 4 is $\gamma$-well-conditioned with $\gamma = \alpha / \max_h |\mathbb{I}_h^\dagger|^{1/2}$.*

*Proof.* We first show condition (1) in Assumption 1. Suppose $h > H - m$ and hence the core tests are the full futures, which have length smaller than $m$. Then for any $x \in \mathbb{R}^{d_h}$, $d_h = \prod_{s=h}^H |\mathbb{X}_s|$, we have

$$\max_\pi \sum_{\omega_h} \left|m_h(\omega_h)^\top x\right| \cdot \pi(\omega_h) = \max_\pi \sum_{\omega_h} |x[\omega_h]|\, \pi(\omega_h) \le \|x\|_1 ,$$

where $x[\omega_h]$ indexes the component of the vector $x$ corresponding to the future $\omega_h$.

Now suppose $h \le H - m$ (and hence the core tests consist of $m$-step futures). Then, we have,

$$\max_\pi \sum_{\omega_h} \left|m_h(\omega_h)^\top x\right| \pi(\omega_h) = \max_\pi \sum_{\omega_h} \left|m(\omega_h)^\top \boldsymbol{G}_h \boldsymbol{G}_h^\dagger x\right| \cdot \pi(\omega_h)$$

$$\le \max_\pi \sum_{\omega_h} \sum_{i^\dagger \in \mathbb{I}_h^\dagger} \left|m(\omega_h)^\top \boldsymbol{G}_h e_{i^\dagger}\right| \left|e_{i^\dagger}^\top \boldsymbol{G}_h^\dagger x\right| \cdot \pi(\omega_h).$$

Now observe that for any policy $\pi$ and any $i^\dagger \in \mathbb{I}_h^\dagger$, we have

$$\sum_{\omega_h} \left|m(\omega_h)^\top \boldsymbol{G}_h e_{i^\dagger}\right| \cdot \pi(\omega_h) = \sum_{\omega_h} \left|\tilde{m}(\omega_h)^\top \boldsymbol{G}_h^\dagger \boldsymbol{G}_h e_{i^\dagger}\right| \cdot \pi(\omega_h)$$

$$= \sum_{\omega_h} \overline{\mathbb{P}}\left[\omega_h \mid i^\dagger\right] \pi(\omega_h)$$

$$= \sum_{\omega_h} \mathbb{P}^\pi\left[\omega_h \mid i^\dagger\right] = 1,$$

where we used the definition of $m_h(\omega_h) := \tilde{m}_h(\omega_h)^\top \boldsymbol{G}_h^\dagger$, and $[\tilde{m}_h(\omega_h)]_{i^\dagger} := \overline{\mathbb{P}}[\omega_h \mid i^\dagger]$. Recall that $\pi(\omega_h)$ is such that for any fixed sequence of observations $\omega_h^o$, $\sum_{\omega_h^a} \pi(\omega_h^o, \omega_h^a) = 1$.

Putting this observation together with the preceding inequality yields

$$\max_\pi \sum_{\omega_h} \left|m_h(\omega_h)^\top x\right| \pi(\omega_h)$$

$$\le \sum_{i^\dagger \in \mathbb{I}_h^\dagger} \left|e_{i^\dagger}^\top \boldsymbol{G}_h^\dagger x\right|$$

$$= \left\|\boldsymbol{G}_h^\dagger x\right\|_1 \le \left\|\boldsymbol{G}_h^\dagger\right\|_1 \cdot \|x\|_1$$

$$\le \frac{\sqrt{\left|\mathbb{I}_h^\dagger\right|}}{\alpha} \|x\|_1 ,$$

where the final inequality is from the relation between the one-norm and two-norm $\left\|\boldsymbol{G}_h^\dagger\right\|_1 \le \sqrt{\left|\mathbb{I}_h^\dagger\right|} \left\|\boldsymbol{G}_h^\dagger\right\|_2$, and $\left\|\boldsymbol{G}_h^\dagger\right\|_2 \le \frac{1}{\alpha}$, by the assumption on its singular values.

Now we show condition (2) in Assumption 1. For ease of notation, we denote $x_{t(h)}$ by $x_h$. When $h > H - m$, note that $[M_h(x_h)]_{q_{h+1}, q_h} = \mathbf{1}\{q_h = (x_h, q_{h+1})\}$, for all $q_h \in \mathbb{Q}_h, q_{h+1} \in \mathbb{Q}_{h+1}$. Hence, we have

$$\max_\pi \sum_{x_h} \|M_h(x_h)z\|_1 \, \pi(x_h) = \|z\|_1 \,.$$

Now, when $h \leq H - m$, by a similar line of reasoning to the proof for condition (1), we have,

$$\max_\pi \sum_{x_h} \|M_h(x_h)z\|_1 \, \pi(x_h|\tau_{h-1}) \leq \max_\pi \sum_{(x_h, q_{h+1}) \in \mathbb{X}_h \times \mathbb{Q}_{h+1}} \sum_{i^\dagger \in \mathbb{I}_h^\dagger} \left| e_{q_{h+1}}^\top M_h(x_h) \boldsymbol{G}_h e_{i^\dagger} \right| \cdot \left| e_{i^\dagger} \boldsymbol{G}_h^\dagger z \right| \pi(x_h|\tau_{h-1})$$

$$\stackrel{(a)}{=} \max_\pi \sum_{(x_h, q_{h+1}) \in \mathbb{X}_h \times \mathbb{Q}_{h+1}} \sum_{i^\dagger \in \mathbb{I}_h^\dagger} \left| m_h(x_{t(h)}, q_{h+1}) \boldsymbol{G}_h e_{i^\dagger} \right| \cdot \left| e_{i^\dagger} \boldsymbol{G}_h^\dagger z \right| \pi(x_h|\tau_{h-1})$$

$$\stackrel{(b)}{=} \max_\pi \sum_{i^\dagger} \left( \sum_{(x_h, q_{h+1})} \overline{\mathbb{P}} \left[ x_h, q_{h+1} \mid i^\dagger \right] \pi(x_h|\tau_{h-1}) \right) \left| e_{i^\dagger}^\top \boldsymbol{G}_h^\dagger z \right|$$

where step (a) uses the definition of $M_h$ and step (b) uses the definition of $m_h(\omega_h)^\top := \tilde{m}_h(\omega_h)^\top \boldsymbol{G}_h^\dagger$ and $[\tilde{m}_h(\omega_h)]_{i^\dagger} := \overline{\mathbb{P}}[\omega_h \mid i^\dagger]$. Now note that,

$$\sum_{(x_h, q_{h+1})} \overline{\mathbb{P}} \left[ x_h, q_{h+1} \mid i^\dagger \right] \pi(x_h|\tau_{h-1}) = \sum_{x_h} \sum_{\mathsf{act}(q_{h+1})} \sum_{\mathsf{obs}(q_{h+1})} \overline{\mathbb{P}} \left[ x_h, \mathsf{obs}(q_{h+1}) \mid i^\dagger, \mathsf{act}(q_{h+1}) \right] \pi(x_h)$$

$$= \sum_{\mathsf{act}(q_{h+1})} 1$$

$$= \left| \mathbb{Q}_{h+1}^A \right|,$$

where the second line is since for any fixed action sequence, the sum over the probabilities of all observation sequences is 1.

Thus, putting this together, we obtain the following,

$$\max_\pi \sum_{x_h} \|M_h(x_h)z\|_1 \, \pi(x_h|\tau_{h-1}) \leq \left| \mathbb{Q}_{h+1}^A \right| \cdot \left\| \boldsymbol{G}_h^\dagger z \right\|_1$$

$$\leq \frac{\sqrt{\left| \mathbb{I}_h^\dagger \right|} \left| \mathbb{Q}_{h+1}^A \right|}{\alpha} \|z\|_1 \,,$$

where the last line again follows by the assumption on the singular values of $\boldsymbol{G}_h$. $\qquad \square$

## J  Proof of Theorem 4: UCB Algorithm for Generalized PSRs (Team Setting)

In this section, we prove Theorem 4 which states that Algorithm 1 returns a near-optimal policy in a polynomial number of iterations. The proof is adapted from [35] and generalized to our setting with generalized PSRs (Definition 3). The proof is organized into several subsections. In Appendix J.1, we show that the total variation distance between trajectories under the true model and the estimated model can be bounded in terms of the estimation error of the observable operators $\{M_h\}_h$. In Appendix J.2 we state some general results on maximum likelihood estimation which show that the MLE model has small error on the collected dataset. In Appendix J.3 we prove that the bonus term is an upper confidence bound for the total variation distance. In Appendix J.4 we show that the estimation error is sublinear in the number of iterations (i.e., $O(\sqrt{K})$). Finally, in Appendix J.5 we put this all together to prove the theorem.

### J.1  Properties of Generalized PSRs

Recall that a PSR model $\theta = (\boldsymbol{M}, \psi_0, \phi_H)$ consists of operators $\boldsymbol{M} = \{M_h\}_{h=1}^{H-1}$, $M_h : \mathbb{X}_h \to \mathbb{R}^{d_h \times d_{h-1}}$, $\phi_H : \mathbb{X}_H \to \mathbb{R}^{d_{H-1}}$ (assumed to be the identity mapping), and $\psi_0$ (assumed to be known

for the purposes of presentation). Recall that, for any trajectory $\tau_{h-1} = (x_1, \ldots, x_{h-1})$, under model $\theta$, we have

$$
\begin{aligned}
M_h(x_h)\overline{\psi}_{h-1}(\tau_{h-1}) &= \frac{\psi_h(\tau_h)}{\overline{\mathbb{P}}_\theta(\tau_{h-1})} \\
&= \frac{\psi_h(\tau_h)}{\overline{\mathbb{P}}_\theta(x_h \mid \tau_{h-1})\overline{\mathbb{P}}_\theta(\tau_{h-1})}\overline{\mathbb{P}}_\theta(x_h \mid \tau_{h-1}) \\
&= \overline{\psi}_h(\tau_h)\overline{\mathbb{P}}_\theta(x_h \mid \tau_{h-1})
\end{aligned}
\tag{28}
$$

Here, the notation $\overline{\mathbb{P}}_\theta(x_h \mid \tau_{h-1})$ means the probability of $x_h$ conditioned on the history $\tau_{h-1}$, with all actions executed. In particular, if $x_h$ is an action, then $\overline{\mathbb{P}}_\theta(x_h \mid \tau_{h-1}) = 1$ and $M_h(x_h)\overline{\psi}_{h-1}(\tau_{h-1}) = \overline{\psi}_h(\tau_h)$.

The following proposition shows that the total variation distance between the distribution of trajectories of two PSR models can be bounded in terms of the difference in their observable operators.

**Proposition 4.** *For any policy $\pi$ and $\theta, \widehat{\theta} \in \Theta$, we have,*

$$
D_{\text{TV}}\left(\mathbb{P}_{\widehat{\theta}}^\pi, \mathbb{P}_\theta^\pi\right) \leq \sum_{h=1}^H \sum_{\tau_H \in \mathbb{H}_H} \pi(\tau_h)\left|\widehat{m}_h(\omega_h)^\top\left(\widehat{M}_h(x_h) - M_h(x_h)\right)\psi_{h-1}(\tau_{h-1})\right|,
$$

$$
D_{\text{TV}}\left(\mathbb{P}_{\widehat{\theta}}^\pi, \mathbb{P}_\theta^\pi\right) \leq \sum_{h=1}^H \sum_{\tau_H \in \mathbb{H}_H} \pi(\tau_h)\left|m_h(\omega_h)^\top\left(\widehat{M}_h(x_h) - M_h(x_h)\right)\widehat{\psi}_{h-1}(\tau_{h-1})\right|,
$$

*Proof.* The probability of any trajectory $\tau_H = (x_1, \ldots, x_H)$ can be written in terms of products of the observable operators $M_h(x_h)$ of a PSR model (Equation (1)). Hence, we have,

$$
\begin{aligned}
D_{\text{TV}}\left(\mathbb{P}_{\widehat{\theta}}^\pi, \mathbb{P}_\theta^\pi\right) &= \frac{1}{2}\sum_{\tau_H}\left|\mathbb{P}_{\widehat{\theta}}^\pi(\tau_H) - \mathbb{P}_\theta^\pi(\tau_H)\right| \\
&= \frac{1}{2}\sum_{\tau_H}\pi(\tau_H) \cdot \left|\left(\prod_{h=1}^H \widehat{M}_h(x_h)\right)\psi_0 - \left(\prod_{h=1}^H M_h(x_h)\right)\psi_0\right| \\
&\leq \frac{1}{2}\sum_{\tau_H}\pi(\tau_H)\sum_{h=1}^H\left|\widehat{m}_h(x_{h+1:H})^\top\left(\widehat{M}_h(x_h) - M_h(x_h)\right)\psi_{h-1}(\tau_{h-1})\right|,
\end{aligned}
$$

where the second line follows by the triangle inequality after noting that for any trajectory $\tau_H = x_{1:H} \in \mathbb{H}_H$, the following holds for any $h = 1, \ldots, H$,

$$
\left(\prod_{h=1}^H \widehat{M}_h(x_h)\right)\psi_0 - \left(\prod_{h=1}^H M_h(x_h)\right)\psi_0 = \widehat{m}_h(x_{h+1:H})^\top\widehat{M}_h(x_h)\widehat{\psi}_{h-1}(x_{1:h-1}) - m_h(x_{h+1:H})^\top M_h(x_h)\psi_{h-1}(x_{1:h-1}).
$$

By the same argument, we obtain the second inequality,

$$
\begin{aligned}
D_{\text{TV}}\left(\mathbb{P}_{\widehat{\theta}}^\pi, \mathbb{P}_\theta^\pi\right) &= \frac{1}{2}\sum_{\tau_H}\left|\mathbb{P}_{\widehat{\theta}}^\pi(\tau_H) - \mathbb{P}_\theta^\pi(\tau_H)\right| \\
&\leq \frac{1}{2}\sum_{\tau_H}\pi(\tau_H)\sum_{h=1}^H\left|m_h(x_{h+1:H})^\top\left(\widehat{M}_h(x_h) - M_h(x_h)\right)\widehat{\psi}_{h-1}(\tau_{h-1})\right|.
\end{aligned}
$$

$\square$

In this result, recall that we assume $\psi_0$ is known to the agent, to simplify the presentation. If $\psi_0$ was not known, there would be another term due to the estimation as $\widehat{\psi}_0$ [see 33, Lemma C.3]. Note that the sample complexity of estimating $\psi_0$ is small compared to learning the other parameters.

## J.2 General Results on MLE

In this section, we state some general results on maximum likelihood estimation which ultimately guarantee that the estimated model produced by the procedure in Algorithm 1 has a small estimation error. The results are stated without proof. The proofs are given in [35] and use standard techniques on MLE analysis [94]. This ultimately leads us to a lemma which states that the estimation error of the MLE model is small on the collected data.

The first proposition states that the log-likelihood of the true model $\theta^*$ is large compared to any other model.

**Proposition 5** (Proposition 4 of [35]). *Fix $\varepsilon < \frac{1}{KH}$. With probability at least $1 - \delta$, for any $\overline{\theta} \in \overline{\Theta}_\varepsilon$ and any $k \in [K]$, the following holds:*

$$\forall \overline{\theta} \in \overline{\Theta}_\varepsilon, \sum_h \sum_{(\tau_h, \pi) \in \mathcal{D}_h} \log \mathbb{P}^\pi_{\overline{\theta}}(\tau_h) - 3 \log \frac{K \left| \overline{\Theta}_\varepsilon \right|}{\delta} \leq \sum_h \sum_{(\tau_h, \pi) \in \mathcal{D}^k_h} \log \mathbb{P}^\pi_{\theta^*}(\tau_h)$$

$$\forall \overline{\theta} \in \overline{\Theta}_\varepsilon, \sum_{(\tau_H, \pi) \in \mathcal{D}^k} \log \mathbb{P}^\pi_{\overline{\theta}}(\tau_H) - 3 \log \frac{K \left| \overline{\Theta}_\varepsilon \right|}{\delta} \leq \sum_{(\tau_h, \pi) \in \mathcal{D}^k_h} \log \mathbb{P}^\pi_{\theta^*}(\tau_h)$$

The second proposition provides an upper bound on the total variation distance between the distributions of futures given histories on the empirical history of trajectories. This result ensures that the model estimated by Algorithm 1 is accurate on the sampled trajectories.

**Proposition 6** (Proposition 5 in [35]). *Fix $p_{\min}$ and $\varepsilon \leq \frac{p_{\min}}{KH}$. Let $\Theta^k_{\min} = \left\{ \theta : \forall h, (\tau_h, \pi) \in \mathcal{D}^k_h, \mathbb{P}^\pi_\theta(\tau_h) \geq p_{\min} \right\}$. Then, with probability at least $1 - \delta$, for any $k \in [K], \theta \in \Theta^k_{\min}$, we have,*

$$\sum_h \sum_{(\tau_h, \pi) \in \mathcal{D}^k_h} \mathrm{D}^2_{\mathrm{TV}} \left( \mathbb{P}^\pi_\theta(\omega_h | \tau_h), \mathbb{P}^\pi_{\theta^*}(\omega_h | \tau_h) \right) \leq 6 \sum_h \sum_{(\tau_h, \pi) \in \mathcal{D}^k_h} \log \frac{\mathbb{P}^\pi_{\theta^*}(\tau_H)}{\mathbb{P}^\pi_\theta(\tau_H)} + 31 \log \frac{K \left| \overline{\Theta}_\varepsilon \right|}{\delta}.$$

The next proposition is standard in the analysis of maximum likelihood estimation. $\mathrm{D}_{\mathrm{H}}$ denotes the Hellinger distance.

**Proposition 7** (Proposition 6 of [35]). *Let $\varepsilon < \frac{1}{K^2 H^2}$. Then, with probability at least $1 - \delta$, the following holds for all $\theta \in \Theta$ and $k \in [K]$,*

$$\sum_{\pi \in \mathcal{D}^k} \mathrm{D}^2_{\mathrm{H}} \left( \mathbb{P}^\pi_\theta(\tau_H), \mathbb{P}^\pi_{\theta^*}(\tau_H) \right) \leq \frac{1}{2} \sum_{(\tau_H, \pi) \in \mathcal{D}^k} \log \frac{\mathbb{P}^\pi_{\theta^*}(\tau_H)}{\mathbb{P}^\pi_\theta(\tau_H)} + 2 \log \frac{K \left| \overline{\Theta}_\varepsilon \right|}{\delta}.$$

The final proposition of this section states that when $p_{\min}$ is chosen as in Theorem 4, the true model $\theta^*$ lies in the constraint $\Theta^k_{\min}$ with high probability.

**Proposition 8.** *Fix $p_{\min} \leq \frac{\delta}{KH \prod_{h=1}^H |\mathbb{X}_h|}$. Then, with probability at least $1 - \delta$, we have $\theta^* \in \Theta^k_{\min}$ $\forall k$.*

*Proof.* For each $k \in [K]$, we have $\theta^* \in \Theta^k_{\min}$ if $\mathbb{P}^{\pi^k}_{\theta^*}(\tau^k_h) \geq p_{\min}$ for all $h \in [H]$, $(\tau^k_h, \pi^k) \in \mathcal{D}^k_h$. Consider the probability of $\theta^*$ violating this constraint for some trajectory in the dataset. For each $k, h, (\tau^k_h, \pi^k)$, we have

$$\mathbb{P} \left[ \mathbb{P}^{\pi^k}_{\theta^*}(\tau^k_h) < p_{\min} \right] = \mathbb{E}_\pi \left[ \mathbb{P} \left[ \mathbb{P}^{\pi^k}_{\theta^*}(\tau^k_h) < p_{\min} \mid \pi^k = \pi \right] \right]$$

$$= \mathbb{E}_\pi \left[ \sum_{\tau_h \in \mathbb{H}_h} \mathbb{P}^\pi_{\theta^*}(\tau^k_h = \tau_h) \mathbf{1}\{ \mathbb{P}^\pi_{\theta^*}(\tau_h) < p_{\min} \} \right]$$

$$< \sum_{\tau_h \in \mathbb{H}_h} p_{\min}$$

$$= |\mathbb{H}_h| \, p_{\min}$$

$$\leq \frac{\delta}{KH}.$$

In the above, the first line is by the law of total probability, where the expectation is over the policy $\pi^k$ used while collecting the $(h, k)$-th trajectory, and the inner probability is over trajectories $\tau_h^k$. The second line calculates the probability of the event $\{\mathbb{P}_{\theta^*}^{\pi^k}(\tau_h^k) < p_{\min}\}$. Taking a union bound over $k \in [K]$, $h \in [H]$, and $(\tau_h, \pi) \in \mathcal{D}_h$ implies that $\mathbb{P}\left[\theta^* \in \Theta_{\min}^k\right] \geq 1 - \delta$. $\qquad\square$

In what follows, let $\mathcal{E}_\omega, \mathcal{E}_\pi, \mathcal{E}_{\min}$ be the events in Propositions 6 to 8, respectively. Let $\mathcal{E} = \mathcal{E}_\omega \cap \mathcal{E}_\pi \cap \mathcal{E}_{\min}$ be the intersection of all events. Propositions 6 to 8 guarantee the event $\mathcal{E}$ occurs with high probability, $\mathbb{P}[\mathcal{E}] \geq 1 - 3\delta$, by a union bound.

The following result states that the estimated model is accurate on the past exploration policies and dataset of collected trajectories. This holds for both the conditional probabilities of futures given past trajectories in the dataset as well as over full trajectories. The result follows from the MLE analysis in Propositions 6 to 8.

**Lemma 4.** *Let $\beta = 31 \log \frac{K|\overline{\Theta}_\varepsilon|}{\delta}$, and suppose $\varepsilon \leq \frac{\delta}{K^2 H^2 \prod_h |\mathbb{X}_h|}$, where $\overline{\Theta}_\varepsilon$ is the optimistic $\varepsilon$-net in Proposition 3. Then, under event $\mathcal{E}$, the following holds,*

$$\sum_h \sum_{(\tau_h, \pi) \in \mathcal{D}_h^k} D_{\text{TV}}^2 \left(\mathbb{P}_{\widehat{\theta}^k}^\pi(\omega_h | \tau_h), \mathbb{P}_{\theta^*}^\pi(\omega_h | \tau_h)\right) \leq 7\beta, \text{ and}$$

$$\sum_{\pi \in \mathcal{D}^k} D_{\text{H}}^2 \left(\mathbb{P}_{\widehat{\theta}^k}^\pi(\tau_H), \mathbb{P}_{\theta^*}^\pi(\tau_H)\right) \leq 7\beta,$$

*Proof.* The proof follows by Propositions 6 to 8. The argument is direct and is identical to Lemma 1 of [35]. $\qquad\square$

## J.3   UCB for Total Variation Distance

**Notation.** Let $m^*, \{M_h^*\}_h$ be the observable operators of the true PSR $\theta^*$, and let $\{\widehat{M}_h^k\}_h$ be the algorithm's estimates of the observable operators corresponding to $\widehat{\theta}^k$.

Recall that Proposition 4 shows that the total variation distance between the distribution over trajectories of two PSRs is bounded by the estimation error of the observable operators $M_h$. The following result constructs a bound on the estimation error of the observable operators $M_h(x_h)$. The proof is adapted from [35, Lemma 2] to our setting with generalized PSRs.

**Lemma 5.** *Under event $\mathcal{E}$, for any policy $\pi$ and $k \in [K]$, we have,*

$$\sum_{\tau_H} \left| m^\star(\omega_h)^\top \left(\widehat{M}_h^k(x_h) - M_h^\star(x_h)\right) \widehat{\psi}_{h-1}^k(\tau_{h-1}) \right| \pi(\tau_H) \leq \mathbb{E}_{\tau_{h-1} \sim \mathbb{P}_{\widehat{\theta}^k}^\pi} \left[\alpha_{h-1}^k \left\|\widehat{\psi}_{h-1}^k(\tau_{h-1})\right\|_{(\widehat{U}_{h-1}^k)^{-1}}\right]$$

*where,*

$$\widehat{U}_{h-1}^k = \lambda I + \sum_{\tau_{h-1} \in \mathcal{D}_{h-1}^k} \left[\widehat{\psi}_h^k(\tau_{h-1}) \widehat{\psi}_h^k(\tau_{h-1})^\top\right]$$

$$\left(\alpha_{h-1}^k\right)^2 = \frac{4\lambda Q_A^2 d}{\gamma^4} + \frac{4 \max_{s \in \mathcal{A}} |\mathbb{X}_s|^2 Q_A^2}{\gamma^2} \sum_{\tau_{h-1} \in \mathcal{D}_{h-1}^k} D_{\text{TV}}^2 \left(\mathbb{P}_{\widehat{\theta}^k}^{u_{h-1}^{\exp}}\left(\omega_{h-1}^o \mid \tau_{h-1}, \omega_{h-1}^a\right), \mathbb{P}_{\theta^*}^{u_{h-1}^{\exp}}\left(\omega_{h-1}^o \mid \tau_{h-1}, \omega_h^a\right)\right)$$

*Proof.* To ease notation, we index the future trajectories $\omega_{h-1} = (x_h, \ldots, x_H) \in \mathbb{F}_{h-1}$ by $i$ and history trajectories $\tau_{h-1} = (x_1, \ldots, x_{h-1}) \in \mathbb{H}_{h-1}$ by $j$. We denote $m^\star(\omega_h)^\top \left(\widehat{M}_h^k(x_h) - M_h^\star(x_h)\right)$ as $w_i^\top$, $\widehat{\psi}_h^k(\tau_{h-1})$ as $x_j$, and $\pi(\omega_{h-1} | \tau_{h-1})$ as $\pi_{i|j}$.

The following bound follows from the Cauchy-Schwarz inequality,

$$\sum_{\tau_H} \left| m^\star(\omega_h)^\top \left( \widehat{M}_h^k(x_h) - M_h^\star(x_h) \right) \widehat{\psi}_{h-1}^k(\tau_{h-1}) \right| \pi(\tau_H)$$

$$\overset{(a)}{=} \sum_{\omega_{h-1}} \sum_{\tau_{h-1}} \left| m^\star(\omega_h)^\top \left( \widehat{M}_h^k(x_h) - M_h^\star(x_h) \right) \overline{\widehat{\psi}}_h^k(\tau_{h-1}) \right| \pi(\omega_{h-1}|\tau_{h-1}) \mathbb{P}_{\widehat{\theta}^k}^\pi(\tau_{h-1})$$

$$= \sum_i \sum_j \left| w_i^\top x_j \right| \pi_{i|j} \mathbb{P}_{\widehat{\theta}^k}^\pi(j)$$

$$= \sum_i \sum_j \left( \pi_{i|j} \cdot \mathrm{sign}(w_i^\top x_j) w_i \right)^\top x_j \cdot \mathbb{P}_{\widehat{\theta}^k}^\pi(j)$$

$$= \sum_j \left( \sum_i \pi_{i|j} \cdot \mathrm{sign}(w_i^\top x_j) w_i \right)^\top x_j \cdot \mathbb{P}_{\widehat{\theta}^k}^\pi(j)$$

$$= \mathbb{E}_{j \sim \mathbb{P}_{\widehat{\theta}^k}^\pi} \left[ \left( \sum_i \pi_{i|j} \cdot \mathrm{sign}(w_i^\top x_j) w_i \right)^\top x_j \right]$$

$$\overset{(b)}{\leq} \mathbb{E}_{j \sim \mathbb{P}_{\widehat{\theta}^k}^\pi} \left[ \|x_j\|_{(\widehat{U}_{h-1}^k)^{-1}} \left\| \sum_i \pi_{i|j} \cdot \mathrm{sign}(w_i^\top x_j) \cdot w_i \right\|_{\widehat{U}_{h-1}^k} \right].$$

Step (a) follows from the fact that $\widehat{\psi}_{h-1}^k(\tau_{h-1}) = \overline{\widehat{\psi}}_h^k(\tau_{h-1}) \cdot (\widehat{\phi}_{h-1}^k)^\top \widehat{\psi}_{h-1}^k(\tau_{h-1}) = \overline{\widehat{\psi}}_h^k(\tau_{h-1}) \cdot \overline{\mathbb{P}}_{\widehat{\theta}^k}[\tau_{h-1}]$ and $\overline{\mathbb{P}}_{\widehat{\theta}^k}[\tau_{h-1}] \cdot \pi(\tau_H) = \pi(\omega_{h-1}|\tau_{h-1}) \cdot \mathbb{P}_{\widehat{\theta}^k}^\pi(\tau_{h-1})$. Step (b) is the Cauchy-Schwarz inequality.

Fix $\tau_{h-1} = j_0$. Let $I_1 := \left\| \sum_i \pi_{i|j_0} \cdot \mathrm{sign}(w_i^\top x_{j_0}) \cdot w_i \right\|_{\widehat{U}_{h-1}^k}^2$, which we bound next. By the definition of $\widehat{U}_{h-1}^k$, we partition this term into two parts,

$$I_1 = \underbrace{\lambda \left\| \sum_i \pi_{i|j_0} \cdot \mathrm{sign}(w_i^\top x_{j_0}) \cdot w_i \right\|_2^2}_{I_2} + \underbrace{\sum_{j \in D_{h-1}^\tau} \left[ \left( \sum_i \pi_{i|j_0} \cdot \mathrm{sign}(w_i^\top x_{j_0}) \cdot w_i \right)^\top x_j \right]^2}_{I_3}.$$

We bound $I_2$ and $I_3$ separately. By the triangle inequality, $\sqrt{I_2}$ is bound by a sum of two terms,

$$\sqrt{I_2} = \sqrt{\lambda} \max_{z \in \mathbb{R}^{d_{h-1}}: \|z\|_2=1} \left| \sum_i \pi_{i|j_0} \cdot \mathrm{sign}(w_i^\top x_{j_0}) \cdot w_i^\top z \right|$$

$$\overset{(a)}{\leq} \sqrt{\lambda} \max_{\|z\|_2=1} \sum_{\omega_{h-1}} \left| m^\star(\omega_h^\top) \left( \widehat{M}_h^k(x_h) - M_h^\star(x_h) \right) z \right| \pi(\omega_{h-1}|j_0)$$

$$\overset{(b)}{\leq} \sqrt{\lambda} \max_{\|z\|_2=1} \sum_{\omega_{h-1}} \left| m^\star(\omega_h)^\top \widehat{M}_h^k(x_h) z \right| \pi(\omega_{h-1}|j_0)$$

$$+ \sqrt{\lambda} \max_{\|z\|_2=1} \sum_{\omega_{h-1}} \left| m^\star(\omega_h)^\top M_h^\star(x_h) z \right| \pi(\omega_{h-1}|j_0),$$

where step (a) is by the definition of $w_i^\top, \pi_{i|j_0}$ and the triangle inequality, and step (b) is by the triangle inequality.

Consider the first term. It can be bound via the definition of $\gamma$-well-conditioning as follows,

$$\max_{\|z\|_2=1} \sum_{\omega_{h-1}} \left| m^\star(\omega_h)^\top \widehat{M}_h^k(x_h) z \right| \pi(\omega_{h-1}|j_0)$$

$$= \max_{\|z\|_2=1} \sum_{x_h} \left( \sum_{\omega_h} \left| m^\star(\omega_h)^\top \widehat{M}_h^k(x_h) z \right| \pi(\omega_h|j_0, x_h) \right) \pi(x_h|j_0)$$

$$\overset{(a)}{\leq} \frac{1}{\gamma} \max_{\|z\|_2=1} \sum_{x_h} \left\| \widehat{M}_h^k(x_h) z \right\|_1 \pi(x_h|j_0)$$

$$\overset{(b)}{\leq} \frac{1}{\gamma} \max_{\|z\|_2=1} \frac{\left| \mathbb{Q}_{h+1}^A \right| \|z\|_1}{\gamma}$$

$$\overset{(c)}{\leq} \frac{\sqrt{d} Q_A}{\gamma^2}$$

where step (a) is by the first condition in Assumption 1, step (b) is by the second condition of Assumption 1, and step (c) is by the fact that $\max_{z \in \mathbb{R}^{d_{h-1}} : \|z\|_2=1} \|z\|_1 = \sqrt{d_{h-1}} \leq \sqrt{d}$ and $\left| \mathbb{Q}_{h+1}^A \right| \leq Q_A$. In the above, note that we used the $\gamma$-well-conditioning of PSR $\widehat{\theta}^k$ in step (a) and the $\gamma$-well-conditioning of PSR $\theta^*$ in step (b). The second term in $\sqrt{I_2}$ admits an identical bound, simply by using the well-conditioning of the PSR $\theta^*$ in both steps. Hence, we have that

$$I_2 \leq 4 \frac{\lambda d Q_A^2}{\gamma^4}. \tag{29}$$

Now we upper bound $I_3$,

$$I_3 \leq \sum_{\tau_{h-1} \in \mathcal{D}_{h-1}^k} \left( \sum_{\omega_{h-1}} \left| m^\star(\omega_h)^\top \left( \widehat{M}_h^k(x_h) - M_h^\star(x_h) \right) \widehat{\overline{\psi}}^k(\tau_{h-1}) \right| \pi(\omega_{h-1}|j_0) \right)^2$$

$$\leq \sum_{\tau_{h-1} \in \mathcal{D}_{h-1}^k} \left( \underbrace{\sum_{\omega_{h-1}} \left| m^\star(\omega_h)^\top \left( \widehat{M}_h^k(x_h) \widehat{\overline{\psi}}^k(\tau_{h-1}) - M_h^\star(x_h) \overline{\psi}^\star(\tau_{h-1}) \right) \right| \pi(\omega_{h-1}|j_0)}_{I_4} \right.$$

$$\left. + \underbrace{\sum_{\omega_{h-1}} \left| m^\star(\omega_h)^\top M_h^\star(x_h) \left( \widehat{\overline{\psi}}^k(\tau_{h-1}) - \overline{\psi}^\star(\tau_{h-1}) \right) \right| \pi(\omega_{h-1}|j_0)}_{I_5} \right)^2$$

$$=: \sum_{\tau_{h-1} \in \mathcal{D}_{h-1}^k} (I_4 + I_5)^2$$

where the second equality follows from the triangle inequality by adding and subtracting $m^*(\omega_h)^\top M_h^*(x_h) \overline{\psi}^*(\tau_{h-1})$ inside the absolute value. We now bound each of $I_4$ and $I_5$.

$$I_4 := \sum_{\omega_{h-1}} \left| m^\star(\omega_h)^\top \left( \widehat{M}_h^k(x_h) \widehat{\overline{\psi}}^k(\tau_{h-1}) - M_h^\star(x_h) \overline{\psi}^\star(\tau_{h-1}) \right) \right| \pi(\omega_{h-1}|j_0)$$

$$\overset{(a)}{=} \sum_{\omega_{h-1}} \left| m^\star(\omega_h)^\top \left( \overline{\mathbb{P}}_{\widehat{\theta}^k}[x_h \mid \tau_{h-1}] \widehat{\overline{\psi}}_h(\tau_h) - \overline{\mathbb{P}}_{\theta^*}[x_h \mid \tau_{h-1}] \overline{\psi}_h^*(\tau_h) \right) \right| \pi(\omega_{h-1}|j_0)$$

$$\overset{(b)}{=} \sum_{x_h} \left( \sum_{\omega_h} \left| m^\star(\omega_h)^\top \left( \overline{\mathbb{P}}_{\widehat{\theta}^k}[x_h \mid \tau_{h-1}] \widehat{\overline{\psi}}_h(\tau_h) - \overline{\mathbb{P}}_{\theta^*}[x_h \mid \tau_{h-1}] \overline{\psi}_h^*(\tau_h) \right) \right| \pi(\omega_h|j_0, x_h) \right) \pi(x_h|j_0)$$

$$\overset{(c)}{\leq} \frac{1}{\gamma} \sum_{x_h} \left\| \overline{\mathbb{P}}_{\widehat{\theta}^k}[x_h \mid \tau_{h-1}] \widehat{\overline{\psi}}_h(\tau_h) - \overline{\mathbb{P}}_{\theta^*}[x_h \mid \tau_{h-1}] \overline{\psi}_h^*(\tau_h) \right\|_1 \pi(x_h|j_0)$$

$$\overset{(d)}{=} \frac{1}{\gamma} \sum_{x_h} \sum_{q_h \in \mathbb{Q}_h} \left| \overline{\mathbb{P}}_{\widehat{\theta}^k}[x_h, q_h \mid \tau_{h-1}] - \overline{\mathbb{P}}_{\theta^*}[x_h, q_h \mid \tau_{h-1}] \right| \pi(x_h|j_0)$$

where step (a) is by the fact that $M_h(x_h) \overline{\psi}_{h-1}(\tau_{h-1}) = \overline{\mathbb{P}}[x_h \mid \tau_{h-1}] \overline{\psi}(\tau_h)$, as shown in Equation (28), step (b) uses $\omega_{h-1} = (x_h, \omega_h)$ and $\pi(\omega_{h-1}|j_0) = \pi(x_h|j_0)\pi(\omega_h|j_0, x_h)$, step (c) is by Assumption 1, and step (d) follows by the definition $\overline{\psi}_h$, $[\overline{\psi}_h(\tau_h)]_l = \mathbb{P}_\theta[q_h^l \mid \tau_h]$.

Now, we turn to bound the $I_5$ term. We have

$$I_5 = \sum_{\omega_h} \sum_{x_h} \left| m_h^\star(\omega_h)^\top M_h^\star(x_h) \left( \widehat{\overline{\psi}}^k(\tau_{h-1}) - \overline{\psi}^\star(\tau_{h-1}) \right) \right| \pi(\omega_h|j_0, x_h) \pi(x_h|j_0)$$

$$\overset{(a)}{=} \sum_{\omega_{h-1}} \left| m_{h-1}^\star(\omega_{h-1})^\top \left( \widehat{\overline{\psi}}^k(\tau_{h-1}) - \overline{\psi}^\star(\tau_{h-1}) \right) \right| \pi(\omega_{h-1}|j_0)$$

$$\overset{(a)}{\leq} \frac{1}{\gamma} \left\| \widehat{\overline{\psi}}^k(\tau_{h-1}) - \overline{\psi}^\star(\tau_{h-1}) \right\|_1$$

$$= \frac{1}{\gamma} \sum_{q_{h-1} \in \mathbb{Q}_{h-1}} \left| \overline{\mathbb{P}}_{\widehat{\theta}^k}[q_{h-1} \mid \tau_{h-1}] - \overline{\mathbb{P}}_{\theta^*}[q_{h-1} \mid \tau_{h-1}] \right|,$$

where step (a) is since $m_h^*(\omega_h)^\top M_h^*(x_h) = m_{h-1}^*(\omega_{h-1})^\top$, step (b) is by the first condition of Assumption 1, and the final equality is again by the definition of $\overline{\psi}$.

Combining the above, we have that,

$$I_3 \leq \sum_{\tau_{h-1} \in \mathcal{D}_{h-1}^k} (I_4 + I_5)^2$$

$$\leq \sum_{\tau_{h-1} \in \mathcal{D}_{h-1}^k} \left( \frac{1}{\gamma} \sum_{x_h \in \mathbb{X}_h} \sum_{q_h \in \mathbb{Q}_h} \left| \overline{\mathbb{P}}_{\widehat{\theta}^k}[x_h, q_h \mid \tau_{h-1}] - \overline{\mathbb{P}}_{\theta^*}[x_h, q_h \mid \tau_{h-1}] \right| \pi(x_h|\tau_{h-1}) \right.$$

$$\left. + \frac{1}{\gamma} \sum_{q_{h-1} \in \mathbb{Q}_{h-1}} \left| \overline{\mathbb{P}}_{\widehat{\theta}^k}[q_{h-1} \mid \tau_{h-1}] - \overline{\mathbb{P}}_{\theta^*}[q_{h-1} \mid \tau_{h-1}] \right| \right)^2$$

$$\leq \frac{1}{\gamma^2} \cdot \sum_{\tau_{h-1} \in \mathcal{D}_{h-1}^k} \left( \sum_{(x_h, q_h) \in \mathbb{X}_h \times \mathbb{Q}_h} \left| \overline{\mathbb{P}}_{\widehat{\theta}^k}[x_h, q_h \mid \tau_{h-1}] - \overline{\mathbb{P}}_{\theta^*}[x_h, q_h \mid \tau_{h-1}] \right| \pi(x_h|\tau_{h-1}) \right.$$

$$\left. + \sum_{q_{h-1} \in \mathbb{Q}_{h-1}} \left| \overline{\mathbb{P}}_{\widehat{\theta}^k}[q_{h-1} \mid \tau_{h-1}] - \overline{\mathbb{P}}_{\theta^*}[q_{h-1} \mid \tau_{h-1}] \right| \right)^2$$

Now, we decompose the summations above over $\mathbb{X}_h \times \mathbb{Q}_h$ and $\mathbb{Q}_{h-1}$ into separate summations over observation futures and action futures. That is, $(x_h, q_h)$ is decomposed into $(\omega_{h-1}^a, \omega_{h-1}^o)$, where $\omega_{h-1}^a = \mathtt{act}(x_h, q_h)$ and $\omega_{h-1}^o = \mathtt{obs}(x_h, q_h)$, and the summations are over $\omega_{h-1}^a \in \mathtt{act}(\mathbb{X}_h \times \mathbb{Q}_h)$

and $\omega_{h-1}^o \in \text{obs}(\mathbb{X}_h \times \mathbb{Q}_h)$. Similarly, $q_{h-1}$ can be decomposed into $(q_{h-1}^o, q_{h-1}^a) \in \text{obs}(\mathbb{Q}_{h-1}) \times \text{act}(\mathbb{Q}_{h-1})$. Hence, the bound on $I_3$ can be written as,

$$I_3 \leq \frac{1}{\gamma^2} \cdot \sum_{\tau_{h-1} \in \mathcal{D}_{h-1}^k} \left( \sum_{\omega_{h-1}^a} \sum_{\omega_{h-1}^o} \left| \overline{\mathbb{P}}_{\widehat{\theta}^k}\left[\omega_{h-1}^o \mid \tau_{h-1}, \omega_{h-1}^a\right] - \overline{\mathbb{P}}_{\theta^*}\left[\omega_{h-1}^o \mid \tau_{h-1}, \omega_h^a\right] \right| \pi(x_h | \tau_{h-1}) \right.$$

$$\left. + \sum_{q_{h-1}^a} \sum_{q_{h-1}^o} \left| \overline{\mathbb{P}}_{\widehat{\theta}^k}\left[q_{h-1}^o \mid \tau_{h-1}, q_{h-1}^a\right] - \overline{\mathbb{P}}_{\theta^*}\left[q_{h-1}^o \mid \tau_{h-1}, q_{h-1}^a\right] \right| \right)^2$$

$$\leq \frac{1}{\gamma^2} \cdot \sum_{\tau_{h-1} \in \mathcal{D}_{h-1}^k} \left( \sum_{\omega_{h-1}^a \in \mathbb{Q}_{h-1}^{\text{exp}}} \sum_{\omega_{h-1}^o} \left| \overline{\mathbb{P}}_{\widehat{\theta}^k}\left[\omega_{h-1}^o \mid \tau_{h-1}, \omega_{h-1}^a\right] - \overline{\mathbb{P}}_{\theta^*}\left[\omega_{h-1}^o \mid \tau_{h-1}, \omega_{h-1}^a\right] \right| \right)^2$$

$$= \frac{1}{\gamma^2} \left| \mathbb{Q}_{h-1}^{\text{exp}} \right|^2 \cdot \sum_{\tau_{h-1} \in \mathcal{D}_{h-1}^k} \text{D}_{\text{TV}}^2 \left( \mathbb{P}_{\widehat{\theta}^k}^{\mathbf{u}_{h-1}^{\text{exp}}}\left(\omega_{h-1}^o \mid \tau_{h-1}, \omega_{h-1}^a\right), \mathbb{P}_{\theta^*}^{\mathbf{u}_{h-1}^{\text{exp}}}\left(\omega_{h-1}^o \mid \tau_{h-1}, \omega_{h-1}^a\right) \right).$$

Where the second inequality is by the definition of $\mathbb{Q}_{h-1}^{\text{exp}} = \text{act}(\mathbb{X}_h \times \mathbb{Q}_h \cup \mathbb{Q}_{h-1})$. Here, the second summation is over $\omega_{h-1}^o \in \text{obs}(\mathbb{X}_h \times \mathbb{Q}_h \cup \mathbb{Q}_{h-1})$. The final equality uses the fact the under the policy $\mathbf{u}_{h-1}^{\text{exp}}$ the probability of each action sequence $\omega_{h-1}^o$ is $1/\left|\mathbb{Q}_{h-1}^{\text{exp}}\right|$. Note that $\left|\mathbb{Q}_{h-1}^{\text{exp}}\right| \leq |\text{act}(\mathbb{X}_h \times \mathbb{Q}_h)| + |\text{act}(\mathbb{Q}_{h-1})|$, and hence we have $\left|\mathbb{Q}_{h-1}^{\text{exp}}\right| \leq 2 \max_{s \in \mathcal{A}} |\mathbb{X}_s| Q_A$ for all $h$. Hence, we have,

$$I_3 \leq 4 \max_{s \in \mathcal{A}} |\mathbb{X}_s|^2 Q_A^2 \frac{1}{\gamma^2} \sum_{\tau_{h-1} \in \mathcal{D}_{h-1}^k} \text{D}_{\text{TV}}^2 \left( \mathbb{P}_{\widehat{\theta}^k}^{\mathbf{u}_{h-1}^{\text{exp}}}\left(\omega_{h-1}^o \mid \tau_{h-1}, \omega_{h-1}^a\right), \mathbb{P}_{\theta^*}^{\mathbf{u}_{h-1}^{\text{exp}}}\left(\omega_{h-1}^o \mid \tau_{h-1}, \omega_h^a\right) \right).$$

$$(30)$$

Putting this together with the bounds on $I_2$ and $I_3$, we get that,

$$I_1 \leq \frac{4\lambda Q_A^2 d}{\gamma^4} + 4 \max_{s \in \mathcal{A}} |\mathbb{X}_s|^2 Q_A^2 \frac{1}{\gamma^2} \sum_{\tau_{h-1} \in \mathcal{D}_{h-1}^k} \text{D}_{\text{TV}}^2 \left( \mathbb{P}_{\widehat{\theta}^k}^{\mathbf{u}_{h-1}^{\text{exp}}}\left(\omega_{h-1}^o \mid \tau_{h-1}, \omega_{h-1}^a\right), \mathbb{P}_{\theta^*}^{\mathbf{u}_{h-1}^{\text{exp}}}\left(\omega_{h-1}^o \mid \tau_{h-1}, \omega_h^a\right) \right)$$

$$=: \left(\alpha_{h-1}^k\right)^2,$$

completing the proof. $\qquad \square$

Using the above bound on the difference between the observable operators of the true model and the estimated model, we now bound the total variation distance between the distributions of trajectories through Proposition 4.

**Lemma 6.** *Under even $\mathcal{E}$, the total variation distance between the estimated model at iteration $k$, $\widehat{\theta}^k$, and the true model $\theta^*$, is bounded by,*

$$\text{D}_{\text{TV}}\left(\mathbb{P}_{\widehat{\theta}^k}^\pi(\tau_H), \mathbb{P}_{\theta^*}^\pi(\tau_H)\right) \leq \alpha \cdot \mathbb{E}_{\tau_H \sim \mathbb{P}_{\widehat{\theta}^k}^\pi}\left[ \sqrt{\sum_{h=0}^{H-1} \left\| \widehat{\widetilde{\psi}}^k(\tau_h) \right\|_{(\widehat{U}_h^k)^{-1}}^2} \right], \qquad (31)$$

*for any policy $\pi$, where*

$$\alpha^2 = \frac{4\lambda H Q_A^2 d}{\gamma^4} + 28 \max_{s \in \mathcal{A}} |\mathbb{X}_s|^2 Q_A^2 \frac{1}{\gamma^2} \beta$$

*Proof.* Consider $\alpha_{h-1}^k$ in the previous lemma. We have that,

$$\sum_{h=1}^H \left(\alpha_{h-1}^k\right)^2$$

$$= \frac{4\lambda H Q_A^2 d}{\gamma^4} + 4 \max_{s \in \mathcal{A}} |\mathbb{X}_s|^2 Q_A^2 \frac{1}{\gamma^2} \sum_{h=1}^H \sum_{\tau_{h-1} \in \mathcal{D}_{h-1}^k} \text{D}_{\text{TV}}^2 \left( \mathbb{P}_{\widehat{\theta}^k}^{\mathbf{u}_{h-1}^{\text{exp}}}\left(\omega_{h-1}^o \mid \tau_{h-1}, \omega_{h-1}^a\right), \mathbb{P}_{\theta^*}^{\mathbf{u}_{h-1}^{\text{exp}}}\left(\omega_{h-1}^o \mid \tau_{h-1}, \omega_h^a\right) \right)$$

$$\leq \frac{4\lambda H Q_A^2 d}{\gamma^4} + 4 \max_{s \in \mathcal{A}} |\mathbb{X}_s|^2 Q_A^2 \frac{1}{\gamma^2} 7\beta =: \alpha^2,$$

where the inequality is by the bound on the total variation distance established in Lemma 4.

Now, by Proposition 4, the total variation distance is bounded by the estimation error:

$$
\begin{aligned}
& D_{\text{TV}}\left(\mathbb{P}^\pi_{\widehat{\theta}^k}(\tau_H), \mathbb{P}^\pi_{\theta^*}(\tau_H)\right) \\
& \overset{(a)}{\leq} \sum_{h=1}^{H} \sum_{\tau_H} \left| m^\star(\omega_h)^\top \left(\widehat{M}^k_h(x_h) - M^\star_h(x_h)\right) \widehat{\psi}^k_{h-1}(\tau_{h-1}) \right| \pi(\tau_H) \\
& \overset{(b)}{\leq} \sum_{h=1}^{H} \mathbb{E}^\pi_{\tau_{h-1} \sim \mathbb{P}_{\widehat{\theta}^k}} \left[ \alpha^k_{h-1} \left\| \widehat{\overline{\psi}}^k_{h-1}(\tau_{h-1}) \right\|_{(\widehat{U}^k_{h-1})^{-1}} \right] \\
& \overset{(c)}{\leq} \alpha \cdot \mathbb{E}_{\tau_H \sim \mathbb{P}^\pi_{\widehat{\theta}^k}} \left[ \sqrt{\sum_{h=0}^{H-1} \left\| \widehat{\overline{\psi}}^k(\tau_h) \right\|^2_{(\widehat{U}^k_h)^{-1}}} \right],
\end{aligned}
$$

where step (a) is by Proposition 4, step (b) is by Lemma 5, and step (c) is by the Cauchy-Schwarz inequality and the calculation above bounding $\sum_h \left(\alpha^k_{h-1}\right)^2$. $\qquad \square$

A direct corollary is the following bound on the error in the estimated value function, which establishes that the bonus term $\widehat{b}^k$ gives an upper confidence bound.

**Corollary 4** (Upper confidence bound). *Under the event $\mathcal{E}$, for any $k \in [K]$, any reward function $R : \prod_{h \in [H]} \mathbb{X}_h \to [0, 1]$, and any policy $\pi$, we have,*

$$
\left| V^R_{\widehat{\theta}^k}(\pi) - V^R_{\theta^*}(\pi) \right| \leq V^{\widehat{b}^k}_{\widehat{\theta}^k},
$$

*where $\widehat{b}^k(\tau_H) = \min\left\{ \alpha \sqrt{\sum_h \left\| \widehat{\overline{\psi}}^k(\tau_h) \right\|^2_{(\widehat{U}^k_h)^{-1}}}, 1 \right\}$.*

*Proof.* By a direct calculation,

$$
\begin{aligned}
\left| V^R_{\widehat{\theta}^k}(\pi) - V^R_{\theta^*}(\pi) \right| &= \left| \sum_{\tau_H} R(\tau_H) \mathbb{P}^\pi_{\widehat{\theta}^k}(\tau_H) - \sum_{\tau_H} R(\tau_H) \mathbb{P}^\pi_{\theta^*}(\tau_H) \right| \\
& \overset{(a)}{\leq} \sum_{\tau_H} \left| \mathbb{P}^\pi_{\widehat{\theta}^k}(\tau_H) - \mathbb{P}^\pi_{\theta^*}(\tau_H) \right| \\
& = D_{\text{TV}}\left(\mathbb{P}^\pi_{\widehat{\theta}^k}(\tau_H), \mathbb{P}^\pi_{\theta^*}(\tau_H)\right) \\
& \overset{(b)}{\leq} \alpha \cdot \mathbb{E}_{\tau_H \sim \mathbb{P}^\pi_{\widehat{\theta}^k}} \left[ \sqrt{\sum_{h=0}^{H-1} \left\| \widehat{\overline{\psi}}^k(\tau_h) \right\|^2_{(\widehat{U}^k_h)^{-1}}} \right] \\
& \overset{(c)}{\leq} \alpha \sum_{\tau_H} \widehat{b}^k(\tau_H) \mathbb{P}^\pi_{\widehat{\theta}^k}(\tau_H) \\
& =: V^{\widehat{b}^k}_{\widehat{\theta}^k}
\end{aligned}
$$

where step (a) is by the triangle inequality and the fact that $R(\tau_H) \in [0, 1]$, step (b) is by Lemma 6, and step (c) is by the definition of $\widehat{b}^k$. $\qquad \square$

## J.4 $\sum_{k=1}^{K} V^{\widehat{b}^k}_{\widehat{\theta}^k}$ is sublinear

The next step is to prove that $\sum_{k=1}^{K} V^{\widehat{b}^k}_{\widehat{\theta}^k} = O(\sqrt{K})$. To do that, we first prove that the estimated prediction features and the ground-truth prediction features can be related through the total-variation distance between the estimated model and the true model.

**Lemma 7.** *Under event $\mathcal{E}$, for any $k \in [K]$, we have:*

$$\mathbb{E}_{\tau_H \sim \mathbb{P}_{\theta^*}^{\pi}} \left[ \sqrt{\sum_{h=0}^{H-1} \left\| \widehat{\overline{\psi^k}}(\tau_h) \right\|_{(\widehat{U}_h^k)^{-1}}^2} \right]$$

$$\leq \frac{2HQ_A}{\sqrt{\lambda}} \mathsf{D}_{\mathrm{TV}} \left( \mathbb{P}_{\theta^*}^{\pi}(\tau_h), \mathbb{P}_{\widehat{\theta}^k}^{\pi}(\tau_h) \right) + \left( 1 + \frac{2 \max_{s \in \mathcal{A}} |\mathbb{X}_s| Q_A \sqrt{7r}\beta}{\sqrt{\lambda}} \right) \sum_{h=0}^{H-1} \mathbb{E}_{\tau_h \sim \mathbb{P}_{\theta^*}^{\pi}} \left\| \overline{\psi^*}(\tau_h) \right\|_{(U_h^k)^{-1}}$$

*Proof.* First, we recall the definition of $\widehat{U}_h^k$, and we define its ground-truth counterpart replacing estimated features with true features,

$$\widehat{U}_h^k = \lambda I + \sum_{\tau \in \mathcal{D}_h^k} \widehat{\overline{\psi^k}}(\tau_h) \widehat{\overline{\psi^k}}(\tau_h)^\top,$$

$$U_h^k = \lambda I + \sum_{\tau \in \mathcal{D}_h^k} \overline{\psi^*}(\tau_h) \overline{\psi^*}(\tau_h)^\top.$$

For any trajectory $\tau_H \in \mathbb{H}_H$, we have,

$$\sqrt{\sum_{h=0}^{H-1} \left\| \widehat{\overline{\psi^k}}(\tau_h) \right\|_{(\widehat{U}_h^k)^{-1}}^2} \overset{(a)}{\leq} \sum_{h=0}^{H-1} \left\| \widehat{\overline{\psi^k}}(\tau_h) \right\|_{(\widehat{U}_h^k)^{-1}}$$

$$\leq \frac{1}{\sqrt{\lambda}} \sum_{h=0}^{H-1} \left\| \widehat{\overline{\psi^k}}(\tau_h) - \overline{\psi^*}(\tau_h) \right\|_2 + \sum_{h=0}^{H-1} \left( 1 + \frac{\sqrt{r} \sqrt{\sum_{\tau_h \in \mathcal{D}_h^k} \left\| \widehat{\overline{\psi^k}}(\tau_h) - \overline{\psi^*}(\tau_h) \right\|_2^2}}{\sqrt{\lambda}} \right) \left\| \overline{\psi^*}(\tau_h) \right\|_{(U_h^k)}$$

where step (a) is simply using $\|x\|_2 \leq \|x\|_1$ and step (b) is by the identity [35, Lemma 13]. Note that $r$ is the rank of the PSR and $r \geq \mathrm{rank}(\{ \widehat{\overline{\psi^k}}(\tau_h) : \tau_h \in \mathbb{H}_h \}), \mathrm{rank}(\{ \overline{\psi^*}(\tau_h) : \tau_h \in \mathbb{H}_h \})$.
Moreover, we have,

$$\left\| \widehat{\overline{\psi^k}}(\tau_h) - \overline{\psi^*}(\tau_h) \right\|_2 \leq \left\| \widehat{\overline{\psi^k}}(\tau_h) - \overline{\psi^*}(\tau_h) \right\|_1$$

$$\overset{(a)}{=} \sum_{q_h \in \mathbb{Q}_h} \left| \overline{\mathbb{P}}_{\widehat{\theta}^k} \left[ q_h^o \mid \tau_h, q_h^a \right] - \overline{\mathbb{P}}_{\theta^*} \left[ q_h^o \mid \tau_h, q_h^a \right] \right|$$

$$\overset{(b)}{\leq} 2 \max_{s \in \mathcal{A}} |\mathbb{X}_s| Q_A \mathsf{D}_{\mathrm{TV}} \left( \mathbb{P}_{\widehat{\theta}^k}^{\mathsf{u}_{h-1}^{\exp}}(\cdot | \tau_h), \mathbb{P}_{\theta^*}^{\mathsf{u}_{h-1}^{\exp}}(\cdot | \tau_h) \right),$$

where we used the definition of $\overline{\psi}$ in (a) and the definition of the $\mathsf{u}_{h-1}^{\exp}$ in (b).
Now, by Lemma 4, we have,

$$\sqrt{\sum_{h=0}^{H-1} \left\| \widehat{\overline{\psi^k}}(\tau_h) \right\|_{(\widehat{U}_h^k)^{-1}}^2} \leq \frac{1}{\sqrt{\lambda}} \sum_{h=0}^{H-1} \left\| \widehat{\overline{\psi^k}}(\tau_h) - \overline{\psi^*}(\tau_h) \right\|_2 + \sum_{h=0}^{H-1} \left( 1 + \frac{\sqrt{r} \sqrt{\sum_{\tau_h \in \mathcal{D}_h^k} \left\| \widehat{\overline{\psi^k}}(\tau_h) - \overline{\psi^*}(\tau_h) \right\|_2^2}}{\sqrt{\lambda}} \right) \left\| \overline{\psi^*}(\tau_h) \right\|_{(U_h^k)}$$

$$\leq \frac{1}{\sqrt{\lambda}} \sum_{h=0}^{H-1} \left\| \widehat{\overline{\psi^k}}(\tau_h) - \overline{\psi^*}(\tau_h) \right\|_2 + \left( 1 + \frac{2 \max_{s \in \mathcal{A}} |\mathbb{X}_s| Q_A \sqrt{7r}\beta}{\sqrt{\lambda}} \right) \sum_{h=0}^{H-1} \left\| \overline{\psi^*}(\tau_h) \right\|_{(U_h^k)^{-1}},$$

where the first line is combining the calculations above and the second line is by the estimation guarantee of Lemma 4.

The first term can be bounded in expectation under $\mathbb{P}_{\theta^*}^\pi$ for any $\pi$ as follows,

$$\sum_{h=0}^{H-1} \mathbb{E}_{\tau_h \sim \mathbb{P}_{\theta^*}^\pi} \left[ \left\| \widehat{\psi^k}(\tau_h) - \overline{\psi^*}(\tau_h) \right\|_2 \right] \leq \sum_{h=0}^{H-1} \mathbb{E}_{\tau_h \sim \mathbb{P}_{\theta^*}^\pi} \left[ \left\| \widehat{\psi^k}(\tau_h) - \overline{\psi^*}(\tau_h) \right\|_1 \right]$$

$$\leq \sum_{h=0}^{H-1} \sum_{\tau_h} \left\| \widehat{\psi^k}(\tau_h) \left( \mathbb{P}_{\theta^*}^\pi(\tau_h) - \mathbb{P}_{\widehat{\theta}^k}^\pi(\tau_h) \right) + \overline{\widehat{\psi^k}}(\tau_h) \mathbb{P}_{\widehat{\theta}^k}^\pi(\tau_h) - \overline{\psi^*}(\tau_h) \mathbb{P}_{\theta^*}^\pi(\tau_h) \right\|_1$$

$$\overset{(a)}{\leq} \sum_{h=0}^{H-1} \sum_{\tau_h} \left\| \widehat{\psi^k}(\tau_h) \right\|_1 \left| \mathbb{P}_{\theta^*}^\pi(\tau_h) - \mathbb{P}_{\widehat{\theta}^k}^\pi(\tau_h) \right| + \left\| \overline{\widehat{\psi^k}}(\tau_h) \mathbb{P}_{\widehat{\theta}^k}^\pi(\tau_h) - \overline{\psi^*}(\tau_h) \mathbb{P}_{\theta^*}^\pi(\tau_h) \right\|_1$$

$$\overset{(b)}{\leq} \sum_{h=0}^{H-1} \sum_{\tau_h} \left( \left\| \widehat{\psi^k}(\tau_h) \right\|_1 \left| \mathbb{P}_{\theta^*}^\pi(\tau_h) - \mathbb{P}_{\widehat{\theta}^k}^\pi(\tau_h) \right| + \left\| \widehat{\psi^k}(\tau_h) - \psi^*(\tau_h) \right\|_1 \pi(\tau_h) \right)$$

$$\overset{(c)}{\leq} 2 Q_A \sum_{h=0}^{H-1} \mathsf{D}_{\mathsf{TV}} \left( \mathbb{P}_{\theta^*}^\pi(\tau_h), \mathbb{P}_{\widehat{\theta}^k}^\pi(\tau_h) \right)$$

$$\overset{(d)}{\leq} 2 H Q_A \mathsf{D}_{\mathsf{TV}} \left( \mathbb{P}_{\theta^*}^\pi(\tau_h), \mathbb{P}_{\widehat{\theta}^k}^\pi(\tau_h) \right),$$

where step (a) is the triangle inequality, step (b) is the definition of $\overline{\psi}(\tau_h)$, step (c) is since $\left\| \widehat{\psi^k}(\tau_h) \right\|_1 \leq \left| \mathbb{Q}_h^A \right| \leq Q_A$ for any $\tau_h$ and the definition of $\psi(\tau_h)$, and step (d) is simply $\mathsf{D}_{\mathsf{TV}} \left( \mathbb{P}_{\theta^*}^\pi(\tau_h), \mathbb{P}_{\widehat{\theta}^k}^\pi(\tau_h) \right) \geq \mathsf{D}_{\mathsf{TV}} \left( \mathbb{P}_{\theta^*}^\pi(\tau_h), \mathbb{P}_{\widehat{\theta}^k}^\pi(\tau_h) \right)$.

Putting this together concludes the proof,

$$\mathbb{E}_{\tau_H \sim \mathbb{P}_{\theta^*}^\pi} \left[ \sqrt{\sum_{h=0}^{H-1} \left\| \widehat{\psi^k}(\tau_h) \right\|_{(\widehat{U}_h^k)^{-1}}^2} \right]$$

$$\leq \frac{2 H Q_A}{\sqrt{\lambda}} \mathsf{D}_{\mathsf{TV}} \left( \mathbb{P}_{\theta^*}^\pi(\tau_h), \mathbb{P}_{\widehat{\theta}^k}^\pi(\tau_h) \right) + \left( 1 + \frac{2 \max_{s \in \mathcal{A}} |\mathbb{X}_s| Q_A \sqrt{7 r \beta}}{\sqrt{\lambda}} \right) \sum_{h=0}^{H-1} \mathbb{E}_{\tau_h \sim \mathbb{P}_{\theta^*}^\pi} \left\| \overline{\psi^*}(\tau_h) \right\|_{(U_h^k)^{-1}}.$$

$\square$

The following lemma bounds the cumulative estimation error of the probability of trajectories. It can be proved via an $\ell_2$ Eluder argument [38, 76, 77]. A significant portion of the proof is very similar to that of Proposition 4, involving an exchange of $\widehat{(\cdot)}$ and $(\cdot)^*$. We include the proof for completeness.

**Lemma 8.** *Under event $\mathcal{E}$, for any $h \in \{0, \ldots, H-1\}$, we have*

$$\sum_k \mathsf{D}_{\mathsf{TV}} \left( \mathbb{P}_{\theta^*}^{\pi^k}(\tau_H), \mathbb{P}_{\widehat{\theta}^k}^{\pi^k}(\tau_H) \right) \lesssim \frac{\max_{s \in \mathcal{A}} |\mathbb{X}_s| Q_A \sqrt{\beta}}{\gamma} \sqrt{r H K \log \left( 1 + \frac{d Q_A K}{\gamma^4} \right)}.$$

*Here, $a \lesssim b$ indicates that there is an absolute positive constant $c$ s.t. $a \leq c \cdot b$.*

*Proof.* Recall that, by the first inequality in Proposition 4, we have:

$$\mathsf{D}_{\mathsf{TV}} \left( \mathbb{P}_{\theta^*}^{\pi^k}(\tau_H), \mathbb{P}_{\widehat{\theta}^k}^{\pi^k}(\tau_H) \right) \leq \sum_{h=1}^{H} \sum_{\tau_H} \left| \widehat{m}^k(\omega_h)^\top \left( \widehat{M}_h^k(x_h) - M_h^\star(x_h) \right) \psi^\star(\tau_{h-1}) \right| \pi^k(\tau_H)$$

This is very similar to the inequality in Lemma 5, with the difference being that the quantities associated with the estimated model and the true model are exchanged. Since both correspond to a PSR, the analysis follows a similar series of steps. We will use analogous notation to Lemma 5. We index the future trajectory $\omega_{h-1} = (x_h, \ldots, x_H)$ by $i$ and history trajectory $\tau_{h-1} = (x_1, \ldots, x_{h-1})$ by $j$. We denote $\widehat{m}^k(\omega_h)^\top \left( \widehat{M}_h^k(x_h) - M_h^\star(x_h) \right)$ as $w_i$, $\overline{\psi}^\star(\tau_{h-1})$ as $x_j$, and $\pi(\omega_{h-1} | \tau_{h-1})$ as $\pi_{i|j}$.

Define the matrix,

$$\Lambda_h^k = \lambda_0 I + \sum_{t<k} \mathbb{E}_{j\sim \mathbb{P}_{\theta^\star}^{\pi t}}\left[ x_j x_j^\top \right]$$

where $\lambda_0$ is a constant to be determined later.

For any policy $\pi$, using a similar calculation as in Lemma 5, we have,

$$\sum_{\tau_H} \left| \widehat{m}^k(\omega_h)^\top \left( \widehat{M}_h^k(x_h) - M_h^\star(x_h) \right) \psi^\star(\tau_{h-1}) \right| \pi^k(\tau_H)$$

$$= \mathbb{E}_{j\sim \mathbb{P}_{\theta^\star}^{\pi k}}\left[ \sum_i \pi_{i|j} \left| w_i^\top x_j \right| \right]$$

$$= \mathbb{E}_{j\sim \mathbb{P}_{\theta^\star}^{\pi k}}\left[ \left( \sum_i \pi_{i|j}\mathrm{sign}(w_i^\top x_j)w_i \right)^\top x_j \right]$$

$$\leq \mathbb{E}_{j\sim \mathbb{P}_{\theta^\star}^{\pi k}}\left[ \|x_j\|_{\Lambda_h^\dagger} \left\| \sum_i \pi_{i|j}\mathrm{sign}(w_i^\top x_j)w_i \right\|_{\Lambda_h} \right]$$

where the last line is the Cauchy-Schwarz inequality.

Fix $j = j_0$ and consider the term: $\left\| \sum_i \pi_{i|j_0}\mathrm{sign}(w_i^\top x_{j_0})w_i \right\|_{\Lambda_h}$ in the above. This term can be partitioned in the same manner as in Lemma 5 by simply using the definition of $\Lambda_h$ and expanding,

$$\left\| \sum_i \pi_{i|j_0}\mathrm{sign}(w_i^\top x_{j_0})w_i \right\|_{\Lambda_h}^2$$

$$= \underbrace{\lambda_0 \left\| \sum_i \pi_{i|j_0}\cdot \mathrm{sign}(w_i^\top x_{j_0})\cdot w_i \right\|_2^2}_{I_1} + \underbrace{\sum_{t<k} \mathbb{E}_{j\sim \mathbb{P}_{\theta^\star}^{\pi k}}\left[ \left( \sum_i \pi_{i|j_0}\cdot \mathrm{sign}(w_i^\top x_{j_0})\cdot w_i^\top x_j \right)^2 \right]}_{I_2}.$$

We bound each term separately. The process is nearly identical to the proof of Lemma 5, but we show it for completeness.

$\sqrt{I_1}$ is bounded by the sum of two terms,

$$\sqrt{I_1} = \sqrt{\lambda_0} \max_{z\in\mathbb{R}^{d_{h-1}}:\|z\|_2=1} \left| \sum_i \pi_{i|j_0}\cdot \mathrm{sign}(w_i^\top x_{j_0})\cdot w_i^\top z \right|$$

$$\overset{(a)}{\leq} \sqrt{\lambda_0} \max_{\|z\|_2=1} \sum_{\omega_{h-1}} \left| \widehat{m}^k(\omega_h)^\top \left( \widehat{M}_h^k(x_h) - M_h^\star(x_h) \right) z \right| \pi(\omega_{h-1}|j_0)$$

$$\overset{(b)}{\leq} \sqrt{\lambda_0} \max_{\|z\|_2=1} \sum_{\omega_{h-1}} \left| \widehat{m}^k(\omega_h)^\top \widehat{M}_h^k(x_h) z \right| \pi(\omega_{h-1}|j_0)$$

$$+ \sqrt{\lambda_0} \max_{\|z\|_2=1} \sum_{\omega_{h-1}} \left| \widehat{m}^k(\omega_h)^\top M_h^\star(x_h) z \right| \pi(\omega_{h-1}|j_0),$$

where step (a) is the definition of $w_i, \pi_{i|j_0}$, and the triangle inequality, and step (b) is the triangle inequality.

Both terms can be bounded by the $\gamma$-well-conditioning assumption on $\widehat{\theta}^k$ and $\theta^*$. Consider the first term,

$$\max_{\|z\|_2=1} \sum_{\omega_{h-1}} \left| \widehat{m}^k(\omega_h)^\top \widehat{M}_h^k(x_h)z \right| \pi(\omega_{h-1}|j_0)$$

$$= \max_{\|z\|_2=1} \sum_{x_h} \left( \sum_{\omega_h} \left| \widehat{m}^k(\omega_h)^\top \widehat{M}_h^k(x_h)z \right| \pi(\omega_h|j_0, x_h) \right) \pi(x_h|j_0)$$

$$\overset{(a)}{\leq} \max_{\|z\|_2=1} \sum_{x_h} \frac{1}{\gamma} \left\| \widehat{M}_h^k(x_h)z \right\|_1 \pi(x_h|j_0)$$

$$\overset{(b)}{\leq} \frac{1}{\gamma} \max_{\|z\|_2=1} \frac{\left| \mathbb{Q}_{h+1}^A \right| \|z\|_1}{\gamma}$$

$$\overset{(c)}{\leq} \frac{\sqrt{d}Q_A}{\gamma^2}$$

where step (a) is by the first condition in Assumption 1, step (b) is by the second condition of Assumption 1, and step (c) is by the fact that $\max_{z\in\mathbb{R}^{d_{h-1}}:\|z\|_2=1} \|z\|_1 = \sqrt{d_{h-1}} \leq \sqrt{d}$ and $\left| \mathbb{Q}_{h+1}^A \right| \leq Q_A$. In the above, note that we used the $\gamma$-well-conditioning of PSR $\widehat{\theta}^k$ in both step (a) and step (b). The second term in $\sqrt{I_1}$ admits an identical bound, simply by using the well-conditioning of the PSR $\widehat{\theta}^k$ in the first step and $\theta^*$ in the second step. Hence, we have that

$$I_1 \leq 4\frac{\lambda_0 dQ_A^2}{\gamma^4}. \tag{32}$$

Now, we consider the term $I_2$

$$I_2 \leq \sum_{t<k} \mathbb{E}_{\tau_{h-1}\sim\mathbb{P}_{\theta^\star}^{\pi^k}} \left[ \left( \sum_{\omega_{h-1}} \left| \widehat{m}^k(\omega_h)^\top \left( \widehat{M}_h^k(x_h) - M_h^\star(x_h) \right) \overline{\psi}^*(\tau_{h-1}) \right| \pi(\omega_{h-1}|j_0) \right)^2 \right]$$

$$\leq \sum_{t<k} \mathbb{E}_{j\sim\mathbb{P}_{\theta^\star}^{\pi^k}} \left[ \left( \underbrace{\sum_{\omega_{h-1}} \left| \widehat{m}^k(\omega_h)^\top \widehat{M}_h(x_h) \left( \overline{\psi}^\star(\tau_{h-1}) - \widehat{\overline{\psi}}^k(\tau_{h-1}) \right) \right| \pi(\omega_{h-1}|j_0)}_{I_3} \right. \right.$$

$$\left. \left. + \underbrace{\sum_{\omega_{h-1}} \left| \widehat{m}^k(\omega_h)^\top \left( \widehat{M}_h^k(x_h)\widehat{\overline{\psi}}^k(\tau_{h-1}) - M_h^\star(x_h)\overline{\psi}^\star(\tau_{h-1}) \right) \right| \pi(\omega_{h-1}|j_0)}_{I_4} \right)^2 \right]$$

$$=: \sum_{t<k} \mathbb{E}_{j\sim\mathbb{P}_{\theta^\star}^{\pi^k}} (I_3 + I_4)^2$$

where the line follows by the fact that $x \leq |x|$ and the line follows from the triangle inequality by adding and subtracting $\widehat{m}_h(\omega_h)\widehat{M}_h(x_h)\widehat{\overline{\psi}}^k(\tau_{h-1})$ inside the absolute value. We now bound each of $I_3$ and $I_4$.

First, we bound $I_3$ as follows,

$$I_3 = \sum_{\omega_{h-1}} \left| \widehat{m}_h^k(\omega_h)^\top \widehat{M}_h(x_h) \left( \overline{\psi}^\star(\tau_{h-1}) - \widehat{\overline{\psi}}^k(\tau_{h-1}) \right) \right| \pi(\omega_{h-1}|j_0)$$

$$\overset{(a)}{=} \sum_{\omega_{h-1}} \left| \widehat{m}_{h-1}^k(\omega_{h-1})^\top \left( \overline{\psi}^\star(\tau_{h-1}) - \widehat{\overline{\psi}}^k(\tau_{h-1}) \right) \right| \pi(\omega_{h-1}|j_0)$$

$$\overset{(b)}{\leq} \frac{1}{\gamma} \left\| \widehat{\overline{\psi}}^k(\tau_{h-1}) - \overline{\psi}^\star(\tau_{h-1}) \right\|_1$$

$$= \frac{1}{\gamma} \sum_{q_{h-1}\in\mathbb{Q}_{h-1}} \left| \overline{\mathbb{P}}_{\widehat{\theta}^k}[q_{h-1} \mid \tau_{h-1}] - \overline{\mathbb{P}}_{\theta^*}[q_{h-1} \mid \tau_{h-1}] \right|,$$

where step (a) is since $\widehat{m}(\omega_h)^\top \widehat{M}_h(x_h) = \widehat{m}(\omega_{h-1})^\top$, step (b) is by Assumption 1, and the final equality is by the definition of $\overline{\psi}$.

$$
I_4 = \sum_{\omega_{h-1}} \left| \widehat{m}^k(\omega_h)^\top \left( \widehat{M}_h^k(x_h)\widehat{\overline{\psi}}^k(\tau_{h-1}) - M_h^\star(x_h)\overline{\psi}^\star(\tau_{h-1}) \right) \right| \pi(\omega_{h-1}|j_0)
$$

$$
\overset{(a)}{=} \sum_{\omega_{h-1}} \left| \widehat{m}^k(\omega_h)^\top \left( \overline{\mathbb{P}}_{\widehat{\theta}^k}[x_h \mid \tau_{h-1}]\widehat{\overline{\psi}}_h(\tau_h) - \overline{\mathbb{P}}_{\theta^*}[x_h \mid \tau_{h-1}]\overline{\psi}_h^*(\tau_h) \right) \right| \pi(\omega_{h-1}|j_0)
$$

$$
= \sum_{x_h} \left( \sum_{\omega_h} \left| \widehat{m}^k(\omega_h)^\top \left( \overline{\mathbb{P}}_{\widehat{\theta}^k}[x_h \mid \tau_{h-1}]\widehat{\overline{\psi}}_h(\tau_h) - \overline{\mathbb{P}}_{\theta^*}[x_h \mid \tau_{h-1}]\overline{\psi}_h^*(\tau_h) \right) \right| \pi(\omega_h|j_0, x_h) \right) \pi(x_h|j_0)
$$

$$
\overset{(b)}{\leq} \frac{1}{\gamma} \sum_{x_h} \left\| \overline{\mathbb{P}}_{\widehat{\theta}^k}[x_h \mid \tau_{h-1}]\widehat{\overline{\psi}}_h(\tau_h) - \overline{\mathbb{P}}_{\theta^*}[x_h \mid \tau_{h-1}]\overline{\psi}_h^*(\tau_h) \right\|_1 \pi(x_h|j_0)
$$

$$
\overset{(c)}{=} \frac{1}{\gamma} \sum_{x_h} \sum_{q_h \in \mathbb{Q}_h} \left| \overline{\mathbb{P}}_{\widehat{\theta}^k}[x_h, q_h \mid \tau_{h-1}] - \overline{\mathbb{P}}_{\theta^*}[x_h, q_h \mid \tau_{h-1}] \right| \pi(x_h|j_0)
$$

where step (a) is by the fact that $M_h(x_h)\overline{\psi}_{h-1}(\tau_{h-1}) = \overline{\mathbb{P}}[x_h \mid \tau_{h-1}]\overline{\psi}(\tau_h)$, as shown in Equation (28), step (b) is by Assumption 1, and step (c) is since $\left[\overline{\psi}_h(\tau_h)\right]_l = \mathbb{P}_\theta\left[q_h^l \mid \tau_h\right]$.

Combining the above, we have that,

$$
I_2 \leq \sum_{t<k} \mathbb{E}_{j \sim \mathbb{P}_{\theta^\star}^{\pi^k}} (I_3 + I_4)^2
$$

$$
\leq \sum_{t<k} \mathbb{E}_{j \sim \mathbb{P}_{\theta^\star}^{\pi^k}} \left[ \left( \frac{1}{\gamma} \sum_{q_{h-1} \in \mathbb{Q}_{h-1}} \left| \overline{\mathbb{P}}_{\widehat{\theta}^k}[q_{h-1} \mid \tau_{h-1}] - \overline{\mathbb{P}}_{\theta^*}[q_{h-1} \mid \tau_{h-1}] \right| \right.\right.
$$

$$
\left.\left. + \frac{1}{\gamma} \sum_{x_h} \sum_{q_h \in \mathbb{Q}_h} \left| \overline{\mathbb{P}}_{\widehat{\theta}^k}[x_h, q_h \mid \tau_{h-1}] - \overline{\mathbb{P}}_{\theta^*}[x_h, q_h \mid \tau_{h-1}] \right| \pi(x_h|j_0) \right)^2 \right]
$$

$$
= \frac{1}{\gamma^2} \cdot \sum_{t<k} \mathbb{E}_{j \sim \mathbb{P}_{\theta^\star}^{\pi^k}} \left[ \left( \sum_{q_{h-1} \in \mathbb{Q}_{h-1}} \left| \overline{\mathbb{P}}_{\widehat{\theta}^k}[q_{h-1} \mid \tau_{h-1}] - \overline{\mathbb{P}}_{\theta^*}[q_{h-1} \mid \tau_{h-1}] \right| \right.\right.
$$

$$
\left.\left. + \frac{1}{\gamma} \sum_{x_h} \sum_{q_h \in \mathbb{Q}_h} \left| \overline{\mathbb{P}}_{\widehat{\theta}^k}[x_h, q_h \mid \tau_{h-1}] - \overline{\mathbb{P}}_{\theta^*}[x_h, q_h \mid \tau_{h-1}] \right| \pi(x_h|j_0) \right)^2 \right]
$$

$$
\overset{(a)}{\leq} \frac{1}{\gamma^2} \cdot \sum_{t<k} \mathbb{E}_{j \sim \mathbb{P}_{\theta^\star}^{\pi^k}} \left( \sum_{\omega_{h-1}^a \in \mathbb{Q}_{h-1}^{\mathrm{exp}}} \sum_{\omega_{h-1}^o} \left| \overline{\mathbb{P}}_{\widehat{\theta}^k}[\omega_{h-1}^o \mid \tau_{h-1}, \omega_{h-1}^a] - \overline{\mathbb{P}}_{\theta^*}[\omega_{h-1}^o \mid \tau_{h-1}, \omega_h^a] \right| \right)^2
$$

$$
= \frac{\left| \mathbb{Q}_{h-1}^{\mathrm{exp}} \right|^2}{\gamma^2} \cdot \sum_{t<k} \mathbb{E}_{j \sim \mathbb{P}_{\theta^\star}^{\pi^k}} \left[ \mathrm{D}_{\mathrm{TV}}^2 \left( \mathbb{P}_{\widehat{\theta}^k}^{\mathbf{u}_{h-1}^{\mathrm{exp}}} \left( \omega_{h-1}^o \mid \tau_{h-1}, \omega_{h-1}^a \right), \mathbb{P}_{\theta^*}^{\mathbf{u}_{h-1}^{\mathrm{exp}}} \left( \omega_{h-1}^o \mid \tau_{h-1}, \omega_h^a \right) \right) \right]
$$

$$
\overset{(b)}{\leq} \frac{4 \max_{s \in \mathcal{A}} |\mathbb{X}_s|^2 Q_A^2}{\gamma^2} \cdot \sum_{t<k} \mathrm{D}_{\mathrm{H}}^2 \left( \mathbb{P}_{\widehat{\theta}^k}^{\nu_h(\pi^t, \mathbf{u}_{h-1}^{\mathrm{exp}})}(\tau_H), \mathbb{P}_{\theta^*}^{\nu_h(\pi^t, \mathbf{u}_{h-1}^{\mathrm{exp}})}(\tau_H) \right),
$$

where step (a) follows from the definition of $\mathbb{Q}_{h-1}^{\mathrm{exp}}$ (same as Lemma 5), and step (b) is because the Hellinger distance bounds the total variation distance and since $\left| \mathbb{Q}_{h-1}^{\mathrm{exp}} \right| \leq 2 \max_{s \in \mathcal{A}} |\mathbb{X}_s| Q_A$. Hence, we have,

$$
I_2 \leq 4 \max_{s \in \mathcal{A}} |\mathbb{X}_s|^2 Q_A^2 \frac{1}{\gamma^2} \sum_{t<k} \mathrm{D}_{\mathrm{H}}^2 \left( \mathbb{P}_{\widehat{\theta}^k}^{\nu_h(\pi^t, u_{\mathbb{Q}_{h-1}^{\mathrm{exp}}})}(\tau_H), \mathbb{P}_{\theta^*}^{\nu_h(\pi^t, u_{\mathbb{Q}_{h-1}^{\mathrm{exp}}})}(\tau_H) \right).
$$

Now, combining the bound on $I_1$ and $I_2$ allows us to finally bound $\left\|\sum_i \pi_{i|j}\mathrm{sign}(w_i^\top x_j)w_i\right\|_{\Lambda_h}^2$ as follows,

$$
\begin{aligned}
&\left\|\sum_i \pi_{i|j}\mathrm{sign}(w_i^\top x_j)w_i\right\|_{\Lambda_h}^2 \\
&\leq \frac{4\lambda_0 Q_A^2 d}{\gamma^4} + \frac{4\max_{s\in\mathcal{A}}|\mathbb{X}_s|^2 Q_A^2}{\gamma^2}\cdot\sum_{t<k}\mathsf{D}_{\mathsf{H}}^2\left(\mathbb{P}_{\widehat{\theta}^k}^{\nu_h(\pi^t,\mathsf{u}_{h-1}^{\exp})}(\tau_H),\mathbb{P}_{\theta^*}^{\nu_h(\pi^t,\mathsf{u}_{h-1}^{\exp})}(\tau_H)\right)\\
&=:\left(\tilde{\alpha}_{h-1}^k\right)^2,
\end{aligned}
$$

We choose $\lambda_0 = \frac{\gamma^4}{4Q_A^2 d}$, and bound $\tilde{\alpha}^2 := \sum_h \left(\tilde{\alpha}_{h-1}^k\right)^2$ as follows,

$$
\begin{aligned}
\sum_h \left(\tilde{\alpha}_{h-1}^k\right)^2 &= H + \frac{4\max_{s\in\mathcal{A}}|\mathbb{X}_s|^2 Q_A^2\beta}{\gamma^2}\sum_{\pi\in\mathcal{D}^k}\mathsf{D}_{\mathsf{H}}^2\left(\mathbb{P}_{\widehat{\theta}^k}^{\nu_h(\pi^t,\mathsf{u}_{h-1}^{\exp})}(\tau_H),\mathbb{P}_{\theta^*}^{\nu_h(\pi^t,\mathsf{u}_{h-1}^{\exp})}(\tau_H)\right)\\
&\leq H + \frac{28\max_{s\in\mathcal{A}}|\mathbb{X}_s|^2 Q_A^2\beta}{\gamma^2}\\
&\lesssim \frac{\max_{s\in\mathcal{A}}|\mathbb{X}_s|^2 Q_A^2\beta}{\gamma^2},
\end{aligned}
$$

where the second line is by the estimation guarantee of Lemma 4.

Thus, we have,

$$
\begin{aligned}
\mathsf{D}_{\mathsf{TV}}\left(\mathbb{P}_{\theta^\star}^{\pi^k}(\tau_H),\mathbb{P}_{\widehat{\theta}^k}^{\pi^k}(\tau_H)\right) &\leq \sum_{h=1}^H\sum_{\tau_H}\left|\widehat{m}^k(\omega_h)^\top\left(\widehat{M}_h^k(x_h)-M_h^\star(x_h)\right)\psi^\star(\tau_{h-1})\right|\pi^k(\tau_H)\\
&\leq \sum_{h=1}^H\mathbb{E}_{\tau_{h-1}\sim\mathbb{P}_{\theta^\star}^{\pi^k}}\left[\left\|\overline{\psi}^*(\tau_{h-1})\right\|_{\Lambda_h^\dagger}\left\|\sum_i \pi_{i|j}\mathrm{sign}(w_i^\top x_j)w_i\right\|_{\Lambda_h}\right]\\
&\leq \mathbb{E}_{\tau_{h-1}\sim\mathbb{P}_{\theta^\star}^{\pi^k}}\left[\sum_{h=1}^H\left\|\overline{\psi}^*(\tau_{h-1})\right\|_{\Lambda_h^\dagger}\left\|\sum_i \pi_{i|j}\mathrm{sign}(w_i^\top x_j)w_i\right\|_{\Lambda_h}\right]\\
&\overset{(a)}{\leq}\mathbb{E}_{\tau_{h-1}\sim\mathbb{P}_{\theta^\star}^{\pi^k}}\left[\sqrt{\sum_{h=1}^H\left\|\overline{\psi}^*(\tau_{h-1})\right\|_{\Lambda_h^\dagger}^2}\sqrt{\sum_{h=1}^H\left\|\sum_i \pi_{i|j}\mathrm{sign}(w_i^\top x_j)w_i\right\|_{\Lambda_h}^2}\right]\\
&\overset{(b)}{\leq}\tilde{\alpha}\cdot\mathbb{E}_{\tau_{h-1}\sim\mathbb{P}_{\theta^\star}^{\pi^k}}\left[\sqrt{\sum_{h=1}^H\left\|\overline{\psi}^*(\tau_{h-1})\right\|_{\Lambda_h^\dagger}^2}\right]\\
&\leq\tilde{\alpha}\cdot\sqrt{\sum_{h=1}^H\mathbb{E}_{\tau_{h-1}\sim\mathbb{P}_{\theta^\star}^{\pi^k}}\left[\left\|\overline{\psi}^*(\tau_{h-1})\right\|_{\Lambda_h^\dagger}^2\right]},
\end{aligned}
$$

where step (a) is by the Cauchy-Schwarz inequality and step (b) is by the bound established above. Since the total variation distance is bounded above by 2, we have

$$
\mathsf{D}_{\mathsf{TV}}\left(\mathbb{P}_{\theta^\star}^{\pi^k}(\tau_H),\mathbb{P}_{\widehat{\theta}^k}^{\pi^k}(\tau_H)\right)\leq\min\left\{\tilde{\alpha}\cdot\sqrt{\sum_{h=1}^H\mathbb{E}_{\tau_{h-1}\sim\mathbb{P}_{\theta^\star}^{\pi^k}}\left[\left\|\overline{\psi}^*(\tau_{h-1})\right\|_{\Lambda_h^\dagger}^2\right]},2\right\}.
$$

Finally, the proof is completed by summing over $k$ using the elliptical potential lemma as follows,

$$\sum_{k=1}^{K} \mathsf{D}_{\mathrm{TV}}\left(\mathbb{P}_{\theta^\star}^{\pi^k}(\tau_H), \mathbb{P}_{\widehat{\theta}^k}^{\pi^k}(\tau_H)\right) \leq \sum_{k=1}^{K} \min\left\{\tilde{\alpha} \cdot \sqrt{\sum_{h=1}^{H} \mathbb{E}_{\tau_{h-1} \sim \mathbb{P}_{\theta^\star}^{\pi^k}}\left[\left\|\overline{\psi}^*(\tau_{h-1})\right\|_{\Lambda_h^\dagger}^2\right]}, 2\right\}$$

$$\overset{(a)}{\leq} \sqrt{K}\sqrt{\sum_{k=1}^{K}\sum_{h=1}^{H} \min\left\{\tilde{\alpha}^2 \cdot \mathbb{E}_{\tau_{h-1} \sim \mathbb{P}_{\theta^\star}^{\pi^k}}\left[\left\|\overline{\psi}^*(\tau_{h-1})\right\|_{\Lambda_h^\dagger}^2\right], 4\right\}}$$

$$\leq \sqrt{K}\tilde{\alpha}\sqrt{\sum_{k=1}^{K}\sum_{h=1}^{H} \min\left\{\mathbb{E}_{\tau_{h-1} \sim \mathbb{P}_{\theta^\star}^{\pi^k}}\left[\left\|\overline{\psi}^*(\tau_{h-1})\right\|_{\Lambda_h^\dagger}^2\right], 4/\tilde{\alpha}^2\right\}}$$

$$\overset{(b)}{\leq} \sqrt{KH}\tilde{\alpha}\sqrt{(1 + 4/\tilde{\alpha}^2)r\log(1 + K/\lambda_0)}$$

$$\overset{(c)}{\lesssim} \frac{\max_{s\in\mathcal{A}}|\mathbb{X}_s|Q_A}{\gamma}\sqrt{rKH\beta\log(1 + K/\lambda_0)}$$

$$\overset{(d)}{\lesssim} \frac{\max_{s\in\mathcal{A}}|\mathbb{X}_s|Q_A}{\gamma}\sqrt{rKH\beta\log(1 + dQ_AK/\gamma)}.$$

Here, step (a) is uses the relationship between the $\ell_1$ and $\ell_2$ norms $\|\cdot\|_1 \leq \sqrt{d}\|\cdot\|_2$. Step (b) is by the elliptical potential lemma ([35, Lemma 14]; see also [96–98]). Step (c) uses the bound on $\tilde{\alpha}$ established above and the fact that $\sqrt{1 + 4/\tilde{\alpha}^2}$ is bounded by an absolute constant. Step (d) uses the definition of $\lambda_0$ and the fact that $\sqrt{28(1 + 4/\tilde{\alpha}^2)}$ is bounded by an absolute constant. □

Using the two lemmas above, we are now ready to show that $\sum_{k=1}^{K} V_{\widehat{\theta}^k}^{\widehat{b}^k}(\pi^k) = O(\sqrt{K})$. The argument is identical to [35, Lemma 6] and does not require modification for generalized PSRs. We recount the argument for completeness.

**Lemma 9.** *Under the event $\mathcal{E}$, with probability at least $1 - \delta$, we have:*

$$\sum_{k=1}^{K} V_{\widehat{\theta}^k}^{\widehat{b}^k}(\pi^k) \lesssim \left(\sqrt{r} + \frac{Q_A\sqrt{H}}{\gamma}\right)\frac{\max_{s\in\mathcal{A}}^2 Q_A^2 H\sqrt{drH\beta K\beta_0}}{\gamma^2}$$

*where $\beta_0 = \max\{\log(1 + K/\lambda), \log(1 + dQ_AK/\gamma)\}$, and $\lambda = \frac{\gamma\max_{s\in\mathcal{A}}|\mathbb{X}_s|Q_A\beta\max\{\sqrt{r}, Q_A\sqrt{H}/\gamma\}}{\sqrt{dH}}$*

*Proof.* First, we note that,

$$V_{\widehat{\theta}^k}^{\widehat{b}^k}(\pi^k) = \sum_\tau \mathbb{P}_{\widehat{\theta}^k}^{\pi^k}(\tau)\widehat{b}^k(\tau) = \sum_\tau \mathbb{P}_{\theta^\star}^{\pi^k}(\tau)\widehat{b}^k(\tau) + \sum_\tau (\mathbb{P}_{\widehat{\theta}^k}^{\pi^k}(\tau) - \mathbb{P}_{\theta^\star}^{\pi^k}(\tau))\widehat{b}^k(\tau) \leq V_{\theta^\star}^{\widehat{b}^k}(\pi^k) + \mathsf{D}_{\mathrm{TV}}\left(\mathbb{P}_{\widehat{\theta}^k}^{\pi^k}, \mathbb{P}_{\theta^\star}^{\pi^k}\right),$$

where we recall that $\widehat{b}^k(\cdot) \in [0, 1]$. Hence, we may focus on bounding the value of $\widehat{b}^k$ under the true model $\theta^*$ and use the bound on the cumulative total variation estimation error established in Lemma 8.

Recall the definition of the bonus term,

$$\widehat{b}^k(\tau_H) := \min\left\{\alpha\sqrt{\sum_h \left\|\widehat{\overline{\psi}}^k(\tau_h)\right\|_{(\widehat{U}_h^k)^{-1}}}, 1\right\},$$

which is defined in terms of the estimated prediction features $\widehat{\overline{\psi}}$. Recall also that in Lemma 7 we established a bound on the expectation of the prediction features under the true model, which

corresponds to $V_{\theta^*}^{\widehat{b}^k}$. Hence, we proceed to bound $\sum_k V_{\theta^*}^{\widehat{b}^k}(\pi^k)$ as follows,

$$
\sum_k V_{\theta^*}^{\widehat{b}^k}(\pi^k)
$$

$$
= \sum_k \mathbb{E}_{\tau_H \sim \mathbb{P}_{\theta^*}^{\pi^k}} \left[ \widehat{b}^k(\tau_H) \right]
$$

$$
= \sum_k \mathbb{E}_{\tau_H \sim \mathbb{P}_{\theta^*}^{\pi^k}} \left[ \min \left\{ \alpha \sqrt{ \sum_h \left\| \widehat{\overline{\psi}}^k(\tau_h) \right\|^2 }, 1 \right\} \right]
$$

$$
\overset{(a)}{\leq} \sum_{k=1}^K \min \left\{ \alpha \left( 1 + \frac{2 \max_{s \in \mathcal{A}} |\mathbb{X}_s| Q_A \sqrt{7r\beta}}{\sqrt{\lambda}} \right) \sum_{h=0}^{H-1} \mathbb{E}_{\tau_H \sim \mathbb{P}_{\theta^*}^{\pi^k}} \left[ \left\| \overline{\psi}^*(\tau_h) \right\|_{(U_h^k)^{-1}} \right] + \sum_{k=1}^K \frac{\alpha H Q_A}{\sqrt{\lambda}} \mathrm{D_{TV}} \left( \mathbb{P}_{\theta^*}^{\pi^k}, \mathbb{P}_{\widehat{\theta}^k}^{\pi^k} \right), 1 \right\}
$$

$$
\overset{(b)}{\leq} \underbrace{\sum_{k=1}^K \min \left\{ \alpha \left( 1 + \frac{2 \max_{s \in \mathcal{A}} |\mathbb{X}_s| Q_A \sqrt{7r\beta}}{\sqrt{\lambda}} \right) \sum_{h=0}^{H-1} \mathbb{E}_{\tau_H \sim \mathbb{P}_{\theta^*}^{\pi^k}} \left[ \left\| \overline{\psi}^*(\tau_h) \right\|_{(U_h^k)^{-1}} \right], 1 \right\}}_{I_1} + \sum_{k=1}^K \frac{\alpha H Q_A}{\sqrt{\lambda}} \mathrm{D_{TV}} \left( \mathbb{P}_{\theta^*}^{\pi^k}, \mathbb{P}_{\widehat{\theta}^k}^{\pi^k} \right),
$$

where step (a) is by Lemma 7 and step (b) is since $\min(a + b, c) \leq \min(a, c) + b$ when $a, b, c$ are non-negative.

Next, we bound the term $I_1$. Recall the definition of $U_h^k := \lambda I + \sum_{\tau_h \in \mathcal{D}_h^k} \overline{\psi}^*(\tau_h) \overline{\psi}^*(\tau_h)^\top$. Also, note that the process

$$
\left( \mathbb{E}_{\tau_h \sim \mathbb{P}_{\theta^*}^{\pi^k}} \left[ \left\| \overline{\psi}^*(\tau_h) \right\|_{(U_h^k)^{-1}} \right] - \left\| \overline{\psi}^*(\tau_h^{k+1,h+1}) \right\|_{(\widehat{U}_h^k)^{-1}} \right)_{k=1}^K
$$

is a martingale. Hence, by the Azuma-Hoeffding inequality, we have that with probability at least $1 - \delta$,

$$
I_1 \leq \sqrt{2K \log(2/\delta)} + \sum_{k=1}^K \min \left\{ \alpha \left( 1 + \frac{2 \max_{s \in \mathcal{A}} |\mathbb{X}_s| Q_A \sqrt{7r\beta}}{\sqrt{\lambda}} \right) \sum_{h=0}^{H-1} \left\| \overline{\psi}^*(\tau_h^{k+1,h+1}) \right\|_{(U_h^k)^{-1}}, 1 \right\}
$$

$$
\lesssim \sqrt{2K \log(2/\delta)} + \alpha \left( 1 + \frac{2 \max_{s \in \mathcal{A}} |\mathbb{X}_s| Q_A \sqrt{7r\beta}}{\sqrt{\lambda}} \right) H \sqrt{rK \log(1 + K/\lambda)}
$$

$$
\lesssim \alpha \left( 1 + \frac{\max_{s \in \mathcal{A}} |\mathbb{X}_s| Q_A \sqrt{7r\beta}}{\sqrt{\lambda}} \right) H \sqrt{rK \log(1 + K/\lambda)}
$$

where the second line is by the Elliptical potential lemma ([35, Lemma 14]; see also [96–98]).

We now return to bounding $\sum_{k=1}^K V_{\widehat{\theta}^k}^{\widehat{b}^k}(\pi^k)$. For convenience we define $\beta_0 := \max\{\log(1 + K/\lambda), \log(1 + dQ_A K/\gamma)\}$ and we choose $\lambda$ as follows,

$$
\lambda = \frac{\gamma \max_{s \in \mathcal{A}} |\mathbb{X}_s|^2 Q_A \beta \max\{\sqrt{r}, Q_A \sqrt{H}/\gamma\}}{\sqrt{dH}}.
$$

We have,

$$\sum_{k=1}^{K} V_{\widehat{\theta}^k}^{\widehat{b}^k}(\pi^k) \le \sum_{k=1}^{K} V_{\theta^*}^{\widehat{b}^k}(\pi^k) + \sum_{k=1}^{K} \mathtt{D_{TV}}\left(\mathbb{P}_{\widehat{\theta}^k}^{\pi^k}, \mathbb{P}_{\theta^*}^{\pi^k}\right)$$

$$\le I_1 + \left(1 + \frac{\alpha H Q_A}{\sqrt{\lambda}}\right) \sum_{k=1}^{K} \mathtt{D_{TV}}\left(\mathbb{P}_{\theta^*}^{\pi^k}, \mathbb{P}_{\widehat{\theta}^k}^{\pi^k}\right)$$

$$\overset{(a)}{\lesssim} \alpha \left(1 + \frac{\max_{s\in\mathcal{A}}|\mathbb{X}_s| Q_A \sqrt{7r\beta}}{\sqrt{\lambda}}\right) H\sqrt{rK\beta_0} + \frac{\alpha H}{\sqrt{\lambda}} \frac{Q_A^2 \max_{s\in\mathcal{A}}|\mathbb{X}_s|\sqrt{\beta}}{\gamma}\sqrt{rHK\beta_0}$$

$$= \alpha \left(1 + \frac{\max_{s\in\mathcal{A}}|\mathbb{X}_s| Q_A \sqrt{7r\beta}}{\sqrt{\lambda}} + \frac{\max_{s\in\mathcal{A}}|\mathbb{X}_s| Q_A^2 \sqrt{H\beta}}{\gamma\sqrt{\lambda}}\right) H\sqrt{rK\beta_0}$$

$$\overset{(b)}{\lesssim} \left(\frac{Q_A\sqrt{Hd\lambda}}{\gamma^2} + \frac{\max_{s\in\mathcal{A}}|\mathbb{X}_s| Q_A \sqrt{\beta}}{\gamma}\right)\left(1 + \frac{\max_{s\in\mathcal{A}}|\mathbb{X}_s| Q_A \sqrt{\beta}\max\{\sqrt{r}, Q_A\sqrt{H}/\gamma\}}{\sqrt{\lambda}}\right) H\sqrt{rK\beta_0}$$

$$= \left(1 + \frac{\sqrt{dH\lambda}}{\max_{s\in\mathcal{A}}|\mathbb{X}_s|\sqrt{\beta}\gamma}\right)\left(1 + \frac{\max_{s\in\mathcal{A}}|\mathbb{X}_s| Q_A \sqrt{\beta}\max\{\sqrt{r}, Q_A\sqrt{H}/\gamma\}}{\sqrt{\lambda}}\right) \frac{\max_{s\in\mathcal{A}}|\mathbb{X}_s| Q_A H\sqrt{rK\beta\beta_0}}{\gamma}$$

$$\overset{(c)}{=} \left(1 + \sqrt{\frac{Q_A \max\{\sqrt{r}, Q_A\sqrt{H}/\gamma\}\sqrt{dH}}{\gamma}}\right)^2 \frac{\max_{s\in\mathcal{A}}|\mathbb{X}_s| Q_A H\sqrt{rK\beta\beta_0}}{\gamma}$$

$$\lesssim \left(1 + \frac{Q_A\sqrt{dH}\max\left\{\sqrt{r}, Q_A\sqrt{H}/\gamma\right\}}{\gamma}\right)\frac{\max_{s\in\mathcal{A}}|\mathbb{X}_s| Q_A H\sqrt{rK\beta\beta_0}}{\gamma}$$

$$\le \left(\sqrt{r} + \frac{Q_A\sqrt{H}}{\gamma}\right)\frac{\max_{s\in\mathcal{A}}|\mathbb{X}_s| Q_A^2 H\sqrt{rdHK\beta\beta_0}}{\gamma^2},$$

where step (a) is by Lemma 8 and the bound on $I_1$ established above, step (b) uses the definition of $\alpha$ and the fact that $\alpha \lesssim \frac{Q_A\sqrt{Hd\lambda}}{\gamma^2} + \frac{\max_{s\in\mathcal{A}}|\mathbb{X}_s| Q_A\sqrt{\beta}}{\gamma}$, and step (c) is by plugging in the choice of $\lambda$. □

## J.5  Proof of Theorem 4

**Theorem** (Restatement of Theorem 4). *Suppose Assumption 1 holds. Let* $p_{\min} = O\left(\frac{\delta}{KH\prod_{h=1}^{H}|\mathbb{X}_h|}\right)$, $\lambda = \frac{\gamma \max_{s\in\mathcal{A}}|\mathbb{X}_s|^2 Q_A \beta \max\{\sqrt{r}, Q_A\sqrt{H}/\gamma\}}{\sqrt{dH}}$, $\alpha = O\left(\frac{Q_A\sqrt{Hd}}{\gamma^2}\sqrt{\lambda} + \frac{\max_{s\in\mathcal{A}}|\mathbb{X}_s| Q_A\sqrt{\beta}}{\gamma}\right)$, *and let* $\beta = O(\log|\overline{\Theta}_\varepsilon|)$, *where* $\varepsilon = O(\frac{p_{\min}}{KH})$. *Then, with probability at least* $1 - \delta$, *Algorithm 1 returns a model* $\theta^\epsilon$ *and a policy* $\pi$ *that satisfy*

$$V_{\theta^\epsilon}^R(\pi^*) - V_{\theta^\epsilon}^R(\pi) \le \varepsilon, \text{ and } \forall\pi, \ \mathtt{D_{TV}}\left(\mathbb{P}_{\theta^\epsilon}^\pi(\tau_H), \mathbb{P}_{\theta^*}^\pi(\tau_H)\right) \le \varepsilon.$$

*In addition, the algorithm terminates with a sample complexity of,*

$$\tilde{O}\left(\left(r + \frac{Q_A^2 H}{\gamma^2}\right)\frac{rdH^3 \max_{s\in\mathcal{A}}|\mathbb{X}_s|^2 Q_A^4 \beta}{\gamma^4\epsilon^2}\right).$$

*Proof.* By Propositions 6 to 8, the event $\mathcal{E}$ occurs with high probability, $\mathbb{P}[\mathcal{E}] \ge 1 - 3\delta$. Suppose $\mathcal{E}$ holds. Then, by the upper confidence bound established in Corollary 4, if Algorithm 1 terminates, then the following must hold,

$$\forall\pi, \ \mathtt{D_{TV}}\left(\mathbb{P}_{\theta^\epsilon}^\pi(\tau_H), \mathbb{P}_{\theta^*}^\pi(\tau_H)\right) = 2\max_R\left|V_{\theta^\epsilon}^R(\pi) - V_{\theta^*}^R(\pi)\right| \le V_{\theta^\epsilon}^{\widehat{b}^\epsilon}(\pi) \le \epsilon,$$

where the maximization is over reward functions $R : \mathbb{H}_H \to [0, 1]$. The last inequality is simply the termination condition of Algorithm 1.

Now, the difference between the optimal value and the value of $\pi$ (the policy returned by the algorithm) can be bounded as follows,

$$V_{\theta^*}^R(\pi^*) - V_{\theta^*}^R(\pi) = V_{\theta^*}^R(\pi^*) - V_{\theta^\epsilon}^R(\pi^*) + V_{\theta^\epsilon}^R(\pi^*) - V_{\theta^\epsilon}^R(\pi) + V_{\theta^\epsilon}^R(\pi) - V_{\theta^*}^R(\pi)$$
$$\leq 2 \max_\pi V_{\theta^\epsilon}^{\widehat{b}^\epsilon}(\pi) \leq \epsilon,$$

where the inequality follows from the fact that $\pi = \arg\max_\pi V_{\theta^\epsilon}^R(\pi)$ and by Corollary 4.

Recall that by Lemma 9, we have,

$$\sum_{k=1}^K V_{\widehat{\theta}^k, \bar{b}^k}^{\pi^k} \lesssim \left( \sqrt{r} + \frac{Q_A \sqrt{H}}{\gamma} \right) \frac{\max_{s \in \mathcal{A}} |\mathbb{X}_s| \, Q_A^2 H \sqrt{rdHK\beta\beta_0}}{\gamma^2}.$$

By the pigeon-hole principle and the termination condition of Algorithm 1, the algorithm must terminate within

$$K = \tilde{O}\left( \left( r + \frac{Q_A^2 H}{\gamma^2} \right) \frac{rdH^2 Q_A^4 \max_{s \in \mathcal{A}} |\mathbb{X}_s|^2 \beta}{\gamma^4 \epsilon^2} \right)$$

episodes. Since each episode contains $H$ iterations, this implies a sample complexity of

$$K = \tilde{O}\left( \left( r + \frac{Q_A^2 H}{\gamma^2} \right) \frac{rdH^3 Q_A^4 \max_{s \in \mathcal{A}} |\mathbb{X}_s|^2 \beta}{\gamma^4 \epsilon^2} \right).$$

This concludes the proof of Theorem 4. □

# K    Proof of Theorem 5: UCB Algorithm for Generalized PSRs (Game Setting)

**Theorem** (Restatement of Theorem 5). *Suppose Assumption 1 holds. Let $p_{\min} = O\left( \frac{\delta}{KH \prod_{h=1}^H |\mathbb{X}_h|} \right)$,*
$\lambda = \frac{\gamma \max_{s \in \mathcal{A}} |\mathbb{X}_s|^2 Q_A \beta \max\{\sqrt{r}, Q_A \sqrt{H}/\gamma\}}{\sqrt{dH}}$, $\alpha = O\left( \frac{Q_A \sqrt{Hd}}{\gamma^2} \sqrt{\lambda} + \frac{\max_{s \in \mathcal{A}} |\mathbb{X}_s| Q_A \sqrt{\beta}}{\gamma} \right)$, *and let* $\beta = O(\log |\overline{\Theta}_\varepsilon|)$, *where $\varepsilon = O(\frac{p_{\min}}{KH})$. Then, with probability at least $1 - \delta$, Algorithm 2 returns a model $\theta^\epsilon$ and a policy $\pi$ which is an $\varepsilon$-approximate equilibrium (either NE or CCE). That is,*

$$V_{\theta^*}^i(\pi) \geq V_{\theta^*}^{i,\dagger}(\pi^{-i}) - \varepsilon, \; \forall i \in [N].$$

*In addition, the algorithm terminates with a sample complexity of,*

$$\tilde{O}\left( \left( r + \frac{Q_A^2 H}{\gamma^2} \right) \frac{rdH^3 \max_{s \in \mathcal{A}} |\mathbb{X}_s|^2 Q_A^4 \beta}{\gamma^4 \epsilon^2} \right).$$

*Proof.* Recall that the model-estimation portion of Algorithm 2 is identical to Algorithm 1. Hence, by Theorem 4, the returned estimated model $\theta^\varepsilon$ satisfies,

$$\mathsf{D}_{\mathrm{TV}}\left( \mathbb{P}_{\theta^\varepsilon}^{\boldsymbol{\pi}}(\tau_H), \mathbb{P}_{\theta^*}^{\boldsymbol{\pi}}(\tau_H) \right) \leq \varepsilon/2,$$

for any collection of policies $\boldsymbol{\pi} = (\pi^i : i \in [N])$. This implies that $V_{\theta^*}^i(\pi) \geq V_{\theta^\varepsilon}^i(\pi) - \varepsilon/2$ for all $i \in [N]$.

Let $\Gamma^i = \Gamma_{\mathrm{ind}}^i$ in the case of running the algorithm to find a Nash equilibrium and $\Gamma^i = \Gamma_{\mathrm{cor}}^i$ in the case of a coarse correlated equilibrium. Recall that the collection of policies $\pi = (\pi^1, \ldots, \pi^N)$ returned by the algorithm are an equilibrium under $\theta^\varepsilon$. That is, for all $i \in [N]$,

$$V_{\theta^\varepsilon}^i(\pi) = \max_{\tilde{\pi}^i \in \Gamma_{\mathrm{ind}}^i} V_{\theta^\varepsilon}^i(\tilde{\pi}^i, \pi^{-i}) =: V_{\theta^\varepsilon}^{i,\dagger}(\pi^{-i}).$$

Moreover, note that,

$$\left| V_{\theta^\varepsilon}^{i,\dagger}(\pi^{-i}) - V_{\theta^*}^{i,\dagger}(\pi^{-i}) \right| = \left| \max_{\tilde{\pi}^i} V_{\theta^\varepsilon}^i(\tilde{\pi}^i, \pi^{-i}) - \max_{\tilde{\pi}^i} V_{\theta^*}^i(\tilde{\pi}^i, \pi^{-i}) \right|$$
$$\leq \max_{\tilde{\pi}^i} \left| V_{\theta^\varepsilon}^i(\tilde{\pi}^i, \pi^{-i}) - V_{\theta^*}^i(\tilde{\pi}^i, \pi^{-i}) \right|$$
$$\leq \varepsilon/2,$$

where the final inequality is since $D_{\mathrm{TV}}\left(\mathbb{P}^{\boldsymbol{\pi}}_{\theta^\varepsilon}(\tau_H), \mathbb{P}^{\boldsymbol{\pi}}_{\theta^*}(\tau_H)\right) \leq \varepsilon/2$ for any $\boldsymbol{\pi}$. Thus, $V^{i,\dagger}_{\theta^\varepsilon}(\pi^{-i}) \geq V^{i,\dagger}_{\theta^*}(\pi^{-i}) - \varepsilon/2$.

Putting this together, we have,

$$
\begin{aligned}
V^i_{\theta^*}(\pi) &\geq V^i_{\theta^\varepsilon}(\pi) - \varepsilon/2 \\
&= V^{i,\dagger}_{\theta^\varepsilon}(\pi^{-i}) - \varepsilon/2 \\
&\geq V^{i,\dagger}_{\theta^*}(\pi^{-i}) - \varepsilon.
\end{aligned}
$$

Hence, $\pi$ is an $\varepsilon$-approximate equilibrium (either NE or CCE). $\qquad\square$

