# OpenReview forum: "On the Role of Information Structure in Reinforcement Learning for Partially-Observable Sequential Teams and Games"
_NeurIPS.cc/2024/Conference — NeurIPS 2024 poster_

### Official Review · Reviewer_Ugr3 · 2024-06-29

**Soundness:** 3
**Presentation:** 2
**Contribution:** 3
**Rating:** 6
**Confidence:** 4

**Summary:**

This paper proposes an abstract model of RL with information structures. Under this mode, the paper investigates the necessary conditions for sample-efficient learning.

**Strengths:**

The results of this paper are very general, in the sense that they encompass the results of several previous works.

**Weaknesses:**

(1) It is hard to identify the technical novelty & originality of this paper. The POSTs and POSGs are special cases of "generalized PSRs", but the so-called "generalized PSR" is largely a direct abstraction of the analysis framework of Liu et al. 2022b. The algorithm and the proofs are also largely patching the analysis in the previous papers (in particular, Liu et al. 2022b).

(2) It is not clear what new concrete tractable class does this paper identify. Tractable POMDP classes and POMG classes have already been studied in previous works. Several examples of information structure are provided, but there isn't a definition of a new, meaningful, and tractable problem class.

Overall, this paper indeed provides a general framework, but it seems that generality is the only advantage of this framework. However, "general framework" cannot be an excuse. After all, there have already been several general frameworks for sequential decision making. For example, for model-based learning, the so-called Decision-Estimation Coefficient characterizes the statistical complexity of *any* model class [1,2,3,etc]. The SAIL framework of Liu et al. (2022b) is more concrete and it also applies to a broad class of sequential decision making problems. The authors have to justify what is the significance of this framework other than its (vacuous) generality.

[1] Dylan J Foster, Sham M Kakade, Jian Qian, and Alexander Rakhlin. The statistical complexity of interactive decision making.

[2] Dean Foster, Dylan J Foster, Noah Golowich, and Alexander Rakhlin. On the complexity of multi-agent decision making: From learning in games to partial monitoring.

[3] Dylan J Foster, Noah Golowich, and Yanjun Han. Tight guarantees for interactive decision making with the decision-estimation coefficient.

**Questions:**

N/A

---

> ### Author Rebuttal · Authors · 2024-08-07
>
> **Summary of review.** While the reviewer acknowledges the generality of our framework and technical results, they are skeptical about what advantage or insight this analysis grants compared to existing work, and what concrete new tractable model classes it identifies.
>
> Thank you for your review. We hope to address each of your concerns, clarifying the technical contributions of our work and explaining its significance.
>
> ---
>
> The goal of our work is to ***develop an understanding of how "information structure"---the structure of causal dependence between variables in a system---determines the statistical complexity of learning a sequential decision-making problem***.
>
> As you mentioned, there has been a series of recent works that studied classes of tractable POMDPs. However, these previous works tackle a different problem than we do. While many of these works indeed provide a general analysis of statistical complexity, for example in terms of eluder dimension-style quantities, this analysis is not directly linked to the underlying information structure of the system. Thus, in a sense, they are orthogonal to our work as they don't say anything about statistical complexity as a function of information structure or the causal structure in the system's variables.
>
> The main technical contribution of our work is to **relate statistical complexity to information structure**. This is an important perspective with key insights that are missing from existing general analyses like the ones you mentioned. In particular, the conditions provided in these existing general analyses are not directly connected to the underlying structure of system dynamics.
>
> In our work, we carry out an **analysis in terms of the graph representation of the causal structure between system variables**, both observable and unobservable. In particular, we identify a key quantity, defined in terms of $d$-separation operations on the DAG representation of the information structure, that we prove characterizes the statistical complexity of general sequential decision-making problems. A key insight from this analysis is **a systematic way to understand the difficulty of learning and modeling a system's evolution through the underlying causal structure between its variables**, whether those variables are observable or not. In particular, the existence of unobservable system variables that dictate the dynamics of the system can imply a bound on the complexity of observable dynamics that need to be modeled. A simple example of this is POMDPs, where the existence of an unobservable Markovian state implies a bound on the complexity of representing the dynamics of the observables. Our analysis generalizes this to arbitrary information structures.
>
> This characterization of statistical complexity in terms of information structure is, in many cases, a **more interpretable characterization of statistical complexity** than the types of conditions identified in current general analyses in the literature (e.g., in terms of eluder dimension or a SAIL condition, etc.) because it relates directly to the structure of the system dynamics rather than abstract constructs that may not have real-world physical interpretations.
>
> Moreover, the language of information structure is key to **understanding the phenomena of partial observability in a much more powerful way**. Our analysis enables considering a much **more general notion of partial observability**. In standard models, partial observability refers to "emissions" from an underlying Markovian state variable which encodes all relevant information about the system. Under our analysis, we consider a more general notion of partial observability in which system variables evolve according to an arbitrary causal structure (not necessarily Markovian), and the set of observables is specified as an arbitrary subset of all system variables.
>
> Thus, the generality of our framework and analysis provides deep insights into the fundamental theoretical underpinnings of reinforcement learning, characterizing the statistical complexity of learning sequential decision-making problems as a function of the causal structure between system variables.
>
> ---
>
> We will make an effort to emphasize the points explained above in the revised paper to better situate our work in the context of existing literature.
>
> ---
>
> > The POSTs and POSGs are special cases of "generalized PSRs",
>
> It's not quite right to say the POSTs and POSGs are special cases of "generalized PSRs". We develop generalized PSRs as a proof device to obtain an efficient *representation* of POSTs/POSGs which is amenable to learning. In this paper, we prove (by construction) that *certain classes* of POSTs/POSGs admit a well-conditioned representation as a generalized PSR, with a condition specified in terms of information structure. Not all POSTs/POSGs admit such a well-conditioned PSR representation. This information-structural condition identifies a new expanded class of tractable problems.
>
> > Several examples of information structure are provided, but there isn't a definition of a new, meaningful, and tractable problem class.
>
> The concrete tractable class of sequential decision-making problems is identified in terms of its information structure. It is the class of POSTs/POSGs where $|\mathbb{I}_h^\dagger|$ is bounded, defined in terms of the graph representation of information. The corresponding identifiability condition needed is defined in Definition 6, and is analogous to identifiability conditions considered in previous work on partially-observable models like POMDPs.
>
> The insight attained from such a characterization in terms of the causal structure becomes especially clear when considering physical systems, where the set of system variables and their causal dependence may be known, but only a subset of these variables are observable to the learning agent.
>
> ---
>
> We hope our responses address your concerns and look forward to further discussion.

---

> > ### Comment · Reviewer_Ugr3 · 2024-08-12
> >
> > Thank you for your response. The formulation of a "more general notion of partial observability" does make more sense to me.
> >
> > A concern is that the current organization seems to emphasize more on the rather abstract formulation, instead of providing more examples and concrete results. Examples in Figure 2 need more explanation and discussion, and to me, they are much more interesting than Section 4 (which is analogous to the existing literature in my opinion). It will be clearer if you can formally define the classes (a) - (d) in Figure 2, and also write down how Def 6 becomes in these classes. I believe this will address my concern (2).
> >
> > If you can provide a more thorough discussion on the examples of specific information structures, I am willing to raise my score to 5 and favor the acceptance of this paper.

---

> > > ### Author Response · Authors · 2024-08-12
> > >
> > > Dear reviewer,
> > >
> > > Thank you for your response. We appreciate your constructive engagement with us during the reviewer discussion period. We are glad that our elaboration and explanation makes sense to you. We are eager to address your remaining concern.
> > >
> > > > A concern is that the current organization seems to emphasize more on the rather abstract formulation, instead of providing more examples and concrete results.
> > >
> > > This point is well taken. The current version of the paper indeed emphasizes the abstract formulation over concrete examples. We fully agree on the importance of discussing concrete examples to better emphasize the significance of the results and provide a more clear and pedagogical presentation. **We will include a new section in the paper dedicated entirely to doing this**. We appreciate this feedback and we believe it improves the paper.
> > >
> > > First, we would like to briefly explain our choice regarding the current organization of the paper. We initially did have a section in the paper dedicated to doing exactly what you describe: a formal description of all the examples of information structures in Fig 2 together with a discussion of how our results can be applied to each information structure. However, we had to move this to the appendix due to space constraints. With the additional content page allowed in the camera-ready version, we will add a revised version of this section back to the main body of the paper, which we hope addresses your concern and grounds the general abstract results of the paper in concrete examples.
> > >
> > > We point to Appendix D in the paper, "Examples of Information Structures and their Rank", which perhaps provides exactly the type of discussion you are describing. For each information structure depicted in Fig 2, it provides:
> > > 1. a formal description of how it can be represented in the POST/POSG framework (i.e., what is the variable structure, information structure, etc., as per Def 4), and
> > > 2. a discussion of what our main results imply about the statistical complexity of each information structure. That is, it identifies information structural state space and its size (hence, by Theorem 3, the statistical complexity).
> > >
> > > To give an idea of this discussion (in case you are unable to view the paper in detail at the moment), we briefly summarize the discussion on the "point-to-point real-time communication with feedback" information structure (Fig 2 c) below. Please see the full section in the appendix for details on the other information structures.
> > >
> > > ---
> > > Let $x\_t$ be a Markov source. At time $t$, the encoder receives the source $x\_t \in \mathbb{X}$ and encodes sending a symbol $z\_t \in \mathbb{Z}$. The symbol is sent through a memoryless noisy channel which outputs $y\_t$ to the receiver. The decoder produces the estimate $\hat{x}\_t$. The output of the noisy channel is also fed back to the encoder. The encoder and decoder have a full memory of their previous actions (i.e., "messages").
> > >
> > > The sequence of variables is $x\_t, z\_t, y\_t, \hat{x}\_t, x\_{t+1}, z\_t, ...$, and map onto the index sets $\mathcal{S, A, O}$. $x\_t$ and $y\_t$ are (observable) system variables, and hence in $\mathcal{S} = \mathcal{O}$. $z\_t$ and $\hat{x}\_t$ are action variables (for the encoder and decoder, resp.), and hence are in $\mathcal{A}$. The information structure is given by:
> > > $$\mathcal{I}(x\_t) = \\{x\_{t-1}\\}, \mathcal{I}(z\_t) = \\{x\_{1:t}, y\_{1:t-1}, z\_{1:t-1}\\}, \mathcal{I}(y\_t) = \\{z\_t\\}, \mathcal{I}(\hat{x}\_t)=\\{y\_{1:t}\\}.$$
> > >
> > > By Theorem 1, the information structural state is given by:
> > > $$\mathcal{I}^\dagger(x\_t) = \\{x\_{t-1}\\}, \mathcal{I}^\dagger(z\_t) = \\{x\_t\\}, \mathcal{I}^\dagger(y\_t) = \\{x\_t, z\_t\\}, \mathcal{I}^\dagger(\hat{x}\_t) = \\{ x\_t \\}.$$
> > >
> > > Thus, by Theorem 1, the rank of the observable dynamics of this information structure is bounded by $r \leq |\mathbb{X}| |\mathbb{Z}|$.
> > >
> > > (Note, here, that $\mathcal{I}^\dagger(\cdot)$ means $\mathcal{I}\_h^\dagger$ for $h$ corresponding to the variable "$\cdot$"; that is, at the time point $h$ corresponding to "$\cdot$". Similarly for $\mathcal{I}^\dagger(\cdot)$. This is explained in Appdx D.)
> > >
> > > ---
> > >
> > > Please let us know whether this addresses your concern or if you have any remaining questions.
> > >
> > > > Examples in Figure 2 need more explanation and discussion, and to me, they are much more interesting than Section 4 (which is analogous to the existing literature in my opinion).
> > >
> > > On this, we briefly note that Section 4 is essential for our statistical complexity results, and generalizes existing PSR construction results and learnability conditions in the literature to arbitrary information structures. In particular, the $\mathcal{I}^\dagger$-weakly-revealing condition is crucially with respect to the information structural state---a new quantity we define in this work---rather than some assumed Markovian latent state (e.g., as in the case of existing conditions for learnability of POMDPs).

---

> ### Comment · Reviewer_Ugr3 · 2024-08-12
>
> Thank you for your detailed response. Indeed, it is better to move part of the discussion in Appendix D to the main text. Moreover, I would also like to know how the revealing condition (Definition 6) becomes in these examples. It will be good to see whether Definition 6 reduces to a more intuitive (or natural) condition, and what concrete insight it may provide into the learnability of partially observable systems.
>
> I will raise my rating as promised.

---

> > ### Author Response · Authors · 2024-08-12
> >
> > Dear reviewer,
> >
> > We appreciate your continued engagement and are glad that we were able to address your concerns.
> >
> > On your question about the weakly-revealing condition in these cases: the weakly revealing condition is defined with respect to the information structural state $\mathbb{I}_h^\dagger$ at each time step. That is, $m$-step futures must be weakly-revealing for the information structural state. More precisely, for each observable time step $h$, for two different distributions over the information structural state $\nu_1, \nu_2 \in \mathcal{P}(\mathbb{I}_h^\dagger)$ (with disjoint support), the distributions of $m$-step future observations that they observe are different.
> >
> > The importance of this condition is its generalization of existing observability conditions, which are defined with respect to an assumed Markovian state. Our analysis extends this to arbitrary information structures and identifies the quantity that the observability condition (which is necessary) must "reveal". That is, it identifies the collection of latent variables in the system that future observations must be sufficiently informative about to enable sample-efficient learning.
> >
> > For example, in the case of the "point-to-point real-time communication with feedback" information structure, the weakly-revealing condition states that future observations (which include the encodings/symbols and future sources $y_t, x_{t+1}, y_{t+1}, ...$) must be weakly-revealing about the source $x_t$. In the mean-field information structure, future observations (which include the local states of agents $s_{t+1}^1, ..., s_{t+1}^n, ...$) must be weakly revealing about the mean-field state at the previous time point $s_t^{mf} = n^{-1} \cdot \sum_i s_t^i$.
> >
> > We thank you again for your engagement with our work and for your feedback which has helped us improve the presentation of the paper.

---

### Official Review · Reviewer_w9Kg · 2024-07-05

**Soundness:** 3
**Presentation:** 2
**Contribution:** 2
**Rating:** 6
**Confidence:** 3

**Summary:**

The paper presents a new representation of sequential decision making problems with an information structure. They show that the information structure can be used to estimate the complexity of the decision-making problems and can guide learning algorithm development.

**Strengths:**

1. The authors present a good study on complexity of decision-making problems in finite state and action spaces
1. They identify scalable decision-making problems for learning applications.
1. The work appears to be technically sound, but I haven't checked all the proofs

**Weaknesses:**

1. If I understood correctly, the authors are dealing with finite state and action spaces and the complexity measure is effectively a rank of a certain matrix that they construct. If the state or the action spaces are infinite or continuous then the measure becomes infinite for all systems and we cannot use the measure for judging complexity. If this is so, I see it as a major weakness
1. Presentation can be improved. It’s hard to understand what different concepts mean, say core tests. Can a “layman” summary be given after these kinds of definitions?
1. While I appreciate that the paper is theoretic in nature, it is hard to judge it's practical implications in present or even in future.
1. Some claims need for further clarification or justification. For example, I am not sure that Figure 1 is fully substantiated. Hopefully, the authors can clarify

**Questions:**

1. The dynamics matrix seems to bear resemblance to the Hankel operator for linear systems theory in control literature. The Hankel operator is mapping past inputs to feature outputs, since the system is linear its information structure is rather simple.  The rank of the Hankel operator is actually a well-established complexity measure for linear systems. It would be good to understand if there’s more than just some general similarity.
1. Is there an example of a system that is POST but not POMG or not dec-POMDP as claimed in Figure 1?
1. Are the authors deal with finite state and action spaces? It seems that the dynamics matrix with probabilities of potential futures and histories is finite-dimensional, but it’s not clearly stated
1. “This   assumption is often unrealistic, since general systems may not have “states” per se, and observations may be generated with more complex dependencies.” - I am bit confused what does it mean that the system may not have states. Can you add an example here? I think that POMDPs are generally quite realistic when it comes to single agent systems hence this comment is confusing for me.
1. Please define dec-POMDP, POMG before using the acronyms. Or at least define them in the Appendix.
1. Why in def 5 we need to remove all the arrows pointing to actions? Can actions be dependent on history?
1. What’s $\sigma$ in Def 6?
1. Line 268. If I understand correctly $G_h$ is possibly a non-square matrix, what’s an eigenvalue in this context? Do the authors mean singular value as they speak about rank?

**Limitations:**

limitations are discussed

---

> ### Author Rebuttal · Authors · 2024-08-07
>
> Thank you for your review and feedback. We hope to address of each of your concerns in turn.
>
> ---
>
> **Finite vs Continuous Spaces**
> > ... the authors are dealing with finite state and action spaces ... If this is so, I see it as a major weakness
>
> Our results are stated and proved for finite spaces. Finite spaces reveal the interesting aspects of the role of information structure in studying the statistical complexity of sequential decision-making problems. Introducing continuous spaces can introduce certain technical complications that obfuscate the key message of our information structural analysis. We also note that the majority of previous related work considers finite spaces.
>
> Nonetheless, extensions to continuous spaces are possible. We provide some discussion of this below.
> - Our formulation of Generalized Predictive State Representations can be extended to capture continuous spaces for system variables. This requires a few technical modifications:
>     - In finite spaces, core test sets are a finite set of trajectories $\mathbb{Q}\_{h} = \\{q\_h^{1}, \ldots, q\_h^{d\_h} \\} \subset \mathbb{F}\_h$, where each $q_{h}^j = (x_{h+1}, ..., x_H)$ is a trajectory. In the extension to continuous spaces, the core tests become trajectories of subsets of the underlying continuous space. I.e., $q_h^j = (\mathcal{X}_{h+1}, ..., \mathcal{X}_H)$, where $\mathcal{X}_k \subset \mathbb{X}_k$ is a measurable subset.
>     - In finite spaces, the feature vector $\bar{\psi}(\tau\_h) = (\bar{P}(q\_h^j | \tau\_h))\_j$ is the vector probabilities of core tests. While in finite spaces $q_h^j$ is a single trajectory, in continuous spaces $q_h^j$ is a subset of trajectories, and the probabilities become $\bar{P}(x\_{h+1} \in \mathcal{X}\_{h+1}, ..., x\_{H} \in \mathcal{X}\_{H} | \tau\_h)$.
>     - In finite spaces, the observable operators $M\_h : \mathbb{X}\_h \to \mathbb{R}^{d\_h \times d_\{h-1}}$ map the variable at time $h$ to a matrix. In continuous spaces, they become linear functionals that map functions over $\mathbb{Q}_{h-1}$ to functions over $\mathbb{Q}_h$.
>     - This type of extension to continuous spaces has been considered in standard PSRs and can be extended to our generalized PSR model. From here, the rest of the analysis, including the causal structure analysis using the POST/POSG model and information structure can be appropriately extended.
> - Alternatively, our results can also be applied to continuous spaces via appropriate quantization while accounting for the approximation error due to quantization.
>
> We will provide a discussion of extensions to continuous spaces in the updated manuscript.
>
> ---
>
> **Generality of the POST/POSG Framework and the importance of Information Structure**
>
> > I am not sure that Figure 1 is fully substantiated. Hopefully, the authors can clarify
>
> In Figure 1, we depict the following relationship between our POST/POSG framework and existing models like POMDPs: MDP $\subset$ POMDP $\subset$ Dec-POMDP $\subset$ POST $\subset$ POSG. That is, via their general specification of information structure and variable structure, POST/POSG capture MDPs, POMDPs, and Dec-POMDPs as special cases.
>
> Information structure is the specification of how events (variables) in a system affect each other. MDPs, POMDPs, Dec-POMDPs, and POMGs each assume a particular variable structure and information structure. In particular, they also assume the existence of a *Markovian state* that depends *only* on the state at the previous time step. In our model, we make no such assumption and we allow the information structure to vary across different problem instances, representing the variable structure, information structure, and system dynamics as part of the model specification (by contrast MDPs/POMDPs/etc only represent system dynamics in the model specification and the rest are fixed). In our model, we have a sequence of variables $x_1, x_2, ...$, where each variable depends on an arbitrary subset of the past, can be either a system variable or an action variable, and can be either observable or not observable. This greatly expands the types of system dynamics that can be represented compared to models like POMDPs that assume a Markovian state, and also expands the notion of partial-observability as it now is no-longer defined with respect to an assumed idealized Markovian state.
>
> For further discussion of this, we point to the subsection "llustration: translating to the POST/POSG framework" in Appendix D which provides a detailed worked-through example of how models with fixed information structure such as POMDPs can be naturally represented in the POST/POSG framework.
>
> ---
> **Practical implications**
> > While I appreciate that the paper is theoretic in nature, it is hard to judge it's practical implications in present or even in future.
>
> The primary contribution and focus of our work lies in the development of a foundational understanding of the statistical complexity of reinforcement learning in terms of information structure. Our work has direct implications to representation learning in the context learning sequential decision-making problems. We argue that, when learning in arbitrary dynamic decision-making environments, the fundamental quantity that dictates the statistical complexity of a problem is the size of the information structural state space, $|\mathbb{I}_h^\dagger|$. We note that recent work in machine learning has studied the ability of sequence models like Transformers to learn and represent causal structures in sequential data. Our analysis suggests that the right scaling of the statistical complexity of such tasks, when applied to dynamic decision-making settings, depends on the quantity $|\mathbb{I}_h^\dagger|$. The analysis in this work can inform the design of architectural inductive biases that can better capture and represent such causal structures in order to achieve improved sample efficiency.
>
> We will add further discussion on these practical implications.

---

> ### Author Response · Authors · 2024-08-07
> **Questions & more minor comments**
>
> **Layman explanations of technical definitions**
>
> > It’s hard to understand what different concepts mean, say core tests. Can a “layman” summary be given after these kinds of definitions?
>
> We will expand on the explanations of technical concepts, and aim to provide simple, plain-language explanations. Core test sets are an important concept in reinforcement learning that appears in many settings. In a sequential decision-making problem, a core test set at a time point $h$ is a set of future trajectories starting from time $h$ whose probabilities conditioned on the history encode everything about the history that can affect the future. That is, if we know the vector of these probabilities $(P(q_h^j | \tau_h))_{j \in [d_h]}$, this forms a sufficient statistic of the past, encoding everything we need to know about $\tau_h$ to predict the future. Hence, the size of the core test set (i.e., number of trajectories in the core test set) is a measure of the complexity of the system dynamics.
>
> **Questions & Miscellaneous Comments**
>
> > Please define dec-POMDP, POMG before using the acronyms. Or at least define them in the Appendix.
>
> We will do this, thank you for pointing this out.
>
> > Why in def 5 we need to remove all the arrows pointing to actions? Can actions be dependent on history?
>
> Yes, actions can depend on observation histories in an arbitrary way. This is specified by the information sets of the action variables in our models. For example, in a multi-agent setting, the information sets may specify that each agent can choose an action based on its *own* history of observations and prior actions (or some limited memory of its history of observations/actions, etc.). However, these information sets *do not* affect the size of the key quantity $\mathbb{I}_h^\dagger$ that we refer to as the "information structural state space". In fact, this is a crucial aspect of the information-structural causal analysis, as it greatly reduces the implied statistical complexity. The reason for this is that we only need to model the *system* dynamics (i.e., the dynamics of system variables). In causal language, removing the arrows pointing to actions corresponds to performing a "do" operation on the actions. Under this lens, the core action sequences identified through our information structural analysis are a minimal set of interventions that allow us to learn and model the system.
>
> >Line 268. If I understand correctly $G_h$ is possibly a non-square matrix, what’s an eigenvalue in this context? Do the authors mean singular value as they speak about rank?
>
> This is a typo. We mean singular value. $\sigma_k(A)$ refers to the $k$-th singular value of the matrix $A$. Thank you for pointing this out, we will fix it.
>
> ---
>
> We hope our responses address your concerns and look forward to further discussion.

---

> > ### Comment · Reviewer_w9Kg · 2024-08-11
> >
> > Thank you for a detailed response! Many of the concerns have been addressed, but I haven't found an answer to the following question:
> >
> > > Is there an example of a system that is POST but not POMG or not dec-POMDP as claimed in Figure 1?
> >
> > This may be a trivial and a naive question, but it is not clear to me that POST structure is actually more general than dec-POMDP without a proof. In any case, it would be a good illustration if it is a trivial question

---

> > > ### Author Response · Authors · 2024-08-12
> > >
> > > Thank you for the continued discussion and engagement with our work! We are delighted that many of your concerns have been addressed.
> > >
> > > We are eager to answer your remaining questions and address any remaining concerns. First, we point to Appendix D which provides detailed discussion on the examples of information structures depicted in Figure 2. In particular, for each example of information structure depicted in Figure 2, we provide
> > > 1. a formal description of how it can be represented in the POST/POSG framework (i.e., what is the variable structure, information structure, etc., as per Def 4), and
> > > 2. a discussion of what our main results imply about the statistical complexity of each information structure. That is, it identifies information structural state and its size (hence, the statistical complexity, as per Theorem 3).
> > >
> > > With the additional content page allowed in the camera-ready version, we will incorporate a revised version of this section into the main body of the paper, which we hope will help to ground our general and abstract results in concrete examples.
> > >
> > > > This may be a trivial and a naive question, but it is not clear to me that POST structure is actually more general than dec-POMDP without a proof. In any case, it would be a good illustration if it is a trivial question
> > >
> > > The fact that POST/POSG captures special cases like Dec-POMDP is indeed direct, given the definition. We hope that reading through Appendix D will help elucidate how the POST/POSG framework captures arbitrary information structures, including commonly studied ones like Dec-POMDPs. In particular, we point to the subsection titled "Illustration: translating to the POST/POSG framework" in Appendix D which works through how a POMDP can be explicitly represented within the POST framework.
> > >
> > > To keep the response self-contained, we also provide a summary of how Dec-POMDP can be represented as an instantiation of a POST here.
> > >
> > > ---
> > >
> > > **Representation of Dec-POMDP in the POST framework.**
> > >
> > > Consider a $n$-agent Dec-POMDP given by $(\mathbb{S}, \\{\mathbb{O}^i\\}\_{i \in [n]}, \\{\mathbb{A}^i\\}\_{i \in [n]}, \mathrm{Tr}, \\{\mathrm{Ob}^i\\}\_{i \in [n]})$. The sequence of variables are given by
> > > $$s\_1, o\_1^1, ..., o\_1^n, a\_1^1, ..., a\_1^n, s\_2, o\_2^1, ..., o\_2^n, a\_2^1, ..., a\_2^n, ...$$
> > > Note that the order of observation or action variables for different agents within one time step is arbitrary since they occur simultaneously. In the POST framework, we reindex and denote all variables by "$X_t$". That is, this sequence is denoted by $X\_1, X\_2, ...$, where each $X\_t$ is either a state or an observation/action for some agent. In the POST framework, the variable structure is defined in terms of the variable indices $\mathcal{S, O, A} \subset [T]$, where $T$ is the total time horizon (counting each variable separately). However, we can slightly abuse notation by indexing variables via the notation $s\_t, o\_t^i, a\_t^i$ in order to avoid the messiness of translating each into an integer index.
> > >
> > > With this, we can define each component of the POST:
> > > 1. Variable structure: the "system variables" are $\mathcal{S} = \\{s\_t, o\_t^i, t \in [T], i \in [n] \\}$ (i.e., all state variables and observation variables for all agents); the "action variables" are $\mathcal{A} = \\{a\_t^i, t \in [T], i \in [n]\\}$.
> > > 2. Information structure: the information set of the state variable contains the state variable and joint action at the previous time step, $\mathcal{I}(s\_t) = \\{s\_{t-1}, a\_{t-1}^1, ..., a\_{t-1}^n\\}$; the information set for observation variables of each agent are contains the state variable at that time step, $\mathcal{I}(o\_t^i) = \\{s\_{t}\\}$; the information set for the action variables of each agent contains that agent's history of observations and actions, $\mathcal{I}(a\_t^i) = \\{o\_{1:t}^i, a\_{1:t-1}^i\\}$.
> > > 3. System kernels: Each system variable in $\mathcal{S}$ is either a state variable or an observation variable for some agent. If it is a state variable, the system kernel is the state transition kernel $\mathrm{Tr} \in \mathcal{P}(\mathbb{S} | \mathbb{S} \times \mathbb{A})$. If it is an observation variable, then it is the emission kernel corresponding to that agent $\mathrm{Ob}^i$.
> > > 4. Decision kernels: These are the policies of each agent. For each action in $\mathcal{A}$ there is a decision kernel. In the case of Dec-POMDPs, each action corresponds to a time point $t$ and an agent $i$. It is defined as a kernel mapping $(o\_{1:t}^i, a\_{1:t-1}^i)$ to an action in $\mathbb{O}^i$.
> > > 5. Observability: The observable system variables are $\mathcal{O} = \\{o\_t^i, t \in [T], i \in [n] \\} \subset \mathcal{S}$.
> > > 6. Reward function: this is an arbitrary function of the observables and actions (of all agents). In the case of a team, the reward is the same across all agents.
> > >
> > > ---
> > >
> > > We hope this clarifies things. Please let us know whether your concerns are address or if you have any additional questions.

---

> > > > ### Comment · Reviewer_w9Kg · 2024-08-12
> > > >
> > > > Unfortunately, this does seem to answer my question. If I understood correctly, this shows Dec-POMDP $\subseteq$ POST, but not Dec-POMDP $\subset$ POST. This doesn't show that there's a POST system that's not dec-POMDP. Moreover, there may be a system that you can formulate as POST that doesn't have dec-POMDP form, but a reformulation gives dec-POMDP. Again this may be a naive question, but it is important to answer.

---

> > > > > ### Author Response · Authors · 2024-08-12
> > > > >
> > > > > Thank you for the prompt response. Apologies for the misunderstanding, as we thought you were looking for a demonstration of how Dec-POMDP can be represented as a POST.
> > > > >
> > > > > There is essentially no restriction on the type of dynamics that can be represented by the POST/POSG framework since it allows for *arbitrary* information structures, arbitrary variable structures, and arbitrary transition dynamics. Any system that does not have a Markovian state is an example of a system that can be represented as a POST but not a POMDP/Dec-POMDP/POMG. Figure 2 and Appendix D give some examples of this (e.g., two examples we give are a mean-field information structure and an information structure associated with a point-to-point real-time communication with feedback). However, we emphasize that the key is not merely the generality of the model, but its ability to represent structure in a way that enables a richer analysis.
> > > > >
> > > > > You may be familiar with the Witsenhausen intrinsic model from the control literature: our POST/POSG framework is closely related and builds on it by introducing additional components to better-represent the reinforcement learning setting and notions of partial-observability with respect to the learning algorithm.
> > > > >
> > > > > Your question about reformulations is important to this discussion. Of course, it will often be possible to re-formulate a problem as POMDP (or even an MDP). For example, a system with arbitrary dependence on the past can be reformulated into an MDP by defining the state as an encoding of the entire past at each point in time. Similarly, a POMDP can be translated to belief-MDP. However such reformulations result in an explosion in model size, losing the ability to analyze statistical complexity. Thus, representing information structure explicitly in a general model is essential to analyzing statistical complexity in terms of the true model structure.
> > > > >
> > > > > Although POST/POSG can represent arbitrary sequential decision-making problems in environments with arbitrary dynamics and information structures, many instances of POST/POSG will have unfavorable statistical complexity. The primary contribution of our work is to characterize the statistical complexity of arbitrary decision-making problems as a function of the information structure of the underlying system dynamics.

---

> > > > > > ### Comment · Reviewer_w9Kg · 2024-08-12
> > > > > >
> > > > > > I see your point regarding the formulations and reformulations. I agree that increasing the size of the system (MDP, POMDP) is not a favorable solution.
> > > > > >
> > > > > > > However such reformulations result in an explosion in model size, losing the ability to analyze statistical complexity.
> > > > > >
> > > > > > This is not a straightforward claim to me. I am not sure that the same size reformulations would not exist for say POMDPs (in the single agent setting). Especially, if we are talking about specific examples.
> > > > > >
> > > > > > On the flip side, Theorem 3 seems to indicate that there are non POMDP information structures of similar to POMDP complexity (if I understood correctly). Furthermore, complexity is not directly defined by the cardinality of the state-space, which is an advantage.
> > > > > >
> > > > > > In summary, I see your point and I agree with your claims in Figure 1. I recommend adding a remark on MDP/POMDP reformulations in the Appendix. I think it's a more complicated question and maybe warrant a further investigation. In any case, seems to be out of scope of this work.
> > > > > >
> > > > > > I will have to think a bit more about the score, apologies if this will be after the rebuttal deadline.

---

> > > > > > > ### Author Response · Authors · 2024-08-12
> > > > > > >
> > > > > > > Thank you for your continued engagement.
> > > > > > >
> > > > > > > > On the flip side, Theorem 3 seems to indicate that there are non POMDP information structures of similar to POMDP complexity (if I understood correctly). Furthermore, complexity is not directly defined by the cardinality of the state-space, which is an advantage.
> > > > > > >
> > > > > > > Yes, that's correct. We are glad that you appreciate this.
> > > > > > >
> > > > > > > > In summary, I see your point and I agree with your claims in Figure 1. I recommend adding a remark on MDP/POMDP reformulations in the Appendix. I think it's a more complicated question and maybe warrant a further investigation. In any case, seems to be out of scope of this work.
> > > > > > >
> > > > > > > We appreciate this feedback. We agree that this will be interesting, and we will aim to add such a discussion to the appendix.
> > > > > > >
> > > > > > > > I will have to think a bit more about the score, apologies if this will be after the rebuttal deadline.
> > > > > > >
> > > > > > > We understand. If you have any further questions or concerns before the discussion period ends, please don't hesitate to ask! Thank you again for your questions and your feedback.

---

### Official Review · Reviewer_a41N · 2024-07-13

**Soundness:** 3
**Presentation:** 3
**Contribution:** 3
**Rating:** 7
**Confidence:** 4

**Summary:**

This paper studies the tractability of modeling and learning general sequential decision-making problems with an explicit representation of information structure. This gives a unifying framework that captures a range of commonly studied RL models. Through a graph-theoretic analysis it characterizes the complexity of observable dynamics in sequential decision-making problems. It then gives an upper bound on the sample complexity of learning these problems characterized by their information structure.

**Strengths:**

- The paper is well-written and well-structured. The proposed framework of partially-observable sequential teams (POST) and partially-observable sequential games (POSG) incorporates more general system dynamics than classical MDP/POMDP models through much more general information structures. The model is also practically relevant for application with complex and time-varying dependence on the past or time-varying observations dynamics.
- The paper provides a thorough theoretical foundation to understanding the complexity and tractability of POST and POSG problems. This includes characterizing the complexity of the observable system dynamics, as well as upper bound of on the statistical hardness of learning both through the lens of the information structure. Moreover the results recovers known sample complexity results in POMDPs.
- This paper gives a key insight that the complexity of the dynamics and the statistical hardness of sequential decision-making problems can be characterized in terms of its information structure. The proposed method of observation generalized predictive state representation is also of independent interest for other adjacent problems.

**Weaknesses:**

- The m-step $\mathcal{I}^{\dagger}$ -weakly revealing condition can be stringent. As the paper points out, the condition will be harder to satisfy when $\mathbb{I}_h^{\dagger}$ is large. It would be helpful to address the sensitivity of claimed results when this condition is not satisfied.
- The results of the paper are theoretical in nature, and it could benefit from empirical results to demonstrate the effectiveness of the proposed algorithm and framework on experiments / simulations.

**Questions:**

- How realistic are the assumptions required for the theoretical results, e.g. the m-step $\mathcal{I}^{\dagger}$ -weakly revealing condition, the $\alpha$-robust weakly-revealing property etc.? E.g., what are the some of the implications of these assumptions in a practical context?
- The paper shows that the sample complexity upper bound in Theorem 3 for POST implies a reasonable bound for the case of $\alpha$-weakly POMDPs. Does this sample complexity upper bound also give reasonable upper bounds for other well-studies cases incorporated by the POST framework such as MDPs, Markov games, Dec-POMDPs?

**Limitations:**

The author(s) have adequately addressed the limitations of the work.

---

> ### Author Rebuttal · Authors · 2024-08-07
>
> Thank you for your engagement with our work and your valuable feedback. We are glad our work resonated with you.
>
> We hope to address your concerns and answer your questions below, in turn.
>
> ---
>
> **On the identifiability condition and hardness results**
>
> > The m-step $\mathcal{I}^\dagger$-weakly revealing condition can be stringent. As the paper points out, the condition will be harder to satisfy when $\mathbb{I}_h^\dagger$ is large. It would be helpful to address the sensitivity of claimed results when this condition is not satisfied.
>
> > How realistic are the assumptions required for the theoretical results, e.g. the m-step $\mathcal{I}^\dagger$-weakly revealing condition, the $\alpha$-robust weakly-revealing property etc.? E.g., what are the some of the implications of these assumptions in a practical context?
>
> Our results are of an "upper bound" nature, stating that if the identified condition in terms of information structure holds, then learning is tractable and the statistical complexity scales according to $|\mathbb{I}_h^\dagger|$.
>
> We note that the $\mathcal{I}^\dagger$-weakly revealing condition can be thought of as a statistical identifiability condition. Such an identifiability condition is *necessary*, and is reflective of the fundamental difficulty of the partially-observable setting. For example, in the case of POMDPs, there exist hardness results that state that if an analogous condition does not hold, the statistical complexity can scale exponentially with the relevant quantities (i.e., size of observation/action space).
>
> However, in a practical context, if such a condition does not hold for the true underlying system, it may still hold "approximately". That is, there may exist a statistical model inside the POST/POSG framework that approximates the true system in total variation distance and which is $\mathcal{I}^\dagger$-weakly revealing. In such a case, the implications of our analysis still hold, after accounting for approximation error.
>
> We will add a discussion to the paper on these issues, emphasizing the need for such an identifiability condition, and the practical implications. Thank you for the question.
>
> **Practical implications**
>
> As you know, the primary contribution and focus of our work lies in the development of a foundational understanding of the statistical complexity of reinforcement learning in terms of information structure. The proposed algorithm is developed as a proof device rather than being intended for practical use. However, **our work has direct implications to *representation learning* in the context learning sequential decision-making problems**. When learning in arbitrary dynamic decision-making environments, our analysis reveals that the fundamental quantity that dictates the statistical complexity of a problem is the size of the information structural state space, $|\mathbb{I}_h^\dagger|$. We note that recent work in machine learning has studied the ability of sequence models like Transformers to learn and represent causal structures in sequential data. Our analysis suggests that the right scaling of the statistical complexity of such tasks, when applied to dynamic decision-making settings, depends on the quantity $|\mathbb{I}_h^\dagger|$. The analysis in this work can inform the design of architectural inductive biases that can better capture and represent such causal structures in order to achieve improved sample efficiency.
>
> We will add further discussion on these practical implications.
>
> **Upper bounds implied for well-studied special cases**
>
> > Does this sample complexity upper bound also give reasonable upper bounds for other well-studies cases incorporated by the POST framework such as MDPs, Markov games, Dec-POMDPs?
>
> Yes. The information-structural upper bound established in this work identifies the right quantities with which the statistical complexity scales. We briefly mention this for the case of POMDPs and Dec-POMDPs/POMGs in lines 354-358. This also applies for MDPs and MGs, where $\alpha = 1$ and $\mathbb{I}_h^\dagger = \mathbb{S} \times \mathbb{A}$. However, we note that our analysis recovers the right scaling up to polynomial dependence, but a more tailored analysis can identify tighter polynomial dependence.
>
> ---
>
> We hope our responses address your concerns and look forward to further discussion.

---

> > ### Comment · Reviewer_a41N · 2024-08-11
> > **Thank you**
> >
> > I would like to thank the authors for the detailed response. I think most of my questions are well addressed. I don't have any further questions.
> >
> > I think the author has given a detailed response to Reviewer w9Kg regarding possible extensions to continuous spaces, the importance of information structure and the potential practical relevance of the proposed complexity measure. Reviewer Ugr3 has pointed out a few prior work on general frameworks for statistical complexity of sequential decision making problems. I agree that the paper could benefit from better explanation of its relation to related works in this area. I think the authors have given a reasonable response to situate this work with respect the literature in the rebuttal.
> >
> > I still think this is an interesting work and I will keep my score for the moment.

---

> > > ### Author Response · Authors · 2024-08-12
> > >
> > > We sincerely thank the reviewer for their response. We are delighted that you found most of your questions were well addressed. We also greatly appreciate you being engaged with the discussions with the other reviewers and are glad to hear you found our responses to their comments to be compelling as well.
> > >
> > > We would like to thank you again for your feedback and your engagement! If you have any remaining questions before the end of the rebuttal period, please don't hesitate to ask.

---

### Decision · Program_Chairs · 2024-09-25

**Decision:**

Accept (poster)

**Comment:**

This paper proposes a fairly general formulation of sequential decision problems that exploits so-called “information structure,” which is essentially a catch-all term to describe conditional dependencies of “state” variables and “agent observation” variables on past variables. The key result is a characterization (upper bound) of the sample complexity of RL in such settings in a way that exploits the properties of the graphical model of this information structure. The reviewers raised a number of questions, but the exchange with the reviewers seems to have resolved most of these issues.

On my reading of the paper, however, there is a significant deficiency that makes the paper fully unacceptable unless it is addressed by the authors. There is a massive body of literature on exploiting the variable independence structure (in action, state-dynamics and observation space) of MDPs, POMDPs, DEC-POMDPs, etc. using factored MDP representations. The authors claim that problems have only been studied that assume uniform/regular independence structure or ignore it altogether. Nothing could be further from the truth. For example, the rich literature on using graphical models to represent MDPs, POMDPs, etc. using dynamic Bayesian networks (DBNs) does exactly what the authors claim does not exist. Moreover, numerous algorithms for solving sequential decision problems in a way that exploits these representations have been proposed for the past 30 years (at least since 1994). In addition, various papers study the RL variants of these problems (sometime focused on regret analysis, sometimes sample complexity).

The literature is too broad to do justice here, but for example, see:

* The extensive survey of very early work in this area: Decision-Theoretic Planning: Structural Assumptions and Computational Leverage, Boutilier, Dean, Hanks, J. of AI Research, 1999.

* The voluminous body of work that followed extending these to richer classes of MDPS, POMDPs, dec-POMDPs, games, etc.

* See also early work on influence diagrams (cited in the work above) that focuses especially on observation structure.

* From an RL perspective, the work on regret analysis and sample complexity of RL in factored domains. Just one example: Osband, van Roy, NeurIPS 2014.

I do believe the contribution of this paper is valuable due to the level of generality in brings to the formulation and analysis. But the lack of awareness of precedent is scientifically unacceptable. Unless this paper explains how it differs from past work (and, ideally, how it generalizes existing models, which I think it does), it cannot be published in a forum like NeurIPS.

I note the reviewers raise a number of other questions that should be addressed in revision as well.